# A global 5km monthly potential evapotranspiration dataset (1982–2015) estimated by the Shuttleworth-Wallace model

Shanlei Sun[1], Zaoyin Bi[1], Jingfeng Xiao[2], Yi Liu[3], Ge Sun[4], Weimin Ju[5], Chunwei Liu[6], Mengyuan Mu[7], Jinjian Li[8], Yang Zhou[1], Xiaoyuan Li[1], Yibo Liu[6], Haishan Chen[1]

[1]Collaborative Innovation Center on Forecast and Evaluation of Meteorological Disasters/Key Laboratory of Meteorological Disaster, Ministry of Education/International Joint Research Laboratory on Climate and Environment Change, Nanjing University of Information Science and Technology, Nanjing, China

[2]Earth Systems Research Center, Institute for the Study of Earth, Oceans, and Space, University of New Hampshire, Durham, USA

[3]School of Civil and Environmental Engineering, University of New South Wales, Sydney, Australia

[4]Eastern Forest Environmental Threat Assessment Center, Southern Research Station, USDA Forest Service, Raleigh, USA

[5]International Institute for Earth System Science, Nanjing University, Nanjing, China

[6]Jiangsu Key Laboratory of Agricultural Meteorology, School of Applied Meteorology, Nanjing University of Information Science and Technology, Nanjing, China

[7]ARC Centre of Excellence for Climate Extremes and Climate Change Research Centre, University of New South Wales, Sydney, Australia

[8]School of Atmospheric Sciences/Plateau Atmosphere and Environment Key Laboratory of Sichuan Province, Chengdu University of Information Technology, Chengdu, China

*Correspondence to*: Shanlei Sun (sun.s@nuist.edu.cn)

**Abstract.** As the theoretical upper bound of evapotranspiration (ET) or water use by ecosystems, potential ET (PET) has always been widely used as a variable linking a variety of disciplines, such as climatology, ecology, hydrology, and agronomy. However, substantial uncertainties exist in the current PET methods (e.g., empiric models and single-layer models) and datasets, because of unrealistic configurations of land surface and unreasonable parameterizations. Therefore, this study comprehensively considered interspecific differences in various vegetation-related parameters (e.g., plant stomatal resistance and $CO_2$ effects on stomatal resistance) to calibrate and parametrize the Shuttleworth-Wallace (SW) model for forests, shrubland, grassland and cropland. We derived the parameters using identified daily ET observations with no water stress (i.e., PET) at 96 eddy covariance (EC) sites across the globe. Model validations suggest that the calibrated model could be transferable from known observations to any location. Based on four popular meteorological datasets, relatively realistic canopy height and time-varying land use/land cover and Leaf Area Index, we generated a global 5 km ensemble mean monthly PET dataset that includes two components of potential transpiration (PT) and soil evaporation (PE) for the 1982–2015 time period. Using this new dataset, the climatological characteristics of PET partitioning and the spatio-temporal changes in PET, PE and PT were investigated. The global mean annual PET was 1198.96 mm with PT/PET of 41% and PE/PET of 59%, and moreover controlled by PT and PE over 41% and 59% of the globe, respectively. Globally, the annual PET and PT significantly ($p<0.05$) increases by 1.26 mm/yr and 1.27 mm/yr over the last 34 years, followed by a slight decrease in the annual PE. Overall, the annual PET changes over 53%

of the globe could be attributed to PT, and the rest to PE. The new PET dataset may be used by academic communities and various agencies to conduct climatological analyses, hydrological modelling, drought studies, agricultural water management, and biodiversity conservation. The dataset is available at https://doi.org/10.11888/Terre.tpdc.300193 (Sun et al., 2023).

## 1 Introduction

Potential evapotranspiration (PET) is the maximum amount of water that can be transferred to the air from a given land cover (e.g., land and water), providing an upper limit of the evaporative losses from this land cover (Allen et al., 1998; Milly and Dunne, 2016; Xiang et al., 2020). Commonly, it consists of potential evaporation from soils (PE) and/or transpiration by plants (PT) when the soil water supply for evapotranspiration (ET) process is non-limiting (Thornthwaite, 1948; Xiang et al., 2020). The spatio-temporal differences of PET mainly depend on those of climatic conditions, including net radiation, wind speed, the atmospheric vapour deficit, and thus PET is usually regarded as an accepted proxy for the atmospheric evaporative demand. Additionally, PET has been widely used to estimate actual ET (Sun et al., 2011a; Rao et al., 2011; Liu et al., 2017), a critical variable that links water, energy, and carbon cycles (Sun et al., 2011b), and thus is a key variable for a variety of disciplines, such as climatology, ecology, hydrology and agronomy (Allen et al., 1998; Espadafor et al., 2011; Beven, 2012).

Historically, numerous PET models have been proposed (Singh and Xu, 1997; Xu and Singh, 2000, 2001; Xiang et al., 2020). In general, the PET models can be grouped into four types: mass-transfer-based ones (e.g., Dalton-type models in Table S1; Singh and Xu, 1997), which are based on Dalton's law and take observed wind speed and water vapour pressure as inputs; temperature-based ones (e.g., Thornthwaite equation in Table S1; Thornthwaite, 1948), which take temperature as a proxy for the radiative energy available, along with extraterrestrial radiation estimated from the date of the year and latitude; radiation-based ones (e.g., Turc and Hargreaves models in Table S1; Turc, 1961; Hargreaves and Samani, 1983), which use measured data such as net solar radiation, sunshine hours or cloudiness factors; and combination ones (e.g., Penman-Monteith models, including its original type and variants in Table S1; Penman, 1948; Monteith, 1965; Allen et al., 1998), which combine the energy balance with the mass transfer method. Despite low requirement of climatic variables for the former three types of the PET model, they are lack of comprehensive physical considerations of ET process and heavily rely on empirical factors which are dependent on historical or present-day climate for calibration (Tabari and Talaee, 2011; Aschonitis et al., 2015; Tanguy et al., 2018; Xiang et al., 2020). By contrast, the combination models (e.g., Penman-Monteith models) involve relatively comprehensive physical basis, and thus have been widely used by various scientific communities (McVicar et al., 2007; Mu et al., 2013; Sun et al., 2017, 2022). As one of the most famous Penman-Monteith model variant, the Food and Agriculture Organization of the United Nations (FAO)-56 Penman-Monteith model has been validated against lysimeter data across the globe, obtaining reliable results (Jensen et al., 1990; Itenfisu et al., 2000; Berengena and Gavilán, 2005; Trajkovic, 2007; Liu

et al., 2017; Gong et al., 2017), and recommended as a standard tool for calculating PET with climatic data by the International Commission on Irrigation and Drainage (ICID), the FAO and the American Society of Civil Engineers (ASCE).

Despite the relatively satisfactory performance, the Penman-Monteith models still have inherent shortages for parameterization scheme of land surface. For example, these models set the evaporating and/or transpiration surfaces as a whole (i.e., a so-called "big-leaf"), regardless of differences in processes of soil evaporation and plant transpiration (Stannard, 1993; Yang and Shang et al., 2012; Liu et al., 2015). Over a large region, however, the big leaf assumption is rarely valid. Usually, many vegetation types co-exist over the land, and there are always some parts or periods where or when the vegetation not "closed" (i.e., open canopy that light can penetrate to the ground). The big leaf assumption potentially limits the applicability of the Penman-Monteith models under various vegetation distribution conditions, e.g., better (worse) performance under complete and homogeneous (sparse and inhomogeneous) vegetation distribution conditions (Shuttleworth and Wallace, 1985; Stannard, 1993; Yang and Shang, 2012; Huang et al., 2020). Comprehensively considering differences in processes of soil evaporation and plant transpiration, Shuttleworth and Wallace (1985) extended the Penman-Monteith single-layer models to two-layer model [i.e., Shuttleworth-Wallace (SW) model]. This model divided ET into plant and soil components based on surface resistances to regulate the heat and mass transfer from plant and soil surfaces, as well as aerodynamic resistance to regulate fluxes to the atmosphere (Lagos et al., 2013; Liu et al., 2015; Zhao et al., 2015; Huang et al., 2020). Relative to the Penman-Monteith models, the SW model is the first analytical model combining transpiration and soil evaporation by formulating the different media via which evaporative flux travels as resistances (Kool et al., 2014). This partitioning is crucial for reasonably describing and understanding ET processes (Zhou et al., 2016, 2018). The SW model is generally considered to be more accurate and more physically-based, and has been extensively used at point and regional scales (Brisson et al., 1998; Hu et al., 2009; Kool et al., 2014; Liu et al., 2015; Huang et al., 2020).

It should be noted that there are two major difficulties to run the SW model. Firstly, it has high requirement for meteorological data inputs, including the maximum and minimum air temperatures, relative humidity, wind speed and solar radiation or their proxy data. Commonly, the maximum and minimum air temperatures are measured at meteorological sites, while observations of the other elements are scare, especially when long time series and large spatial coverage are required for climate studies. This may be a major reason that some temperature-based and radiation-based PET models (e.g., Priestley-Taylor and Hargreaves-Samani models) are still widely-used at present, especially for regions with sparse meteorological observations (Aschonitis et al., 2017; Tanguy et al., 2018). Secondly, the SW model is hard to parameterize given the large number of parameters (Brisson et al., 1998; Odhiambo and Irmak, 2011; Kool et al., 2014), and therefore this model is often applied at point and regional scales (Iritz et al., 1999; Yang and Shang, 2012; Liu et al., 2015; Huang et al., 2020; Chen et al., 2022).

Fortunately, with the development of observation and numerical simulation technology during the past decades, various global datasets have become available, including meteorological datasets (e.g., Harris et al., 2020; Molod et al., 2015; Hersbach et

al., 2020; Beck et al., 2022), vegetation height datasets (e.g., Simard et al., 2011; Wang et al., 2016; Potapov et al., 2020; Lang et al., 2021, 2022), land use/land cover (LULC) datasets (e.g., Liu et al., 2020), Leaf Area Index datasets (LAI; e.g., Friedl et al., 2010; Liu et al., 2012; Zhu et al., 2013; Xiao et al., 2014), and eddy covariance (EC) flux observations across various ecosystems (e.g., LaThuile, https://fluxnet.org/data/la-thuile-dataset/; FLUXNET2015, http://fluxnet.fluxdata.org/data/fluxnet2015-dataset/; OzFlux, https://data.ozflux.org.au/portal/home.jspx; and AsiaFlux, http://asiaflux.net/?page_id=23). These released datasets provide a valuable opportunity for parameterizing and driving the SW model at the global scale.

Previous studies have suggested that land surface properties could play a dominant role in controlling variations of the ET process (Sun et al., 2021, 2022). Of the land surface properties, vegetation is the most ever-changing due to plant growth, natural disturbances and anthropogenic disturbances (Liu et al., 2016a, 2016b; Papagiannopoulou et al., 2017; Cavalcante et al., 2019; Zhang et al., 2020). Recently, with climate change and/or intensified human activities, vegetation has greatly changed on regional and even the global scales (Zhu et al., 2016; Chen et al., 2019), including shifts in vegetation types and vegetation greening (i.e., increases in LAI or other vegetation indices), which have altered the allocation of available water and energy (Zhou et al., 2016, 2018; Sun et al., 2022). As an important biophysical parameter of vegetation, the plant stomatal resistance has been widely evidenced to increase with the elevated atmospheric $CO_2$ concentration by numerous observations and numerical simulations, but the increasing magnitudes differed among vegetation types (Wand et al., 1999; Medlyn et al., 2001; Norby et al., 2005; Franks and Beerling, 2008; Lin et al., 2015; Gardner et al., 2022). The increased plant stomatal resistance could in turn reduce plant transpiration and thereby change the water cycle on regional and even global scales (Gedney et al., 2006; Piao et al., 2007; Sun et al., 2014; Zhao and Cao, 2022; Zhan et al., 2022). Especially, such impacts of elevated $CO_2$-induced plant stomatal resistance increase on hydrological processes have been proved to be of significance, especially for the future climate change scenarios with the high $CO_2$ emission (Roderick et al., 2015; Milly and Dunne, 2016; Scheff, 2018; Yang et al., 2019). These suggested that the temporal changes in vegetation (i.e., vegetation types, LAI and plant stomatal resistance) should be involved into the models for accurate PET estimates. However, it is unfortunate that the various existing PET products, such as Climatic Research Unit (CRU) Time-Series (TS) 4.06 (CRU TS4.06; Harris et al., 2020), MOD16 (Running et al., 2017), Global Land Evaporation Amsterdam Model (GLEAM) v3.6 (Miralles et al., 2011; Marten et al., 2017), Priestly-Taylor Jet Propulsion Laboratory (PT-JPL) Model (Fisher et al., 2011) and hPET (Singer et al., 2021), did not fully considered impacts of such vegetation changes on PET, potentially resulting in biases of the estimates from the truth values and then introducing uncertainties into the studies based on these products. As a result, to produce a new PET dataset based on more reasonable parameterizations and more realistic configurations of land surface, is needed.

Through making use of various existing datasets, this study comprehensively considered spatio-temporal differences in land surfaces and elevated $CO_2$-induced biophysical effects on plant stomatal resistance to parameterize the SW model, and

produced a new global monthly PET dataset. Specifically, our objectives are: (1) to calibrate the SW model with the EC flux observations to obtain key parameters for each LULC type, and to evaluate the performance of the calibrated model, (2) to

generate a global monthly PET product (including PE and PT) from 1982 to 2015 with the calibrated SW model and various inputs, and (3) to investigate the climatological characteristics of PET partitioning and the spatio-temporal changes in PET and its two components across the globe during 1982–2015.

## 2. Material and methods

### 2.1 Data sources

**2.1.1 Eddy-covariance measurements**

For fully utilizing the existing EC measurements, this study collected the half-hourly or hourly FLUXNET2015 Tier-2 (http://fluxnet.fluxdata.org/data/fluxnet2015-dataset/), LaThuile (https://fluxnet.org/data/la-thuile-dataset/), AsiaFlux (http://asiaflux.net/?page_id=23), and OzFlux (https://data.ozflux.org.au/portal/home.jspx) datasets. Following Maes et al. (2019), we processed the four datasets to provide the necessary inputs and the ET observations without water limits (i.e., PET)

for calibrating the SW model.

First, we screened the measurements using data quality control. Sites lacking one or more of the basic measurements required for our analysis [i.e., latent heat ($LE$), sensible heat ($H$), soil heat flux ($G$), the net radiation fluxes ($Rn$), wind speed ($u$), air temperature, and relative humidity ($RH$)/actual vapour pressure ($e_a$)/atmospheric water concentration/vapour pressure deficit ($D$)] were not considered further. Considering potential impacts of surface energy imbalance on the results, the corrected half-

hourly or hourly $LE$ and $H$ (Pastorello et al., 2020) were used here. Regarding the major heat fluxes ($LE$, $H$ and $G$), good gap-filled records were retained according to the quality flags provided by the four datasets. When the flags for $Rn$ observations were unavailable, the flags of the downward shortwave radiation were used instead. Mainly due to impacts of interception loss and condensation on accuracy of $H$ or $LE$ measurements (Mizutani et al., 1997), the negative values were masked out. Likewise, all $Rn$ negative values were masked out.

Second, the half-hourly or hourly measurements were aggregated to daytime records. Only days in which more than 70% of the basic data were measured directly were retained, and days with rainfall during midnight-sunset were excluded from the analyses for removing the effects of rainfall interception. Moreover, we only retained sites with 80 or more days for the processing below. Taking 5 W/m$^2$ of top-of-atmosphere incoming shortwave radiation as a minimum threshold, the half-hourly or hourly measurements were aggregated to daytime composites, and then the daytime values were obtained through

subtracting measurements at the first and last (half-)hours for these aggregates.

Third, we identified days with no soil water limits. This study employed an energy balance-based approach to select unstressed days rather than the soil moisture criterion. The major considerations were three-fold: (a) no soil moisture measurements

existed at most of EC sites, (b) the energy balance-based method could effectively remove days in which the ecosystem is not limited by soil moisture availability but stressed by other environment factors (e.g., insect plagues, phenological leaf-out, fires, heat and atmospheric dryness stress, nutrient limitations), and (c) soil moisture may be not a good indicator of water stress due to variable rooting depth and its inaccurate measurements (Powell et al., 2006; Douglas et al., 2009; Martínez-Vilalta et al., 2014). The evaporative fraction [i.e., $EF = LE/(LE + H)$] was selected as the energy balance criterion, and we assumed that under conditions of no water limits a larger fraction of the available energy should be used for evaporating water (Gentine et al., 2007, 2011; Maes et al., 2011). Taking a site as an example, a day was identified to have no water limits when its corresponding $EF$ exceeded the 95th percentile $EF$ threshold. Notably, if fewer than 15 days fulfilled this criterion at some particular sites, the 15 days with the highest $EF$ were used as unstressed days.

Finally, the sites used for further analyses were selected. Despite usage of the corrected half-hourly or hourly $LE$ and $H$ here, we found that evident surface energy imbalance still existed on daytime with Bowen ratio between 0.60 and 1.80. For further reducing potential impacts of this issue, we retained records with Bowen ratio between 0.90 and 1.10. After this, if a certain site had fewer than 8 unstressed days, it was removed. Because the LAI and $CO_2$ concentration were unavailable at some sites, the Global LAnd Surface Satellite (GLASS) LAI and $CO_2$ concentration ($\rho_{CO2}$, ppm) observed at Mauna Loa were used instead. At last, 96 sites were retained, including 73, 5, 3 and 18 sites from FLUXNET2015 Tier-2, AsiaFlux, OzFlux, and LaThuile datasets, respectively (Figure 1). Their basic information was shown in Table S2.

### 2.1.2 Meteorological data

Meteorological inputs are necessary to generate the SW PET, and moreover for reducing uncertainties we collected four monthly or 3-hourly meteorological datasets, i.e., Multi-Source Weather (MSWX)-Past (Beck et al., 2022), National Aeronautics and Space Administration (NASA) Modern Era Reanalysis for Research and Applications (MERRA)-2 (Molod et al., 2015), European Centre for Medium-Range Weather Forecasts Reanalysis (ERA)-5 (Hersbach et al., 2020) and Climatic Research Unit (CRU) TS4.06 (Harris et al., 2020). The detailed information of these datasets is shown in Table 1. The directly used meteorological variables to drive the SW model mainly include air temperature, $u$, $D$, $Rn$ and $G$. Except for air temperature and $u$, the meteorological datasets provide different $D$-related and radiation-related variables (Table 1), and therefore different methods were utilized to estimate $D$ and $Rn$ (details in Texts S1 and S2). Regarding $G$, the mean air temperature-based method was used here (Allen et al., 1998; details in Text S3). Notably, because the CRU TS4.06 lacked direct radiation records, the algorithm of Reddy (1974) was employed here to estimate $Rn$ based on the Clouds and the Earth's Radiant Energy System (CERES) satellite-based monthly net shortwave radiation records (https://asdc.larc.nasa.gov/project/CERES) and the CRU TS4.06 cloud cover data (algorithm and validation in Text S1 and Figure S1, respectively). Moreover, due to no wind speed records in CRU TS4.06, we used the mean wind speed from the other three meteorological datasets as a proxy.

**2.1.3 Land use/land cover, LAI, saturated water content in soil and CO₂ concentration**

Considering time span of the available LULC products at present, the 1982–2015 yearly GLASS-Global Land Cover (GLC) dataset developed by Liu et al. (2020) at a spatial resolution of 5000 m was used here (Table 1), though it has a coarse LULC classification ecosystem (including cropland (CRO), forests (FR), grassland (GRA), shrubland (SHRB), tundra, barren land, and snow/ice). Some studies stated that the accuracy of the GLASS LAI is clearly better than that of others [e.g., Moderate-resolution Imaging Spectroradiometer (MODIS) LAI], especially for forested areas with more realistic and reasonable trajectories representing seasonal variations (Fang et al., 2013; Liang et al., 2013; Xiao et al., 2014, 2016, 2017). Thus, the 1981–2018 8-day GLASS Advanced Very High Resolution Radiometer (AVHRR) LAI product at a spatial resolution of 0.5º was selected in this study (Xiao et al., 2016, 2017; Table 1). The global saturated soil water content in the first soil layer (i.e., 0–0.0451 m) was collected from http://globalchange.bnu.edu.cn/research/soil5.jsp (Dai et al., 2019a, 2019b; Table 1), while the 1850–2013 monthly $CO_2$ concentration at 1° × 1° spatial resolution was downloaded from https://doi.org/10.5281/zenodo.5021361 (Cheng et al., 2022; Table 1). Because this $CO_2$ dataset missed the data in 2014 and 2015, the 1985–2021 monthly Global CO₂ Distribution (GCD) product from Japan Meteorological Agency at a 2° × 2° spatial resolution (https://www.data.jma.go.jp/ghg/kanshi/co2data/co2_mapdata_e.html; Maki et al., 2010; Nakamura et al., 2015; Table 1) was used to estimate the monthly $CO_2$ concentration in the two missing years by the linear regression method. In detail, we firstly resampled the GCD data into 1º × 1º resolution, and used the 1985–2013 GCD as independent variable and Cheng's data as dependent variable to fit the linear regressions for each month and each grid. Subsequently, based on the GCD data, these regressions were used to calculate the monthly $CO_2$ concentration at each grid in 2014 and 2015. The validation results of the established regression were in Figure S2.

**2.1.4 Vegetation canopy height**

To date, the vegetation canopy height ($h$) maps to fully cover the whole globe still lack, although it is needed for the SW model to estimate some key parameters. Therefore, we made use of the existing datasets, mainly including remote sensing-based forest $h$ datasets of Potapov et al. (2020), Wang et al. (2016), Simard et al. (2011) and Lang et al. (2020, 2022; respectively named as $h$-Potapov, $h$-Wang and $h$-Simard, and $h$-Lang; Table 1), and Spatial Production Allocation Model (SPAM) V2.0 crop distribution map (Yu et al., 2020; Table 1), to reconstruct the global vegetation $h$ maps. For each year, the detailed procedures are four-fold: (1) Mapping FR and SHRB $h$. For reducing uncertainties related to retrieval algorithms and source data, the $h$-Potapov, $h$-Wang, $h$-Simard and $h$-Lang datasets were used in this procedure. Notably, we ignored differences in periods of source data for producing the forest $h$ datasets. That is, we assumed that the changes in forest $h$ due to vegetation natural growth were limited during study period. To match GLASS-GLC, the four forest $h$ datasets were firstly resampled to a spatial resolution of 5000 m. If two or more $h$ dataset/s showed non-missing values at a certain FR (SHRB) grid cell, the average of these non-missing values represented the final $h$ at this grid cell. After this, if some FR (SHRB) grid cells still had

missing values, and then the *h* value at each of these grid cells was filled with the mean *h* at the four nearest FR (SHRB) grid cells around this grid cell. (2) Mapping GRA and tundra *h*. Although the *h*-Lang product provided GRA and tundra *h* values over few regions, there were still large areas with missing values and even large uncertainties existed (Huang et al., 2017). Therefore, we specified the GRA and tundra *h* mainly based on the Comprehensive Sequential Classification System (CSCS)-based GRA groups, and the typical GRA and tundra *h* measured at the EC sites and records from literatures. Using vegetation bioclimate characteristics and hydrothermal indictors (e.g., temperature and precipitation), the CSCS method was employed to classify the GLASS-GLC GRA and tundra into 10 groups (Li and Ma, 2009; Liang et al., 2011; Gang et al., 2016). According to the map of the CSCS-based GRA groups and locations of the EC sites, the *h* value for each CSCS-based GRA group was estimated as mean *h* from the EC sites with such GRA group, and lastly the *h* values for 7 CSCS-based GRA groups were determined. As for the remaining 3 CSCS-based GRA groups (i.e., frigid desert GRA, warm desert GRA, and tropical zonal forest steppe GRA), their *h* values were from White (1983), Suttie et al. (2005), Kadeba et al. (2015), Yin et al. (2019) and Prakash et al. (2020). The *h* values of all the CSCS-based GRA groups could be found in Table S3. (3) Mapping CRO *h*. Through overlaying the SPAM V2.0 and GLASS-GLC maps, the CRO were further classified into 42 types, and the *h* for each type was specified based on Table S4 (Allen et al., 1998). (4) Mapping *h* of barren land and snow/ice. As for barren land and snow/ice regions, the *h* was set as 0. Notably, regardless of the *h* map reconstructed at each year, this dataset could not reflect inter-annual variations of *h* except for regions with LULC changes, where the *h* values varied due to LULC changes. In the grid with LULC changes in a certain year, its new *h* value was assigned as the mean *h* value of its four nearest neighboring grids with the same LULC. An example of the reconstructed canopy *h* map in 1982 was shown in Figure S3.

**2.2. Analysis methods**

The workflow of this study mainly included three parts (Figure 2): (1) calibrating the SW model based on the identified EC observations without water limits at 96 EC sites, and validating its performance in site-calibration and cross-validation modes (cyan color in Figure 2), (2) generating the global monthly PET using the calibrated SW model with the final minimum stomatal resistance ($r_{smin}$, s/m) values and various inputs (orange color in Figure 2), and lastly (3) conducing related analyses, i.e., climatological characteristics and spatio-temporal changes of PET and its two components, and climatological characteristics of PET partitioning (green color in Figure 2).

**2.2.1 Description of the Shuttleworth-Wallace (SW) model**

Based on assumption that the water vapour arriving at the reference height is a mixture of evaporation from soil and transpiration from vegetation layer (Figure 3; Shuttleworth and Wallace, 1985, 1990), the SW model can be expressed as:

$$
\begin{cases}
\lambda ET = \lambda T_r + \lambda E & \text{(1a)} \\[4pt]
\lambda T_r = C_c PM_c & \text{(1b)} \\[4pt]
\lambda E = C_s PM_s & \text{(1c)} \\[4pt]
PM_c = \dfrac{\Delta A + (\rho c_p D - \Delta r_a^c A_{soil})/(r_a^a + r_a^c)}{\Delta + \gamma[1 + r_s^c/(r_a^a + r_a^c)]} & \text{(1d)} \\[14pt]
PM_s = \dfrac{\Delta A + [\rho c_p D - \Delta r_a^s (A - A_{soil})]/(r_a^a + r_a^s)}{\Delta + \gamma[1 + r_s^s/(r_a^a + r_a^s)]} & \text{(1e)} \\[14pt]
C_c = [1 + \dfrac{R_c R_a}{R_s(R_c + R_a)}]^{-1} & \text{(1f)} \\[12pt]
C_s = [1 + \dfrac{R_s R_a}{R_c(R_s + R_a)}]^{-1} & \text{(1g)} \\[12pt]
R_a = (\Delta + \gamma) r_a^a & \text{(1i)} \\[4pt]
R_s = (\Delta + \gamma) r_a^s + \gamma r_s^s & \text{(1j)} \\[4pt]
R_c = (\Delta + \gamma) r_a^c + \gamma r_s^c & \text{(1k)}
\end{cases}
$$

where $\lambda ET$ is the total latent heat flux (W/m$^2$), i.e., the sum of canopy ($\lambda T_r$) and vegetation latent heat fluxes ($\lambda E$), where $T_r$ and $E$ represent transpiration and soil evaporation, respectively; $PM_c$ and $PM_s$ are the closed vegetation and bare soil latent heat fluxes (W/m$^2$), respectively, which can be calculated by the Penman formulas under conditions of bare soil (i.e., EQ. 1d) and closed canopy (i.e., EQ. 1e); $C_c$ and $C_s$ represent soil surface resistance (i.e., EQ. 1f; unitless) and canopy resistance coefficients (i.e., EQ. 1g; unitless), respectively; $\Delta$ denotes the slope of the saturated vapor pressure curve (kPa/K); $\gamma$ is the humidity constant (kPa/K); $\lambda$ is the latent heat of evaporation (MJ/kg); $\rho$ is the air density (kg/m$^3$); $c_p$ is the constant pressure specific heat (1013 J/kg/K); $r_a^a$, $r_a^c$ and $r_a^s$ are the aerodynamic resistance (s/m), the vegetation boundary layer resistance (s/m) and the soil boundary layer resistance (s/m), respectively; $r_s^s$ is the soil surface resistance, while $r_s^c$ is the vegetation canopy resistance (s/m). Based on EQ. 1b (EQ.1c), $T_r$ ($E$) can be obtained with $C_c PM_c$ ($C_s PM_s$) divide by $\lambda$.

In EQs. 1b and 1c, $A$ and $A_{soil}$ are the available energy for canopy and soil layers (W/m$^2$), respectively, and are defined as:

$$
\begin{cases}
A = R_n - G & \text{(2a)} \\[4pt]
A_{soil} = R_{n,soil} - G & \text{(2b)} \\[4pt]
R_{n,soil} = R_n \exp(-k_{ex}\mathrm{LAI}) & \text{(2c)}
\end{cases}
$$

where $R_{n,soil}$ is the net radiation fluxes into the soil (W/m$^2$), and can be computed with Beer's Law (i.e., EQ. 2c; Mo et al., 2004); $k_{ex}$ represents light extinction coefficient, which varies with LULC types (Table S5).

Previous studies have stated that determining the five resistance parameters is the key to successfully run the SW model (Hu et al., 2009; Chen et al., 2022). Of these resistance parameters, $r_a^a$ and $r_a^s$ are estimated using Shuttleworth and Gurney (1990), while $r_a^c$ and $r_s^s$ are calculated based on Shuttleworth and Wallace (1985) and Brisson et al. (1998), and Villagarcía et al. (2010), respectively. Detailed information about equations for parameterizing the four resistance parameters can be found in Texts S4–6 and Table S5. Considering large uncertainties of $r_s^c$ (Fisher et al., 2005; Hu et al., 2009, 2013; Wei et al., 2020), this study used the EC measurements to calibrate its value for each LULC type, and the detailed procedures can be found in section 2.2.2.

### 2.2.2 Determination of canopy resistance ($r_s^c$)

In this study, the parameterization of $r_s^c$ is mainly based on a Jarvis-type model, which is based on the hypothesis that stomatal resistance is independently affected by every environmental variable (Jarvis, 1976; Shuttleworth and Wallace, 1985). Here, we

considered effects of LAI, $D$, air temperature ($T$, °C), soil moisture and $\rho_{CO2}$ on $r_s^c$, which correspond to stress functions of $F_1$ (Noilhan and Planton, 1989), $F_2$ (Raab et al., 2015), $F_3$ (Zhou et al., 2005), $F_4$ and $F_5$ (Neitsch et al., 2002). Notably, this study focuses on ET under condition with no soil water limits (namely PET), and thus $F_4$ is set to 1 here. The specific equations are shown as follows:

$$
\begin{cases}
r_s^c = \dfrac{r_{smin}}{LAIe \prod_{i=1}^{N=5} F_i} & \text{(3a)} \\[2ex]
LAIe = \begin{cases} LAI, & LAI \leq 2 \\ 2, & 2 < LAI < 4 \\ LAI/2, & LAI \geq 4 \end{cases} & \text{(3b)} \\[3ex]
F_1 = \dfrac{0.55 \dfrac{R_{ds}}{R_{ds,dbl}} \dfrac{2}{LAI} + \dfrac{r_{smin}}{r_{smax}}}{0.55 \dfrac{R_{ds}}{R_{ds,dbl}} \dfrac{2}{LAI} + 1} & \text{(3c)} \\[3ex]
F_2 = \exp(-0.5D) & \text{(3d)} \\
F_3 = 1 - 0.0016(24.84 - T)^2 & \text{(3e)} \\
F_4 = 1 & \text{(3f)} \\
F_5 = 1 + \left(1 - \dfrac{\rho_{CO2}}{330}\right) \Delta g_{1,CO2} & \text{(3g)}
\end{cases}
$$

where $r_{smax}$ is the maximum canopy resistance (s/m), and is fixed at 5000 s/m here (Crow et al., 2008); $LAIe$ denotes the effective LAI (Gardiol et al., 2003); $R_{ds}$ is the downward shortwave radiation (W/m$^2$); $R_{ds,dbl}$ is the minimum shortwave radiation threshold for vegetation to carry out photosynthesis (W/m$^2$), which is set to 30 W/m$^2$ for forests and 100 W/m$^2$ for other vegetation (Noilhan and Planton, 1989; Lo Seen et al., 1997); $\Delta g_{1,CO2}$ is the multiple of leaf stomatal conductance reduction when $\rho_{CO2}$ is doubled (Table S5; Morison and Gifford, 1984; Field et al., 1995; Saxe et al., 1998; Neitsch et al., 2002; Eckhardt and Ulbrich, 2003; Wu et al., 2017).

Finally, we determined the values of $r_{smin}$. Here, the Monte Carlo method was used to calibrate this parameter using the identified PET observations at each EC site (Hu et al., 2009). The following five steps were taken to determine the $r_{smin}$ values: (1) giving a rough range for $r_{smin}$ (i.e., $r_{smin}$ between 1 s/m and 500 s/m) with references to previous studies (e.g., Zhou et al., 2006; ECWMF, 2007), (2) conducting 5,000 Monte Carlo simulations with the parameter sets randomly sampled from uniform distributions within the given ranges, (3) comparing the estimated SW model-based PET (SW PET) and the EC PET after each simulation based on a validation metric of the Kling-Gupta Efficiency ($KGE$; seen EQ. 4d; Gupta et al., 2012; 5,000 parameter sets corresponded to 5,000 $KGE$ after this step), (4) selecting the mean of $r_{smin}$ with the 10 highest $KGE$ as the optimal value for each site, and (5) classifying the 96 EC sites into 4 types (i.e., FR, SHRB, CRO and GRA) based on the GLASS-GLC classification system (Liu et al., 2020), and then determining the best-fit parameter sets for a given LULC type through averaging the optimal parameter values at the sites with this LULC type. In this way, the final parameter values for FR, CRO, GRA and SHRB were obtained and illustrated in Table S6. Notably, the $r_{smin}$ for tundra, barren land and snow/ice types of the GLASS-GLC are from Zhou et al. (2006) and Zhang et al. (2008), mainly due to lack of the corresponding EC observations.

### 2.2.3 Model validation

Several validation metrics were employed to evaluate the performance of the simulated SW PET. Mean error ($ME$) provides a way to quantify the biases of the estimates relative to measurements, while Root-Mean-Square-Error ($RMSE$) describes the accuracy of estimations. Besides, correlation coefficient ($R$) and $KGE$ [with a range between $-\infty$ and 1.0 (the optimal value)] were used to measure capability of capturing the spatio-temporal variability and the overall performance of the SW PET, respectively. The metrics are expressed as:

$$\begin{cases} ME = \dfrac{\sum_{i=1}^{N}(S_i - O_i)}{N} & \text{(4a)} \\[3mm] RMSE = \sqrt{\dfrac{\sum_{i=1}^{N}(S_i - O_i)^2}{N}} & \text{(4b)} \\[3mm] R = \dfrac{\sum_{i=1}^{N}[(S_i - S)(O_i - O)]}{\sqrt{\sum_{i=1}^{N}(S_i - S)^2 \sum_{i=1}^{N}(O_i - O)^2}} & \text{(4c)} \\[3mm] KGE = 1 - \sqrt{(1 - R^2) + (1 - \alpha^2) + (1 - \beta^2)} & \text{(4d)} \end{cases}$$

where $N$ is the sample number; $S$ denotes the mean of the SW PET averaged $N$ samples; $O$ is for the measured PET; $i$ is the $i$th sample; $\alpha$ is $S/O$; $\beta$ is $\sigma_S/\sigma_O$, where $\sigma_S$ and $\sigma_O$ are the standard deviations of the estimated and the measured PET, respectively. These metrics were first computed based on the daily estimates with the optimal parameters for each site [i.e., parameters obtained in step (4) of the Monte Carlo method] and the observations from the 96 EC sites, and then validation in site-calibration mode was performed. Notably, to evaluate the transferability of the calibrated parameters from known observations to any location and then robustness of the established SW model, the "leave-one-out" cross-validation method was utilized here (Zhang et al., 2019). For each LULC type, the data from one "ungauged" observation was excluded from the Monte Carlo method-based optimization while the data from all other observations of the same LULC type were used for model calibration to obtain the simulated at the "ungauged" position. All four LULC types were actualized in this way. Then, the daily SW PET estimates in the cross-validation mode were compared against the daily observations from the 96 EC sites to further explore the model performance.

### 2.2.4 Development of 5 km global monthly PET (1982–2015) and related analyses

Considering that the SW model was calibrated with the daily EC measurements, it was necessary to examine whether this model could be applicable at the monthly scale. Therefore, we firstly compared the monthly PET estimated based on the daily and monthly meteorological variables from MSWX-Past, MERRA-2 and ERA-5 (not including CRU TS4.06 mainly due to it with a monthly scale). Various validation metrics showed that there were generally no evident differences in the two PET estimates (Figure S4). That is, the model established with the daily EC measurements could be driven using the monthly meteorological variables. Mainly due to the GLASS-GLC with the shortest time span, the global SW PET was produced at GLASS-GLC grid and monthly scale during 1982–2015. Therefore, before running the calibrated SW model, the meteorological, the GLASS LAI, and the $CO_2$ concentration datasets were resampled to a spatial resolution of 5000 m. The

monthly mean meteorological and LAI datasets were taken as inputs to run the calibrated model for estimating the monthly

mean SW PET, PT and PE. At last, the total SW PET (PT and PE) at a certain month were obtained through monthly mean

value multiplying days of this month. Additionally, to reduce uncertainties from meteorological datasets, the ensemble means

of PET, PT and PE based on four meteorological datasets were provided. Using the ensemble mean PET, PT and PE, we

analyzed climatological characteristics and spatio-temporal changes of PET and its two components, and climatological

characteristics of PET partitioning. Notably, the area-weighted method was used to estimate the regional mean PET, PT and

PE.

## 3. Results

### 3.1 Performance of established SW model

The simulated daily PET from the SW model was first evaluated against EC measurements aggregated for all of 96 EC sites

(Figures 4a and 4c). In both site-calibration and cross-validation modes, the SW model could well simulate the daily PET with

most of data points around the 1:1 line, while it should be noted that overestimation and underestimation existed for the lower

and the higher measurements, respectively (Figure 4a). Based on the selected validation metrics, we could conclude that the

model had excellent performance in the two modes, with $R$, $ME$, $RMSE$ and $KGE$ values above 0.85, between -0.03 and -0.01

mm/day, below 0.80 mm/day and above 0.85, receptively. Regarding the mean values of each site (Figures 4b and 4d), the

simulated daily PET could also well follow changes of the observed PET among 96 sites in both modes, as evidenced by $R$

above 0.88, $ME$ between -0.09 and -0.03 mm/day, $RMSE$ below 0.60 mm/day and $KGE$ above 0.85. Moreover, little changes

in the validation metrics from site-calibration to cross-validation indicated limited degradation of the calibrated model (i.e.,

Figure 4a vs. 4c, and Figure 4b vs. 4d).

Figure 5 indicates that in site-calibration and cross-validation modes the SW model estimated daily PET very well for FR,

SHRB, GRA and CRO. The model slightly overestimated lower daily PET but slightly underestimated higher daily PET. The

performance of this model slightly differed among LULC types. $R$ was above 0.83, indicating that the model could successfully

simulated spatio-temporal variability of the daily PET, especially for FR, SHRB and GRA in site-calibration mode. Although

negative and positive $ME$ existed in CRO and GRA and the other two LULC types, respectively, the $ME$ magnitudes were all

below 0.15 mm. According to $RMSE$, the simulated GRA daily PET performed best in each mode, while larger values (between

0.54 mm/day and 0.85 mm/day) occurred in other three LULC types, especially for CRO in site-calibration mode and FR in

cross-validation mode. The $KGE$ was always larger than 0.80 for each LULC type in site-calibration and cross-validation

modes, which indicated that the calibrated model overall had a high performance. When it came to validation based on site

mean values (Figure 6), the simulated PET well followed changes of the observed PET for each LUCC type in the two modes,

with $R$ above 0.74, $ME$ between -0.12 and 0.11 mm, $RMSE$ below 0.70 mm and $KGE$ above 0.74. For each LULC type, the

cross-validation mode had slightly lower performance than the site-calibration mode (Figure 6a vs. 6e, Figure 6b vs. 6f, Figure 6c vs. 6g, and Figure 6d vs. 6h).

For further evaluating the performance of the established SW model, we also computed the validation metrics at each site for each LULC type in site-calibration and cross-validation modes, which are shown in Figure 7. Except for several sites for SHRB, GRA and CRO, the $R$ values were all above 0.80 and more than 50% of the sites had a value above or around 0.90, which indicated that the model had excellent performance in capturing the temporal variability of daily PET (Figure 7a). As for $ME$, the majority of the sites for each LULC type showed a range between -0.50 mm and 0.50 mm (Figure 7b). Moreover, the daily PET for SHRB (other three LULC types) was overestimated (underestimated) by the SW model at more than 50% of the sites. Except for few sites, the $RMSE$ values were below 0.80 mm at more than 75% of the sites for each LULC type, and comparably the $RMSE$-based performance for CRO was the worst generally with $RMSE$ above 0.50 mm (Figure 7c). The majority of the sites had a $KGE$ value above 0.60, and especially for GRA and CRO, more than 75% of the sites had $KGE$ larger than 0.70 (Figure 7d). In general, the model performance in the cross-validation model was similar to only slightly lower than that in the calibration model for each LULC type.

To sum up, the above evaluation indicated that the calibrated parameter of $r_{smin}$ could be transferable from known observations to any location and then the established SW model could well simulate PET across different LULC types. This gave us confidence to employ it to produce a global PET dataset. Notably, for maximizing data utilization, the final values of $r_{smin}$ for each LULC type were determined based on all the EC observations (Table S6), and the related validation results could be found in Figure S5.

**3.2 Climatological characteristics of PET, PT and PE**

As seen from Figures 8a, 8c and 8e, the global mean climatological annual PET, PT and PE were 1198.96 mm, 481.12 mm and 717.74 mm, respectively. Compared to Northern Hemisphere (NH), larger mean climatological annual PT and thus PET appeared in Southern Hemisphere (SH) where a larger proportion of the land surface was covered by vegetation. Among the five KG climate regions, mean climatological annual values of PET and its two components showed evident differences, with a range from 319.29 mm in Polar region to 1590.57 mm in Tropical region for PET, from 37.95 mm in Polar region to 1122.42 mm in Tropical region for PT, and from 248.31 mm in Cold region to 1379.16 mm in Arid region for PE. Except for the Tibetan Plateau with a lower value, climatological annual PET was generally larger than 1000 mm between 60°S and 45°N, covering 62% of the globe (Figure 8b). Especially, climatological annual PET exceeded 1800 mm over northern Africa, Arabian Peninsula, Indian Peninsula, Indo-China Peninsula and northern Australia. The lowest climatological annual PET (< 400 mm) was generally located to the north of 60°N. As seen in Figure 8d, the spatial distribution of climatological annual PT differed from that of climatological annual PET. Due to sparse and even no vegetation and/or unfavorable climate conditions, lower climatological annual PT (< 400 mm) was widely distributed to the north of 18°N and to the south of 18°S, corresponding to

an area percentage of 55%. By contrast, 16% of the globe with larger climatological annual PT (> 1200 mm) was mainly located in Caribbean, Amazon Basin, Congo Bains and Indo-China Peninsula. As for climatological annual PE (Figure 8f), larger values (> 1000 mm) were generally clustered in northwestern North America and South America, northern Africa, western Asia and most of Australia, which corresponded to sparse and even no vegetation and covered 36% of the globe. Notably, lower climatological annual PE (< 400 mm) appeared in tropical rain forests near the Equator, possibly because the dense vegetation limited available energy for PE.

As expected, climatological monthly PET, PT and PE generally exhibited strong seasonal fluctuations because of the combined influences from many environmental factors (Figure 9). The globe and NH showed a similar seasonal cycle for climatological monthly PET, PT and PE, i.e., increasing from January, peaking in July, and afterwards declining, while an opposite seasonal fluctuation existed in SH with a valley in June (Figures 9a–c). Mainly due to differences in environmental factors, the seasonal cycles of climatological monthly PET, PE and PT obviously differed among various KG climate regions (Figures 9d–h). As for Tropical region, no evident seasonal fluctuations existed for the three variables, which slightly fluctuated around 136 mm for monthly PET, 88 mm for monthly PT and 40 mm for monthly PE, respectively (Figure 9d). In Arid, Cold and Polar regions, climatological monthly PE, PT and PE were generally characterized by a peak in July or August (Figures 9e, 9g, and 9h), while Temperate region showed larger values in April–November (Figure 9f).

### 3.3 Climatological characteristics of PET partitioning

In order to know PET partitioning, we estimated the ratios of PT and PE to PET (i.e., PT/PET and PE/PT) at annual and monthly scales (Figures 9 and 10). As depicted in Figures 10a and 10c, the global mean annual PT/PET and PE/PET were 41% and 59%, respectively, indicating that globally the annual PE greatly contributed to the annual PET. Likewise, the annual PE was also the major contributor in NH and SH, with the PE/PET of 62% and 50%, respectively. Overall, the annual PT/PET (PE/PET) had evident regional differences, and was above 53% (below 47%) in Tropical, Temperate and Cold regions but below 12% (above 88%) in other two climate regions (Figures 10a and 10c). These indicated that the annual PET in Tropical, Temperate and Cold regions (Arid and Polar regions) was controlled by PT (PE). Spatially, the annual PE/PET was above 50% mainly in regions to the north of 65ºN, western North America, the Patagonia and the Andes of South America, western and central Asia, northern and southern Africa, and Australia (Figure 10d), while the remaining regions showed PT/PET larger than 50%, especially for Amazon Basin, Congo Basin and southeastern Asia with a value exceeding to 90%. Compared to the annual PT/PET, the annual PE/PET corresponded to an opposite spatial distribution (Figure 10b). In short, the annual PET was dominated by PE and PT over 59% and 41% of the globe, respectively.

In general, the seasonal cycle of PT/PET (PE/PET) for the globe, each hemisphere, and each KG climate region was different from that of PT (PE; Figure 9). As for the globe, NH and Polar region (Figures 9a, 9b and 9h), the monthly PT/PET had two bottoms in April and October and a peak in July, corresponding to tow peaks and a bottom of the monthly PE/PET. In SH and

Temperate (Cold region), the monthly PT/PET firstly increased from January, peaked in July (June), and after that declined, while the monthly PE/PET showed opposite fluctuations (Figures 9c, 9f and 9g). No evident fluctuations of the monthly PT/PET and PE/PET existed in Tropical and Arid regions (Figures 9d and 9e). Comparing the monthly PT/PET and PE/PET in the globe, NH, Arid region and Polar region (Figures 9a, 9b, 9e and 9h), it was not difficult to find that the former was always smaller the latter, indicating that the monthly PE greatly contributed to the monthly PT. Oppositely, the larger monthly PT/PET in Tropical and Temperate regions suggested that PT dominated PET in these regions (Figures 9d and 9f). In particular, the major contributor to the monthly PET for SH (Cold region) varied throughout a year, i.e., PT dominating PET in March–July (May–September) but PE dominating PET in other months (Figures 9c and 9g).

**3.4 Spatio-temporal changes in PET, PT and PE**

Globally, annual PET and PT significantly ($p<0.05$) increased by 1.26 mm/yr and 1.27 mm/yr, respectively, with a slight and insignificant decrease in annual PE (Figures 11a, 11c and 11e). Regarding annual PET and PT, both NH and SH had significant ($p<0.05$) increases, while annual PE oppositely changed in the two hemispheres, i.e., increases for NH but decreases for SH (Figures 11b and 11c). Among various KG climate regions, annual PET (except for Tropical region) and PT were all found to significantly ($p<0.05$) increase but with evident regional differences in magnitudes, i.e., larger PET increases (>1.32 mm/yr) in Arid, Temperate and Cold regions and a maximum increase (1.83 mm/yr) in PT in Temperate region (Figures 11a, 11c and 11e). Except for Polar region, the other climate regions all showed significant ($p<0.05$) changes in annual PE, followed by decreases in Tropical and Temperate regions but increases in Arid and Cold regions. Comparisons between annual trends of PT and PE suggested that the increases in PET could be attributed to increased PT for the globe, each hemisphere and each climate region. Spatially, increases in annual PET covered 78% of the globe, accompanied by 41% of the globe with significant ($p<0.05$) increases and larger values (>4.00 mm/yr) in western US, Amazon Basin, Congo Basin, eastern Europe and eastern China (excluding northeastern part; Figure 11b). As for annual PET, only 5% of the globe had significant ($p<0.05$) reductions. No considering non-vegetation regions, the spatial pattern of annual PT trends was similar to that of annual PET trends (Figure 11d). Overall, significant ($p<0.05$) increases and decreases in annual PT were observed for 28% and 4% of the globe, respectively, especially for Amazon Basin, Congo Basin, southern Africa, Indian Peninsula, eastern Europe and eastern China (excluding northeastern part) with larger increases exceeding to 4.00 mm/yr. As shown in Figure 11f, the increasing trends of annual PE had a wide distribution but significant ($p<0.05$) increases only covered over 26% of the globe and the increases were generally below 3.00 mm/yr. Relative to annual PET and PT, annual PE had more regions showing decreases, with significant ($p<0.05$) reductions over 17% of the globe mainly in northern South America and Australia, southern Africa, and Indian Peninsula. In general, the spatial distribution of major contributors to annual PET trends was similar to that of major contributors to annual PET (Figure 11g vs. Figures 10b and 10d). For example, the major contributor of PE was mainly located in Greenland, southwestern North America, the Patagonia and the Andes of South America, western and central Asia, northern

and southern Africa, and most of Australia (Figure 11g). Oppositely, the remaining regions showed the dominant factor of PT. Overall, the annual PET changes were dominated by PT and PE over 53% and 47% of the globe, respectively.

In general, the globe mean monthly PET and PT significantly ($p<0.05$) increased throughout a year with larger trends ($> 0.10$ mm/yr) during April–October, while the global mean monthly PE insignificantly changed in each month (Figure 12a). For NH (Figures 12b), all months showed significant ($p<0.05$) increases in PET and PT, especially for March–September with trends generally larger than 0.10 mm/yr. Despite most of months with increased PE, only March had significant ($p<0.05$) increases in NH. In SH (Figure 12c), monthly PET showed larger ($> 0.10$ mm/yr) and significant ($p<0.05$) increases during June–October

but generally decreases in other months. The SH PT significantly ($p<0.05$) increased in most of months, with larger values ($> 0.12$ mm/yr) during August–October. As for the SH PE, August and October, and January–March and December corresponded to significant ($p<0.05$) increases and reductions, respectively. Among the five KG climate regions (Figures 12d–h), the monthly PET and PT increased throughout a year except for PET of Tropical region in several months with decreases, and moreover the increases in most of months were significant ($p<0.05$). When it came to the monthly PE changes, most of or all months

exhibited decreases in Tropical and Temperate regions but generally increases in other three KG climate regions. Despite that, no more than 6 months exhibited significant ($p<0.05$) increases or decreases in the monthly PE for each climate regions. Furthermore, we compared trends of the monthly PT and PE for identifying major contributors to the monthly PET changes of the globe, each hemisphere, and each KG climate region (Figure 12). Overall, the monthly PET changes were generally dominated by PT in most of or all months for the globe and each region. However, it should be noted that PE was the major

contributor of PET for some months, i.e., January–March for SH, February–May for Tropical region, March and December for Cold region, and February–May, October and November for Polar region.

## 4. Discussion

### 4.1 Advantages of this new PET dataset and its potential implications

Through considering the joint effects of various ET process-related factors, we have developed a new global PET dataset based

on the SW model in this study. This dataset has several advantages relative to existing global PET datasets, e.g., CRU TS4.06 (Harris et al., 2020), MOD16 (Running et al., 2017), Global Land Evaporation Amsterdam Model (GLEAM) v3.6 (Miralles et al., 2011; Marten et al., 2017), Priestly-Taylor Jet Propulsion Laboratory (PT-JPL) Model (Fisher et al., 2011), and hPET (Singer et al., 2021). First, the dataset considered more realistic land surface information, including spatial differences in LULC and vegetation parameters (e.g., canopy height and $r_{smin}$), and time-varying LULC and LAI datasets, leading the new PET

estimates more realistic. Second, the established SW model used in this study was based on more realistic physical processes and rendered the SW PET dataset an explicit physical significance, and meanwhile provided the PT and PE (which are crucial for understanding PET and ET. Third, a number of studies have found that the elevated atmospheric $CO_2$ concentration could

exert clear impacts on ET process through controlling plant stomatal resistance (Gedney et al., 2006; Piao et al., 2007; Sun et al., 2014; Roderick et al., 2015; Milly and Dunne, 2016; Yang et al., 2019; Zhao and Cao, 2022). Through introducing a stress function of $CO_2$ concentration on $r_s^c$ our SW PET dataset is able to reflect impacts of elevated $CO_2$ on ET.

In view of these advantages, our global PET dataset can apply to multiple properties, e.g., the analysis of rainfall, agriculture, drought, hydrology and biodiversity. First, our SW PET can provide realistic and physical datasets to further understand the impacts of spatio-temporal differences in rainfall changes from perspective of scaling effects of the Clausius-Clapeyron relation between air temperature and moisture-holding capacity (IPCC, 2014; Barbero et al. 2017). For example, through exploring the relationships between evaporative demand (even PT and PE) and rainfall, one can re-untangle different scaling effects of the Clausius-Clapeyron relation between air temperature and moisture-holding capacity. Second, considering PET partitioning into two components of PT and PE, this dataset will be convenient for the agriculture managers to directly use PT estimates for effective agricultural management practices (e.g., seeding, fertilization, irrigation planning and scheduling) and finally for sustaining the grain yield (Allen et al., 1998; Tomas-Burguera et al., 2019). Third, our SW PET dataset provides an opportunity to further understand drought mechanisms, e.g., how and what magnitudes different PET components (i.e., PT and PE), LULC changes, vegetation greening and elevated $CO_2$-indcued vegetation physiological effects (e.g., increased stomatal resistance) impact droughts, which are still unclear until now (Vicente-Serrano et al., 2020). One can use this dataset as inputs to compute drought indices [i.e., Standardized Precipitation-Evapotranspiration Index (SPEI) and self-calibrating Palmer Drought Severity Index (scPDSI); Wells et al., 2004; Vicente-Serrano et al., 2010], and separate the impacts of the aforementioned factors on drought evolution. Fourth, the PET as a crucial input for many hydrological models is closely related to performance of these models (Lu et al., 2003, 2005; Rao et al., 2011; Aouissi et al., 2016; Seiller and Anctil, 2016; Dallaire et al., 2021), and therefore more realistic physical PET estimates will benefit for accurate hydrological modellings and physically understanding hydrological responses to environment changes (e.g., changes in climate and LULC). Fifth, the new PET dataset is of significance for understanding biodiversity responses to local environment changes such as LULC, and PET usually as an effective measure of climatic energy limits organisms directly and/or influences primary productivity and thus food availability (Kerr, 2001; Hawkins et al., 2003; Phillips et al., 2010a).

## 4.2 Uncertainties

The estimated SW PET may involve uncertainties from various sources, such as parameterizations of the SW model and inputs for calibrating and driving the model. These uncertainties are discussed in the following sections.

### 4.2.1 Uncertainties in parameterizations of the SW model

A relatively simple Jarvis-type empirical formula was employed here to describe the impacts of environmental conditions (i.e., temperature, vapor pressure deficit, solar radiation, soil moisture content and $CO_2$ concentration) on $r_a^c$. Whereas the better

performance of the Jarvis-type formula in simulating the impacts of environmental conditions on water and carbon fluxes (Jarvis, 1976; Lhomme, 2001; Wang et al., 2020), the interactive effects between environmental factors on $r_a^c$ are not taken into account by this empirical formula. In reality, environmental factors are interdependent and hence their interaction in the Jarvis-type empirical formula cannot be easily separated. In one word, the "multiplication" form of the Jarvis-type empirical formula potentially biases the estimated $r_a^c$ (Damour et al., 2010; Chen et al. 2022) and then the SW PET estimates.

As an important and undetermined parameter within the Jarvis-type empirical formula for estimating $r_a^c$, $r_{smin}$ was optimized for FR, SHRB, GRA and CRO and the calibrated SW model performance was satisfactory (details in section 3.1). However, it should be noted that this optimized parameter may be the main hindrance for the SW model application over the globe (Liu et al., 2003; Mo et al., 2004; Chen et al., 2022). First, $r_{smin}$ was estimated using EC observations that generally cover the period after 2000 assuming stationary environment conditions. Environment conditions were evidenced to have greatly changed during the past several decades, especially climate conditions, such as global warming (IPCC, 2014), brightening/dimming (Wild, 2009) and frequent extreme events (e.g., droughts; Sheffield et al., 2012; Trenberth et al., 2012). Therefore, $r_{smin}$ may have inter-annual variations due to the changes in these variables (Wever et al., 2002; Winkel et al., 2001), and this finally introduces biases into the PET product, particularly for years with evidently different environment conditions relative to the period after 2000 (Aschonitis et al., 2017). Second, $r_{smin}$ was fixed throughout the year in the SW model, regardless of seasonal variations of this parameter with environmental and vegetation conditions (Winkel et al., 2001; Wever et al., 2002; Douglas et al., 2009). Therefore, no consideration of seasonal cycle of $r_{smin}$ may impact the quality of the PET product (Hu et al., 2009; Zhu et al., 2013; Elfarkh et al., 2021). For instance, Hu et al. (2009) and Zhu et al. (2013) found that ET was systematically overestimated or underestimated using the SW model with fixed parameters.

The canopy light extinction coefficient of $k_{ex}$, which represents a partitioning of radiant energy over vegetation canopy and soil surface, is a key factor affecting ecosystem carbon, water, and energy processes (Tahiri et al., 2006; Zhang et al., 2014). As a result, the accuracy of the SW model was believed to be associated with the $k_{ex}$ parameterization used in this study. Considering the physiological and morphological differences among terrestrial ecosystems (Emami-Bistghani et al., 2012; Zhang et al., 2014), we followed the popular biogeochemical models and remote sensing models of ET and gross primary productivity (e.g., Lund-Potsdam-Jena Dynamic Global Vegetation Model, and Vegetation Photosynthesis Model; Xiao et al., 2004; Sitch et al., 2003), and assumed this coefficient as a constant for every LULC type (Table S1). Despite that, it is notable that the $k_{ex}$ values vary with the growth of plant and/or vegetation coverages (Lindroth and Perttu, 1981; Aubin et al., 2000; Maddoni et al., 2001; Emami-Bistghani et al., 2012; Fauset et al., 2017). Emami-Bistghani et al. (2012) stated that with an increase of vegetation coverages, the $k_{ex}$ values decreased especially in early reproductive stage in sunflower cultivars. It suggested that the fixed $k_{ex}$ within the SW model existed limitations, potentially leading to uncertainties of the PET estimates. Implied by Tahiri et al. (2006), relative to the variable $k_{ex}$ values, the fixed values gave a less precise estimation of plant

transpiration under irrigated maize.

Noteworthily, the SW model only focuses on two processes of ET, namely soil evaporation and vegetation transpiration (Shuttleworth and Wallace, 1985). Another parts of ET, vegetation canopy interception and nighttime ET ($ET_n$) are ignored in this study. It is reported that vegetation canopy interception can occupy a certain proportion in total ET, especially for regions with high LAI and frequent rainfall (Gash et al., 1995; Tourula and Heikinheimo, 1998; Lawrence et al., 2007; Wang and Dickinson, 2012). On rainy days, the vegetation canopy interception may account for a considerable fraction of the total ET

(Tourula and Heikinheimo, 1998; Hu et al., 2009). On average, the fraction of $ET_n$ accounts for approximately 6.3% of the total ET informed by the FLUXNET2015 dataset while 7.9% based on multiple global models (Padrón et al. (2020). This fraction may exceed to 15% in mountain forest with snowy and windy winter. Despite that, to accurately represent the $ET_n$ process is still difficult to date, mainly because the related controlling mechanisms are still not clear (Han et al., 2021). For example, Novick et al (2009) and Groh et al (2019) found that VPD and wind speed had a significant impact on $ET_n$, while

Groh et al. (2019) stated that the contributions of night dew could not be ignored. As an important component of $ET_n$, the nighttime transpiration is not only related to the incomplete stomatal closure (Dawson et al., 2007; Duursma et al., 2019) but also the circular regulation of nighttime water uses by plants (De Dios et al, 2015). However, how the environmental factors alter nighttime transpiration is still disputed. Dawson et al. (2007) and Moore et al. (2008) reported a positive correlation between nighttime transpiration and VPD and soil moisture content, while Barbour and Buckley (2007), Phillips et al. (2010b)

and De Dios et al. (2015) found no or negative correlation between nighttime transpiration and the two variables aforementioned. Moreover, the biological factors (e.g., plant species and ecosystem types) can also significantly influence nighttime transpiration (O'keefe and Nippert, 2018; Zeppel et al., 2014). Therefore, to establish a common model for estimating $ET_n$ across various ecosystems remains challenging. All in all, ignoring vegetation canopy interception and $ET_n$ may underestimate PET (Tourula and Heikinheimo, 1998; Lawrence et al., 2007; Mu et al., 2011; Padrón et al., 2020; Singer et al.;

2021). Subsequent research will be done to integrate these two processes in the SW model to further enhance the model's physical mechanism.

**4.2.2 Uncertainties in datasets for calibrating and driving the SW model**

    The uncertainties in the EC observations for calibrating the SW model and in inputs for producing the global PET can be propagated into the PET estimates. As for the EC observations, the uncertainties mainly come from the selection of unstressed

560 days for obtaining the observed PET, issue of non-closure of the energy balance at the EC system level and the gap-filling methods (e.g., marginal distribution sampling method (MSD); Reichstein et al., 2005). The energy balance-based criterion employed in this study was proved to be efficient to select unstressed days (Maes et al., 2019), but this method may still result in two uncertainties. First, we should note that the higher the LAI/Normalized Difference Vegetation Index (NDVI) is, the larger the *EF* will be (Gentine et al., 2007; Nutini et al., 2014). This suggests that the larger *EF* would like to frequently happen

during the growing season, which usually correspond to higher LAI/NDVI. Therefore, the identified unstressed days tended to involve less ones within non-growing season, potentially introducing large biases in estimated PET during this season. Second, due to frequent water deficits in arid regions, the *EF* threshold may exceed the 95th percentile. What is more, there may be no unstressed days in extreme arid regions, mainly because the soil moisture can hardly reach the field capacity or saturation due to the very limited precipitation. Thus, the identified unstressed days using the energy balance-based criterion may actually include the stressed days in arid regions, and potentially biased the PET estimates. In order to reduce the impacts of the non-closure of the energy balance in the EC system (Wilson et al., 2002; Foken, 2008), we used the corrected half-hourly or hourly EC *LE* and *H* observations and only retained daytime records with Bowen ratio between 0.90 and 1.10 to calibrate the SW model. Although such processing could reduce uncertainties, the imbalance of energy was not fully solved in our study and may lead to inevitable errors in the calibrated parameter of $r_{smin}$ and therefore the global PET estimates (Hu et al., 2009; Elfarkh et al., 2020; Chen et al., 2022). In this study, for maximizing the use of data, the MSD method was employed to fill the gaps in the EC *LE* measurements. However, we should note that if the controlling thermodynamic and kinetic factors of the atmosphere and soil moisture conditions are different between the missing and retrieved moments, the gap-filled *LE* based on the MSD method may be of low confidence, especially when soil moisture has abrupt changes (Jiang et al., 2022). Recently, Jiang et al. (2022) developed a physics-based full-factorial scheme to fill gaps in ET from EC observations, and found that the gap-filled ET with this scheme showed higher confidence relative to the existing typical gap-filling methods. Therefore, to reduce the uncertainties from the MSD-based gap-filled *LE*, the physics-based full-factorial scheme could be a good candidate in the future to fill the ET gaps. Here, to quantify potential impacts of the MSD-based gap-filled values, the SW model was re-calibrated and re-validated against the data points without gap-filling. Relative to the SW model used in this study, the new $r_{smin}$ and the validation metrics changed insignificantly (Figures S6, and S7), suggesting that the uncertainties induced by the gap-filled *LE* were limited.

The SW model used various datasets as inputs, including LULC, LAI, canopy height and meteorological data (e.g., MSWX-Past, CRU TS4.06, ERA5 and MERRA-2) while with certain uncertainties (Fang et al., 2013; Xiao et al., 2017; Liu et al., 2018; Xu et al., 2018). As the reflection of vegetation growth, the accuracy of LAI can affect several key parameters (e.g., $r_a^c$, $r_s^c$, $r_a^a$, and $r_a^s$) and input variables (e.g., $A$ and $A_{soil}$) and influence the quality of the PET estimates. To reduce uncertainties from LAI datasets, this study selected the GLASS AVHRR LAI product with a better overall performance (relative to other popular products, e.g., MODIS, FSGOM, GLOBMAP and GIMMS3g; Fang et al., 2013; Xiao et al., 2017; Liu et al., 2018; Xu et al., 2018). However, we should note that this LAI product slightly underperformance over grassland compared to other products (Liu et al., 2018). Considering that this LAI product was based on the 8-day maximum value composite for removing impacts of cloudy days, the *LAIe* (based on EQ. 3b) was potentially larger than its authentic value due to some leaves covered by rain or snow. Thus, from EQ. 3b, $r_s^c$ may be slightly underestimated, leading to an overestimation in PT and PET.

Considering impacts of LULC changes on PET across the globe, we used the 1982–2015 yearly GLASS-GLC dataset developed by Liu et al. (2020) to separately estimate PET for each LULC type. Although the average overall accuracy for the 34 years each with seven classes is 83% based on 2431 test sample units, the misclassification issues still existed, e.g., Africa, eastern and southern South America, southern Alaska, northern and eastern Australia and south-western Indonesia (Liu et al., 2020). The reconstructed global vegetation canopy height also has limitations, which may raise from (1) uncertainties in the retrieval algorithms and remote sensing data (Simard et al., 2011; Wang et al., 2016; Potapov et al., 2020; Lang et al., 2021, 2022), (2) neglecting the spatial differences in CRO and GRA heights and using an alternative specific value, and (3) not considering the inter-annual changes in the FR and SHRB canopy heights and the intra-annual cycle in the CRO and GRA heights. These limitations undermine the accuracy of the PET estimates. In recent, Peng et al. (2022) proposed a practical method for global estimates of 500 m daily aerodynamic roughness length with a combination of machine learning techniques, wind profile equation, observations from 273 sites and MODIS remote sensing data. Their results showed that the random forest model could well reproduce the magnitude and temporal variability of daily aerodynamic roughness length at most sites for all land cover types. We believed that the aerodynamic roughness length produced by this method has a potential to replace vegetation canopy height as an input to run the SW model, and thus reduce the vegetation canopy height-related uncertainties aforementioned and improve the accuracy of the PET estimates. A series of evaluations of various meteorological datasets across the world suggested the discrepancies among these datasets (Urraca et al., 2018; Hinkelman, 2019; Jourdier, 2020; Zhang et al., 2021). Although the ensemble mean used here may reduce uncertainties from the meteorological datasets, there is still likelihood that the remaining uncertainties might be propagated into the PET estimates. In this study, even though soil does experience water stress, we assumed that the soil water supply for ET was unconstrained in estimating PET. As a result, the two conditions with and without soil water stress corresponded to different meteorological variables, when considering land-atmosphere interaction (Crago and Crowley, 2005; Kahler, and Brutsaert, 2006; Aminzadeh et al., 2016; Maes et al., 2019). For example, air temperature under the unstressed condition would like to be lower than that under the stressed condition, because of the lower sensible heating and the stronger evaporative cooling from the wetter land surface to atmosphere (Maes and Steppe, 2012; Maes et al., 2019). Thus, the mismatch between the assumption of no soil water stress and the observed meteorological variables would like to introduce biases into our PET estimates.

**5 Conclusions**

This study developed a global 5 km monthly PET dataset during 1982–2015 using the calibrated SW model. The model has been well-calibrated against observations at 96 EC flux sites across four major LULC types: forests, shrubland, cropland, and grassland. We mapped spatio-temporal changes in PET and its components (i.e., PT and PE) across the globe. Our PET product has three major improvements relative to the existing PET datasets: (1) it provides the PET estimates by

clearer physical processes, since we take the spatial differences and temporal changes of land surface properties into consideration, (2) it provides not only PET estimates but also PT and PE, and moreover (3) it can take the impacts of elevated $CO_2$ into PET estimation through introducing the stress function of $CO_2$ on $r_s^c$. This dataset can be used by various scientific disciplines (e.g., agronomy, ecology, climatology, hydrology and so on) and policy makers for different operational applications.

**6 Data availability statement**

The product named SW PET with monthly and 5 km resolutions from 1982 to 2015 is freely available at the National Tibetan Plateau Data Center (https://doi.org/10.11888/Terre.tpdc.300193, Sun et al., 2023).

**Author contributions**

SS and ZB designed the research. ZB, YL and XL collected the EC observations and other input data. JX, YL, GS, WJ, CL, MM, JL, YZ, and HC revised the paper.

**Competing interests**

The authors declared that they have no conflict of interest.

**Acknowledgments**

This work was jointly supported by the National Key Research and Development Program of China (Grant No. 2022YFF0801603), National Natural Science Foundation of China (Grant NOs. 42075189 and 42130609), and Natural Science Foundation of Sichuan Province of China (2022NSFSC0215). We thank the data developers, their managers, and funding agencies, whose work and support were essential to data access. The source code for the model used in this study and input files necessary to reproduce the simulations is available from the authors upon request (sun.s@nuist.edu.cn).

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

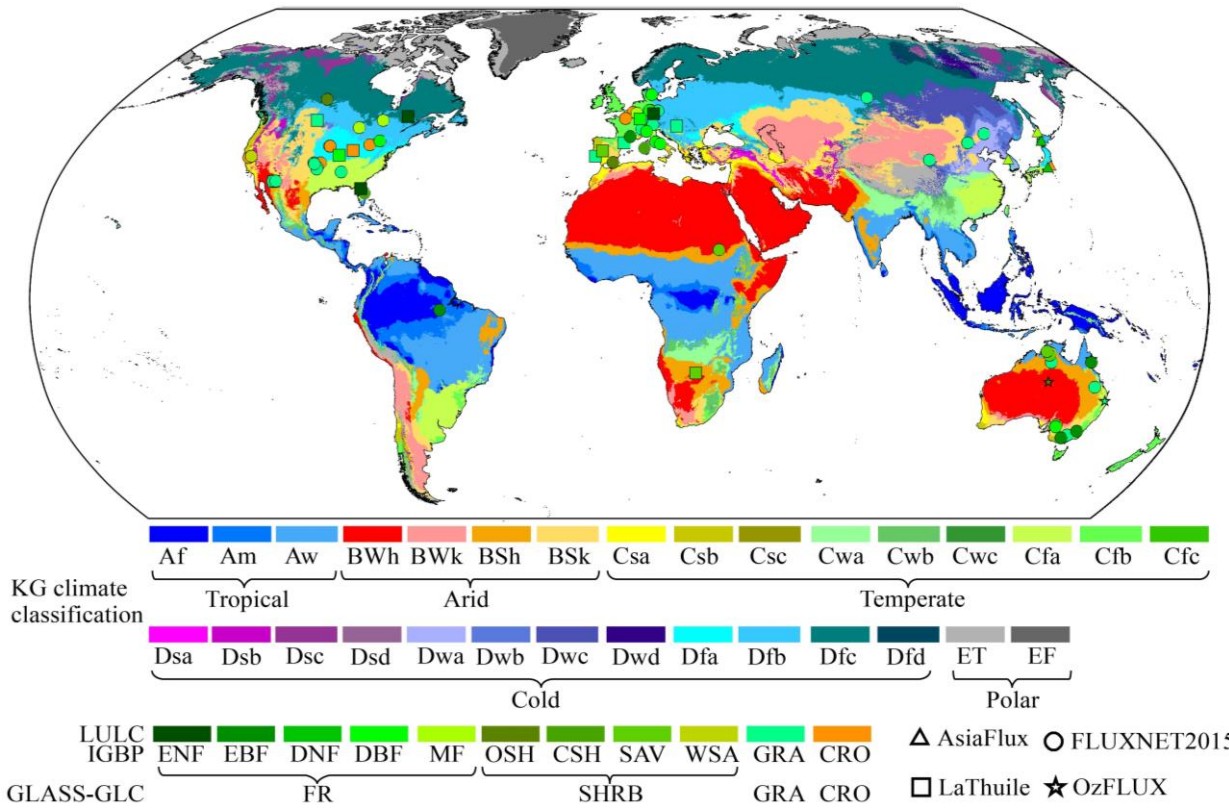

**Figure 1: Locations of the used EC sites in this study over Köppen-Geiger (KG) climate regions (Beck et al., 2018). International Geosphere-Biosphere Programme (IGBP) classification system: CRO—cropland; GRA—grasslands; DBF—deciduous broadleaf forest; EBF—evergreen broadleaf forest; ENF—evergreen needleleaf forest; MF—mixed forest; CSH—closed shrubland; WSA— woody savannah; SAV—savannah; OSH–open shrubland. GLASS-GLC classification system: FR—ENF, EBF, DNF, DBF and MF; SHRB—SAV CSH, OSH and WSA; GRA; CRO.**

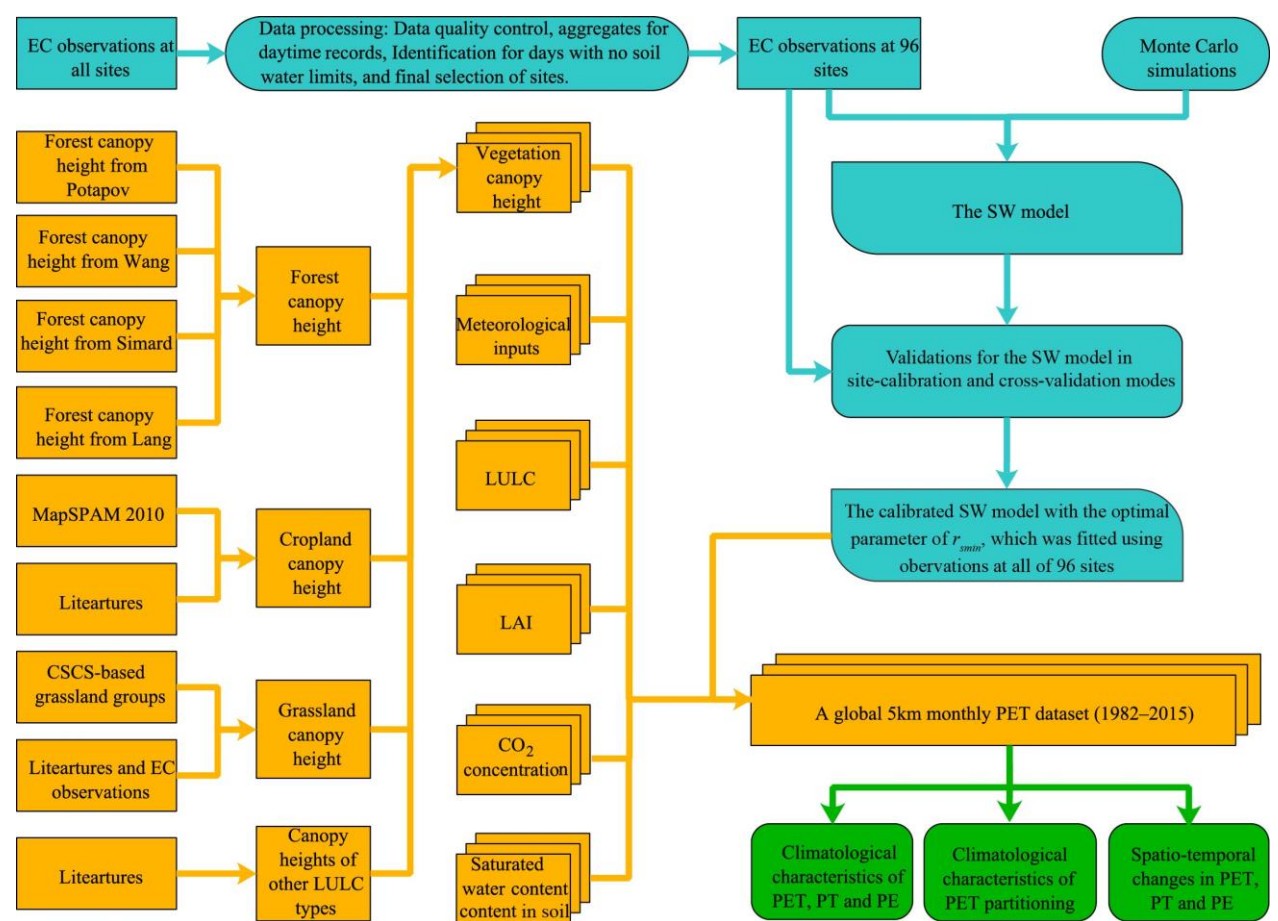

**Figure 2: Workflow of this study. The blue and the yellow-green colors show operating procedures for calibrating and validating the SW model and for producing a global 5 km monthly PET dataset, respectively, while the green color presents the related analyses.**

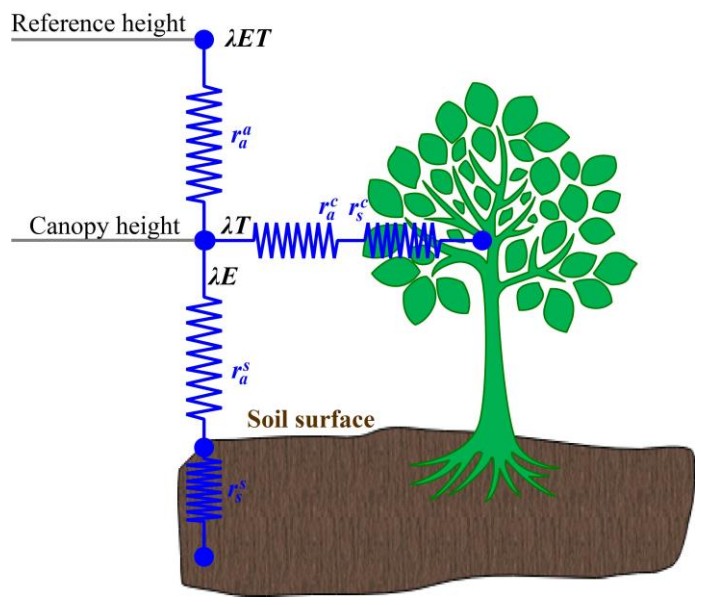

**Figure 3:** Schematic description of the energy partitioning for a canopy with the SW model.

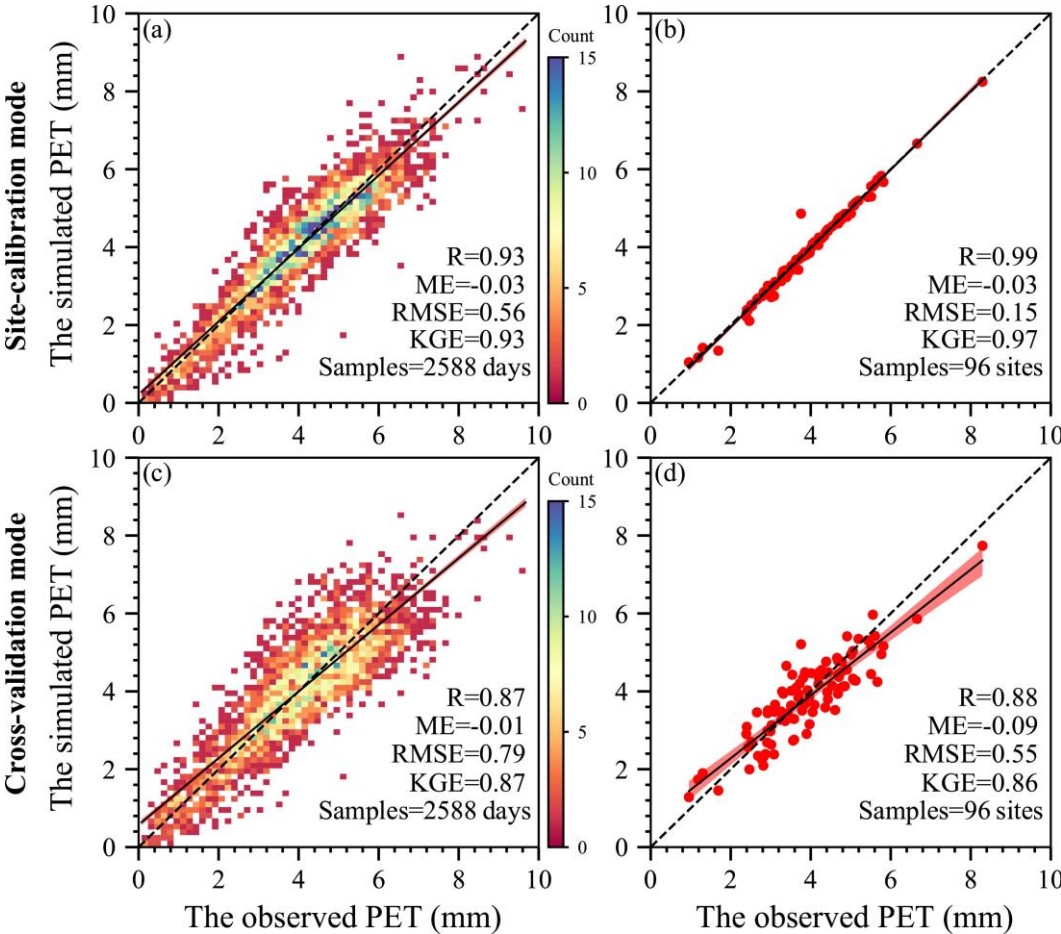

**Figure 4: Scatter-plots of observations against simulations aggregated for all of 96 EC sites. (a): Daily comparison in site-calibration mode. (b): Site mean comparison in site-calibration mode. (c): Daily comparison in cross-validation mode. (d): Site mean comparison in cross-validation mode. In these figures, the dash and the solid lines are the 1:1 line and the regression line with the least square method, while the shadow area represents 95% confidence interval.**

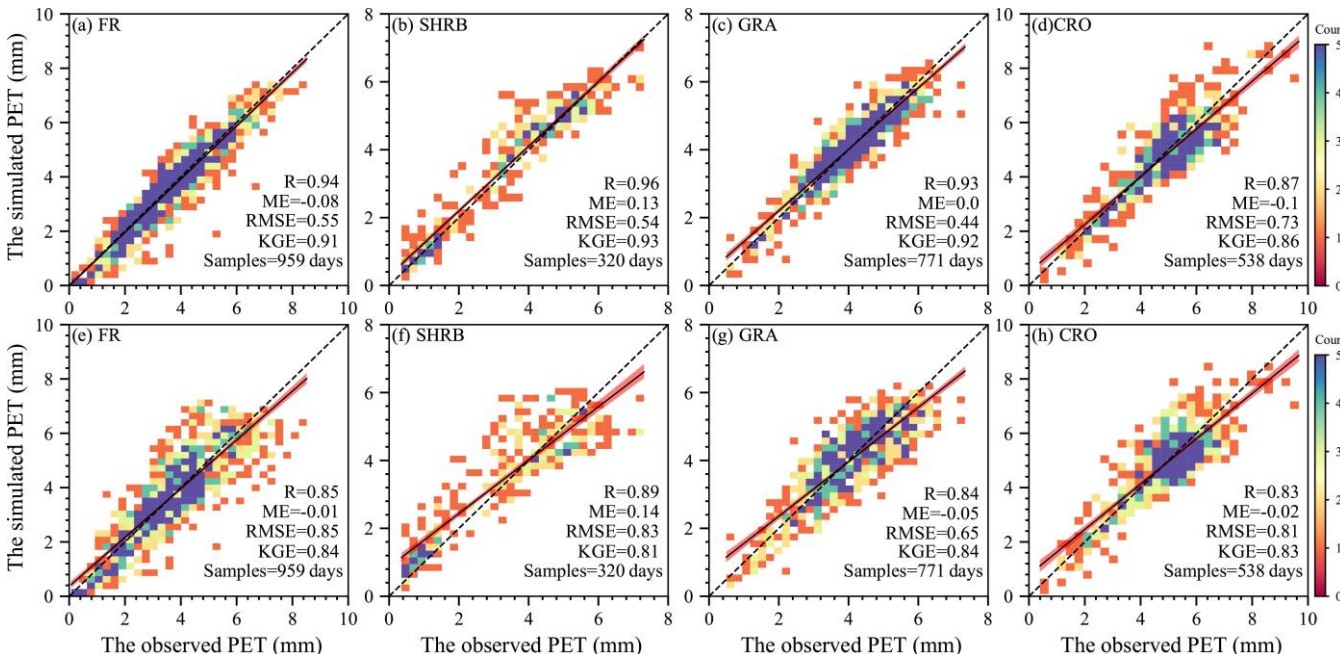

**Figure 5: Scatter-plots of daily observations against simulations aggregated for different LULC types. (a–d): Comparison in site-calibration mode. (e–h): Comparison in cross-validation mode. In these figures, the dash and the solid lines are the 1:1 line and the regression line with the least square method, while the shadow area represents 95% confidence interval.**

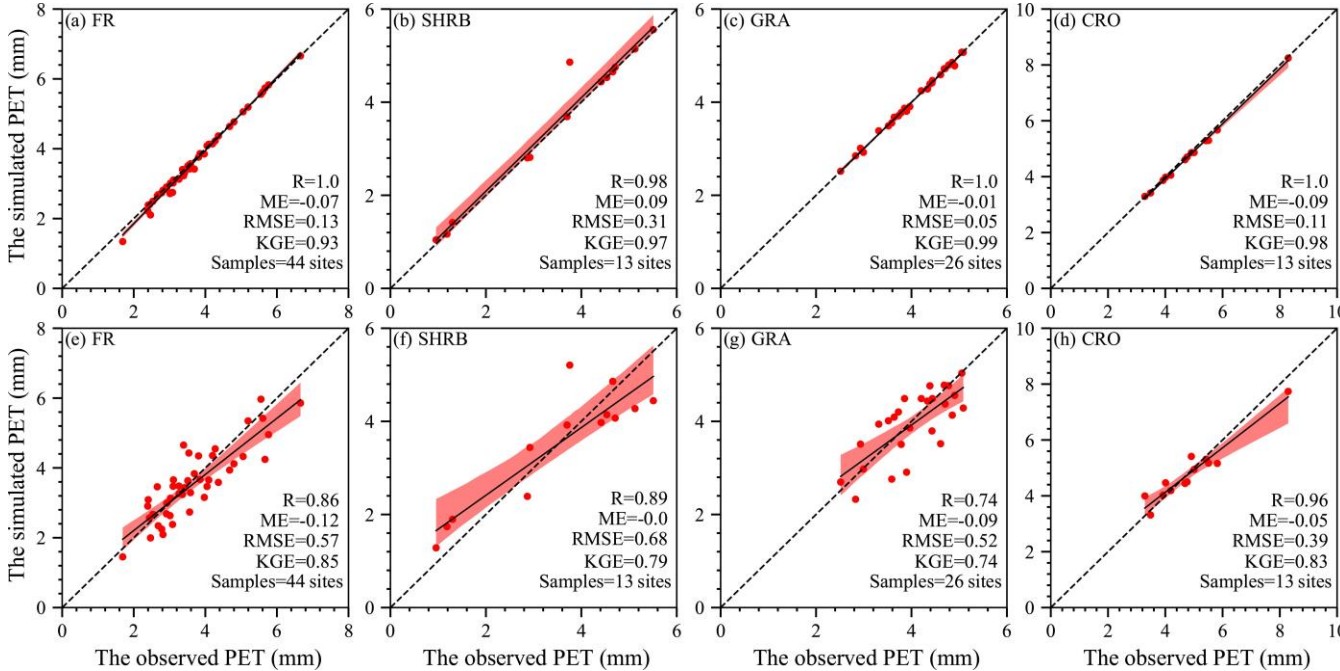

**Figure 6: Scatter-plots of site mean observations against simulations for each LULC type. (a–d): Comparison in site-calibration mode. (e–h): Comparison in cross-validation mode. In these figures, the dash and the solid lines are the 1:1 line and the regression line with the least square method, while the shadow area represents 95% confidence interval.**

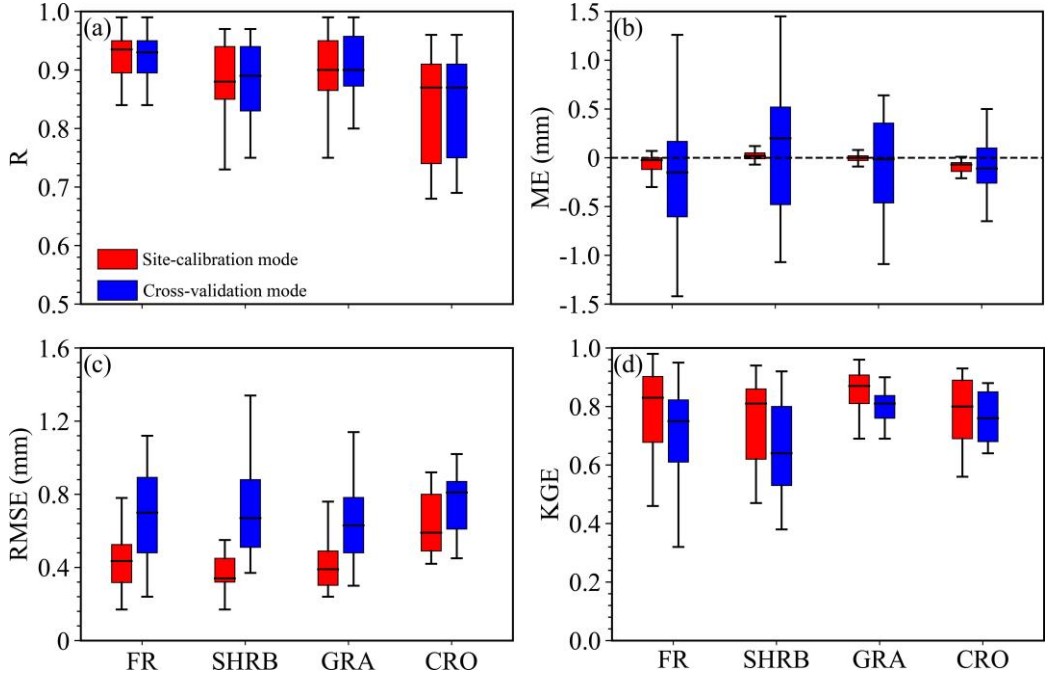

**Figure 7: Boxplots of the validation metrics of daily PET simulations for each LULC type. The whiskers represent the minimum and maximum values of the model performance metrics. The outer edges of the boxes and the horizontal lines within the boxes indicate the 25th, 75th, and 50th percentiles of the validation metrics.**

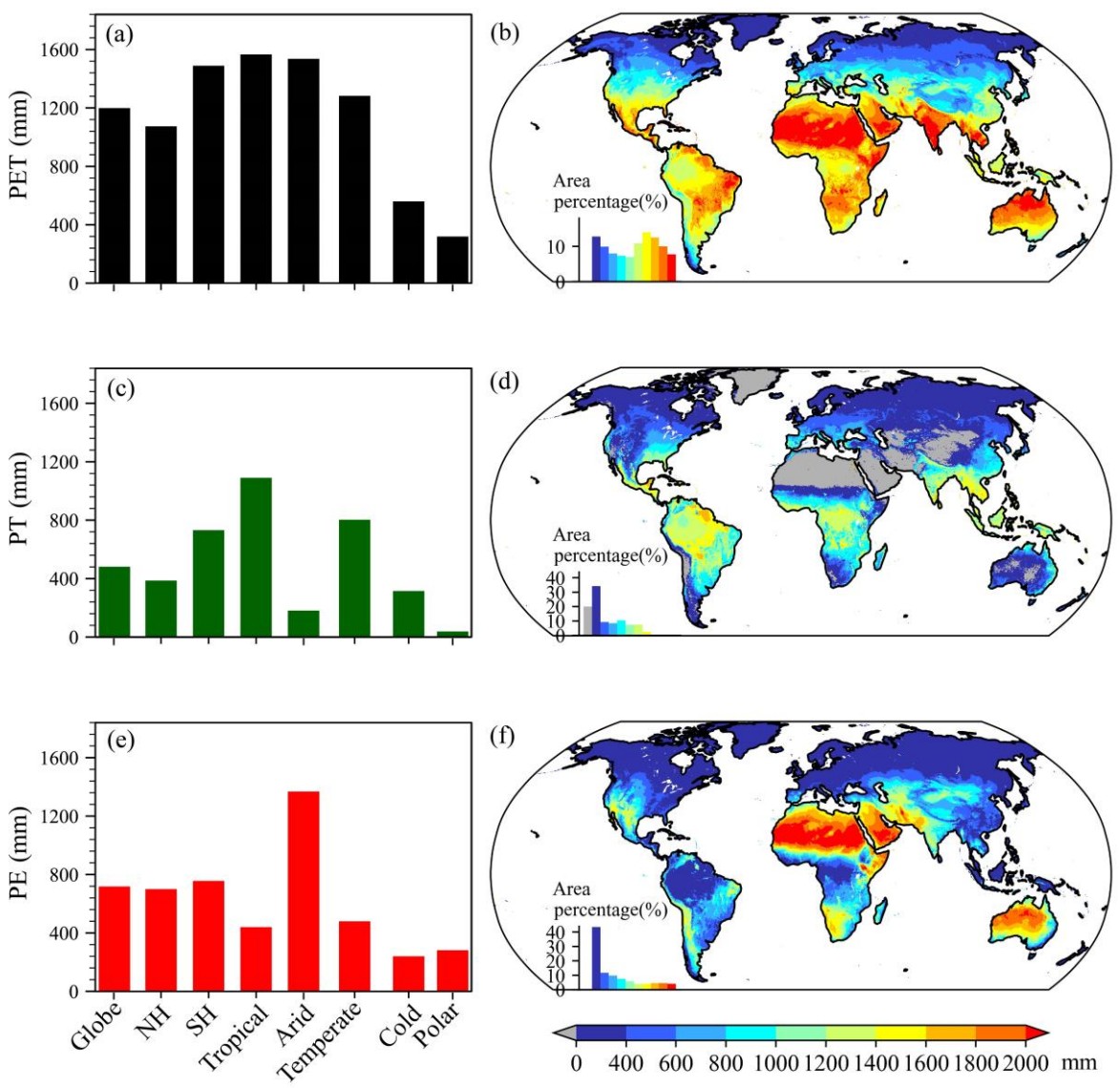

**Figure 8. Climatological annual PET, PE and PT. (a): Climatological annual PET averaged over the globe, Northern Hemisphere (NH) and Southern Hemisphere (SH), and each KG climate region. (b): Spatial distribution of climatological annual PET. (c): Climatological annual PT averaged over the globe, NH and SH, and each KG climate region. (d): Spatial distribution of climatological annual PT. (e): Climatological annual PE averaged over the globe, NH and SH, and each KG climate region. (f): Spatial distribution of climatological annual PE. In (b), (d) and (f), the inset histogram shows area percentage stratified by the amount of annual PET, PT or PE.**

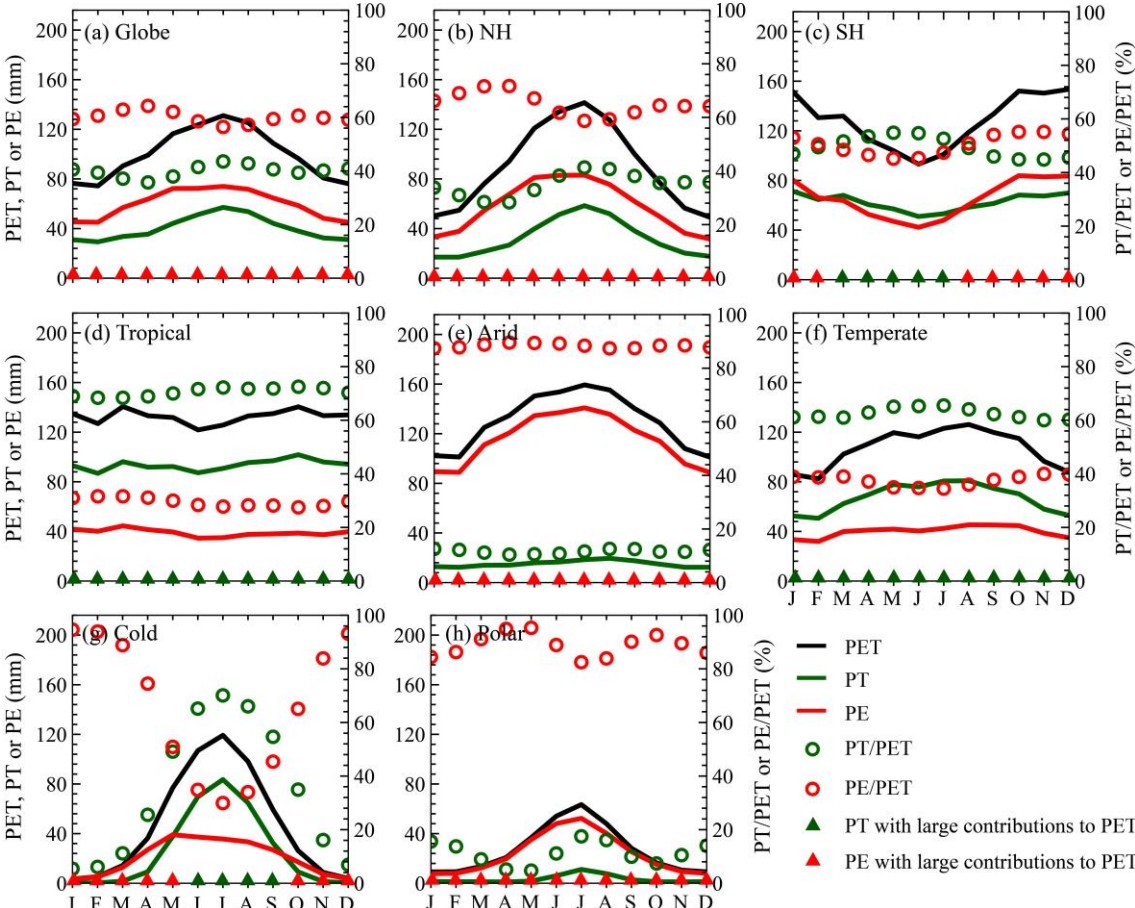

**Figure 9: Climatological monthly PET, PE, PT, PE/PET and PT/PET averaged over the globe, each hemisphere, and each KG climate region.**

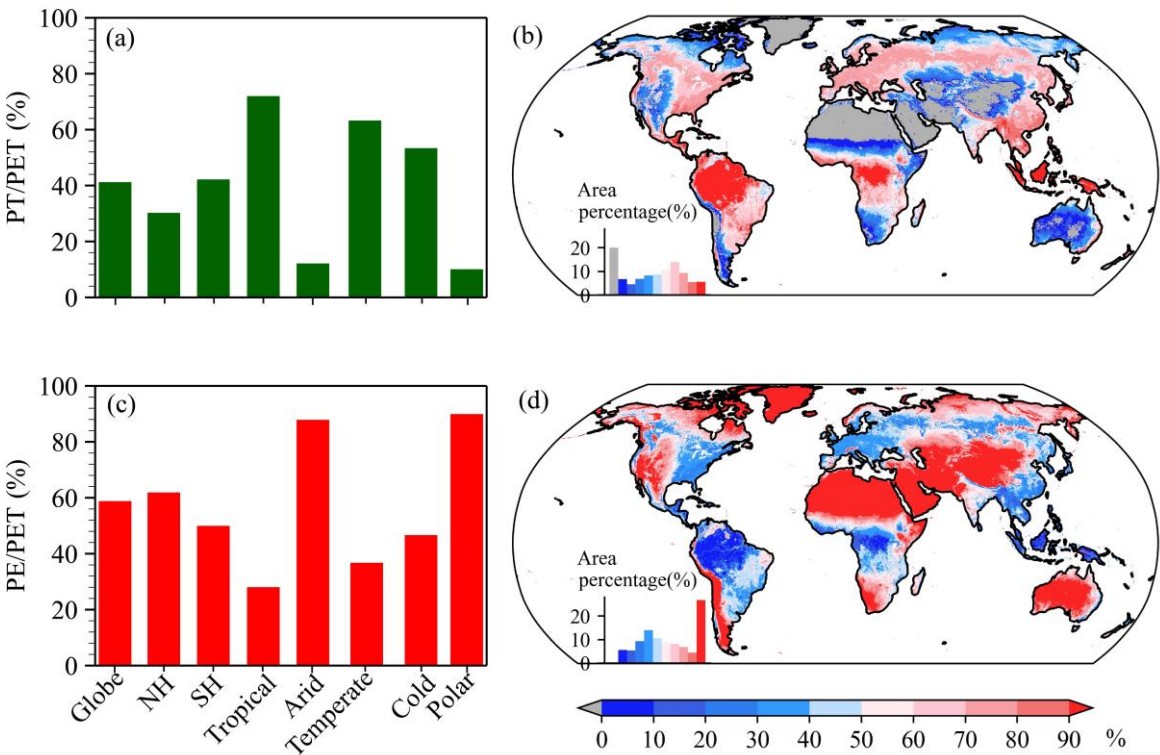

**Figure 10: Characteristics of annual PT/PET and PE/PET. (a): Annual PT/PET averaged over the globe, each hemisphere, and each KG climate region. (b): Spatial distribution of annual PT/PET. (c): Annual PE/PET averaged over the globe, each hemisphere, and each KG climate region. (d): Spatial distribution of annual PE/PET. In (b) and (d), the inset histogram shows area percentage stratified by the amount of annual PT/PET or PE/PET.**

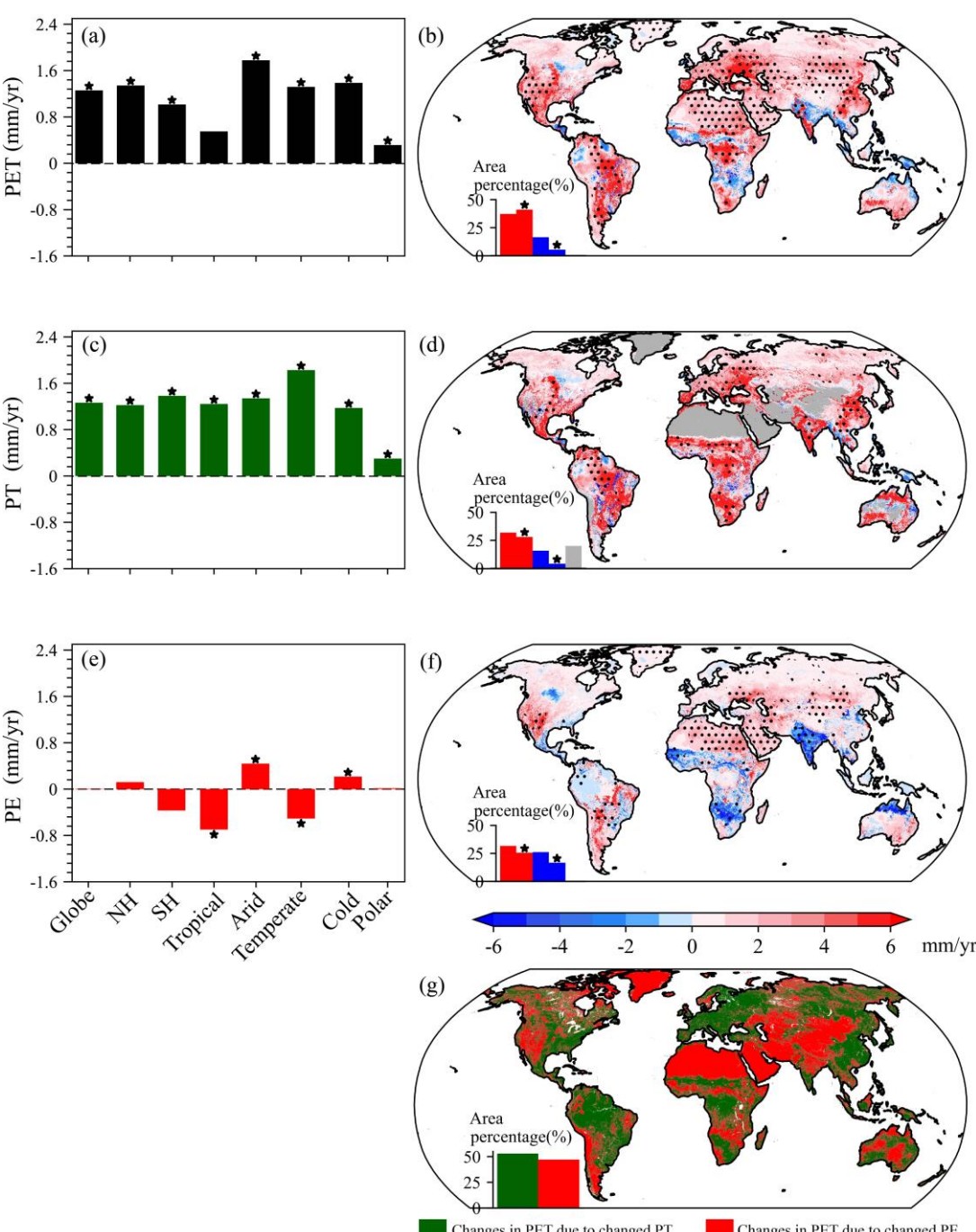

**Figure 11: Characteristics of annual PET, PT and PE trends, and major contributors to annual PET trends. (a): Trends for annual PET averaged over the globe, each hemisphere, and each KG climate region. (b): Spatial distribution of annual PET trends. (c): Trends for annual PT averaged over the globe, each hemisphere, and each KG climate region. (d): Spatial distribution of annual PT trends. (e): Trends for annual PE averaged over the globe, each hemisphere, and each KG climate region. (f): Spatial distribution of annual PE trends. (g): Spatial distribution of major contributors to annual PET trends. In (a), (c) and (e), the star indicates that the trend is significant (*p*<0.05). In (b), (d) and (f), the inset histogram suggests area percentage of insignificant increases (red bar without star), significant increases (red bar with star; *p*<0.05), insignificant decreases (blue bar without star), and significant decreases (blue bar with star; *p*<0.05).**

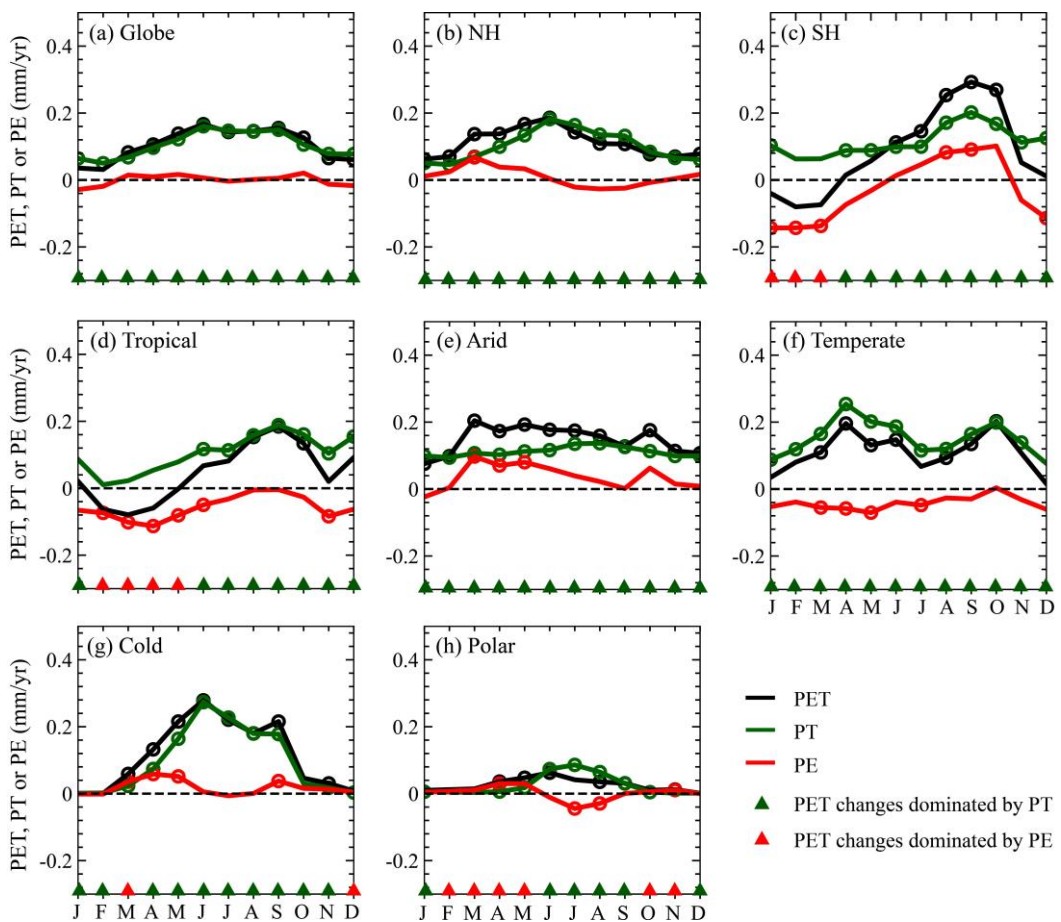

**Figure 12: Changes in monthly PET, PT and PE for the globe, each hemisphere, and each KG climate region. The circle indicate**

**that the trend is significant with *p*<0.05.**

**Table 1: Overview of major inputs for producing the global PET based on the calibrated SW model**

| Datasets | | Basic information | | Sources and references |
|---|---|---|---|---|
| | | Spatio-temporal resolution and time coverage | Variables | |
| Meteorological datasets | MSWX-Past | $0.1° \times 0.1°$; 3-hourly; 1979−present. | Mean, maximum and minimum temperatures, *RH*, *u* at 10 m, downward shortwave radiation and downward longwave radiation. | http://www.gloh2o.org/mswx/; Beck et al. (2022) |
| | CRU TS4.06 | $0.5° \times 0.5°$; monthly; 1901−2021. | Mean, maximum and minimum temperatures, cloud cover and *ea*. | https://crudata.uea.ac.uk/cru/data/hrg/; Harris et al. (2020) |
| | ERA-5 | $0.25° \times 0.25°$; monthly; 1959−present. | Mean, minimum, maximum and dewpoint temperatures, surface pressure, *u* at 10 m, net shortwave radiation and net longwave radiation. | https://cds.climate.copernicus.eu/cdsapp#!/home/; Hersbach et al. (2020) and Berrisford et al. (2021) |
| | MERRA-2 | $0.5° \times 0.625°$; monthly; 1980−present. | Mean, minimuma and maximuma temperatures, specific humidity, surface pressure, *u* at 10 m wind speed, net shortwave radiation and net longwave radiation. | https://disc.gsfc.nasa.gov/; Molod et al. (2015) |
| GLASS AVHRR LAI | | $0.05° \times 0.05°$; 8-day; 1981−2018. | LAI | http://www.glass.umd.edu/; Xiao et al. (2016, 2017) |
| GLASS-GLC | | 5000 m $\times$ 5000 m; yearly; 1982−2015. | LULC | https://doi.org/10.1594/PANGAEA.913496; Liu et al. (2020b) |
| Saturated water content in soil | | $0.0833° \times 0.0833°$; static. | Saturated water content in the first soil layer (i.e., 0–0.0451 m) | http://globalchange.bnu.edu.cn/research/soil5.jsp; Dai et al. (2019a, 2019b) |
| Forest canopy height from Potapov | | 30 m $\times$ 30 m; static. | Forest canopy height | https://glad.umd.edu/dataset/gedi/; Potapov et al. (2020) |
| Forest canopy height from Wang | | 500 m $\times$ 500 m; static. | Forest canopy height | http://www.nsmc.org.cn/NewSite/NSMC/Home/Index.html; Wang et al. (2016) |
| Forest canopy height from Simard | | 1000 m $\times$ 1000 m; static. | Forest canopy height | https://webmap.ornl.gov/wcsdown/dataset.jsp?ds_id=10023; Simard et al. (2011) |
| Forest canopy height from Lang | | $0.5° \times 0.5°$; static. | Forest canopy height | https://doi.org/10.5281/zenodo.5704852; Lang et al. (2021) |
| SPAM V2.0 | | $0.0833° \times 0.0833°$; static. | Cropland distribution map | https://doi.org/10.7910/DVN/PRFF8V; International Food Policy Research Institute (2019) |
| Cropland height | | Static | Height for various cropland | Details in Table S4; Allen et al. (1998) |
| GRA and tundra height | | Static | Typical height for the 5 CSCS-based GRA groups | Details in Table S3 |
| $CO_2$ concentration from Cheng | | $1° \times 1°$; Monthly; 1850−2013 | $CO_2$ concentration | https://doi.org/10.5281/zenodo.5021361; Cheng et al. (2022) |
| GCD $CO_2$ concentration | | $2° \times 2°$; Monthly; 1985−2021 | $CO_2$ concentration | https://www.data.jma.go.jp/ghg/kanshi/co2data/co2_mapdata_e.html; Nakamura et al. (2015) |

Note: Due to unavailable minimum and maximum temperatures for the monthly ERA5 and MERRA-2 datasets, monthly values of the two variables were extracted from their hourly datasets.