# Peer review of "A global 5km monthly potential evapotranspiration dataset (1982–2015) estimated by the Shuttleworth-Wallace model"

_Earth System Science Data, 2023_

## Referee Comment (RC2)

**Review comments**

Sun et al. provide a valuable estimate of global potential evapotranspiration (PET), which is highly beneficial for various research applications. The authors have taken into account various factors that influence PET estimation and have prepared multiple datasets to derive the PET values. The manuscript is well-structured, with clear method descriptions, data processing procedures, and explanations. Overall, the manuscript could be accepted with some comments to further enhance its quality.

**Major comments**

1. What are the valid temporal resolutions (e.g., hourly, monthly) for the inputs used in different models (i.e., equations)

   Different meteorological datasets have varying temporal resolutions, such as 3-hourly for MSWX and monthly for others. When equations are used to calculate PET and related variables (e.g., D, Rn) to derive PET, it is important to consider whether these equations, as presented in the main text and supplementary materials, are valid for different temporal resolution inputs (e.g., 3-hourly vs. monthly). For example, can the SW equation be applied to different temporal resolution inputs (e.g., hourly or mothly)? It would be helpful to provide information on the validity of the equations for different temporal resolutions of inputs, whenever applicable.

   Additionally, the SW model was calibrated based on daily inputs (as shown in Figure 3). However, when applying the SW model globally, monthly inputs were used. The question arises whether it is appropriate to use a daily calibrated model for monthly inputs application.

2. It would be highly valuable if the authors could provide the datasets that were used to derive PET. This would include the following:

   - EC related datasets, e.g., the original datasets after quality control, selected datasets with no soil water limits, etc.

- Finally processed canopy height, and/or its source datasets.

- Land use/land cover, LAI, saturated water content in soil, and the $CO_2$ concentration

Question for the $CO_2$ concentration, the seasonal cycle of $CO_2$ is different among different locations, e.g., between south and north hemisphere, will this affect your PET estimation?

3. About SW model:

   1) How about you add a concept diagram to show the structure of SW and related equation variables. One example for your reference, Figure 5 in

      o Kochendorfer, J. P. and Ramírez, J. A.: Modeling the monthly mean soil-water balance with a statistical-dynamical ecohydrology model as coupled to a two-component canopy model, Hydrol. Earth Syst. Sci., 14, 2099–2120, https://doi.org/10.5194/hess-14-2099-2010, 2010.

[Figure]

   2) EQ 1a, should the Cc be removed. Why not the total latent heat doesn't equal the sum of canopy and vegetation latent heat fluxes, but need multiply the coefficients, could you add some explanation?

   $\lambda ET = C_cPMc + C_sPMs$ ?

   3) It would be helpful if the authors could provide an explanation for the calculation of LAIe in EQ2b and clarify the reasoning behind this approach. What is the underlying assumption or basis for this equation?

- When you calibrate the SW model using EC data with filtering out the rain effects, so most of the LAI should be likely effective (no rain coverage over the leaves), why the LAIe is still calculated in EQ2b (i.e., LAIe is less than LAI when LAI>2)?

- When you apply the calibrated SW model for global, the effective LAI should be considered due the reasons of rain. The LAIe should be related to different conditions (e.g., different rainfall intensity) but not considered in the EQ2b

- It seems there is inconsistence. When calibrating SW model using no-rain effects data, but applying effective LAI (i.e., from EQ2) when the rain effects is small. But the same LAIe equation are used for the global application, when under some conditions rain effects may be large. The PET calculation should also include the maximum ET during rain or after rain events, right?

- I wonder how to consider the LAIe for PET calculation at EC site level and global grid. How is the SW PET sensitivity to LAIe?

- Here I only say rain effects on LAIe, but other factors may also effect LAIe calculations, e.g., snow.

4. It would be beneficial to have a clear explanation of how PT and PE are calculated differently. Currently, there are no specific equations provided to illustrate the calculations for PE and PT. It appears that PT is derived from PMc, while PE is also derived from PMc. It is important to clarify this distinction and explicitly mention that PT and PE are derived from PMc in the text. Additionally, please review the sentence in line 238 that states "while PMc and PMs are the soil and vegetation latent heat fluxes (W/m2)" to ensure the correct explanation of parameters and variables throughout the equations.

Furthermore, it is worth considering the inclusion of discussions on the explicit consideration of plant hydraulics in recent land surface models (e.g., CLM5 in 2019, NOAH-MP in 2021, CoLM in 2022) as it relates to transpiration simulations. I am interested to know whether the SW model implicitly incorporates plant hydraulics or if there are potential improvements that could be made to the PET estimation by integrating

plant hydraulics within the framework of the SW model. Section 4.2 would be an appropriate place to include such discussions.

Related references:

- o https://agupubs.onlinelibrary.wiley.com/doi/10.1029/2018MS001500

- o https://agupubs.onlinelibrary.wiley.com/doi/full/10.1029/2020MS002214

5. You have presented the trends of PET, PE, and PT, as well as the contributions of changes in PE and PT to changes in PET. I am curious to know which factors, such as changes in meteorological forcings, contribute to the observed changes in PET, PE, or PT. For instance, could the global temperature increase be a significant driver?

Furthermore, it would be valuable to include a discussion on how the phenomenon of Earth greening, such as an increase in LAI, may influence your trend analysis. Consider commenting on the potential impacts of Earth greening on the observed trends in PET, PE, and PT.

**Minor comments**

1. In the introduction, the authors mentioned different types of models to calculate the PET, and give examples for each type model (e.g., Penman-Monteith). It would be great if the authors can also provide the equations for these example models in the supplementary, so the readers can better compare them. Also please provide the information about the common temporal resolutions of the inputs for these models, hourly or daily or, …

2. In the supplementary, where is "EQ. S7a", should be EQ S4a ?. Bold the titles of "The ERA-5 D" and "The MERRA-2 D"

3. Lang et al. has another dataset from the webpage: https://langnico.github.io/globalcanopyheight/, what is the difference between this version of data and the Lang data you used in your study. Should this new data better than what the Lang data you used. You may add some discussion of this.

It seems that the canopy height is temporally static, but the LULC changes yearly. How to make the consistence for each year's LULC's canopy height for a given grid if LULC changes happens.

4. L230, $r_{smin}$ should be defined when it first appears.

5. How are the averages of PET, PE, PT are calculated based on grid average or area average? Please mention it in the text.

6.  Check the Figure 8, the colors for scatters and PE PT lines are not consistent.

7. For the calibration of r_smin, it would be helpful to know the range of variation among the 10 r_smin values for each specific plant functional type (PFT) site.

---

## Author Response (AR1)

**RESPONSES TO COMMENTS FROM EDITORS AND REVIEWERS**

Dear Editor and Reviewers,

Thanks very much for your careful consideration of our manuscript entitled "A global 5km monthly potential evapotranspiration dataset (1982–2015) estimated by the Shuttleworth-Wallace model" (No.: ESSD-2023-38) and for providing us valuable comments and suggestions.

We have seriously reviewed all the comments and suggestions, and made the necessary revisions to improve the quality and clarity of our paper. We hope that the revised manuscript meets the requirements for publication in Earth System Science Data.

Enclosed with this letter is a detailed response to the comments and suggestions, along with our corresponding revisions in this manuscript. Thank you once again for your time and attention to our work.

Thank you and best regards.

Yours sincerely,

PhD. Shanlei Sun

Collaborative Innovation Center on Forecast and Evaluation of Meteorological Disasters/Key Laboratory of Meteorological Disaster, Ministry of Education/International Joint Research Laboratory on Climate and Environment Change, Nanjing University of Information Science and Technology, Nanjing, China

E-mail address: sun.s@nuist.edu.cn

**Reviewer#1**

**This manuscript generated a global 5km monthly potential evapotranspiration dataset (1982–2015) estimated by comprehensively considering interspecific differences in various vegetation-related parameters (e.g., plant stomatal resistance and CO₂ effects on stomatal resistance) to calibrate and parametrize the Shuttleworth-Wallace (SW) model. It can be accepted after some revisions. The authors should carefully consider my comments and suggestions below.**

**Responses:** We thank this reviewer very much for the positive comments and the valuable suggestions, which are believed to be very useful for us to improving the study. Seriously according to these suggestions, we have revised this manuscript, and the detailed information could be found below and the revised version.

**Comment 1: Lines 72. Maybe it is necessary to explain more clearly what is vegetation not "closed?**

**Response:** Thanks for your suggestion. The concept of "closed" vegetation emphasizes the important role that vegetation plays in the vegetation-atmosphere interaction (Shuttleworth and Wallace, 1985). Soil evaporation is often overlooked due to its small proportion, such as the PM model treating crop canopy as a single uniform cover (the "big leaf" model), but neglecting soil surface evaporation (Zhou et al, 2006). The "closed" vegetation is mostly distributed in areas with abundant vegetation cover. In areas with sparse or no vegetation, soil evaporation is significant and cannot be ignored, so it's not a "closed" vegetation. Therefore, the relevant explanation has been added in the revised manuscript, i.e., "*open canopy that light can penetrate to the ground*". **(L73)**

*References:*

*Shuttleworth, W. J. and Wallace, J. S.: Evaporation from sparse crops—an energy combination theory. Quarterly Journal of the Royal Meteorological Society, 111, 839–855, 1985.*

*Zhou, M. C., Ishidaira, H., Hapuarachchi, H. P., Magome, J., Kiem, A. S. and Takeuchi, K.: Estimating potential evapotranspiration using Shuttleworth-Wallace model and NOAA-AVHRR NDVI data to feed a distributed hydrological model over the Mekong River basin. Journal of Hydrology, 327(1–2), 151–173, 2006.*

**Comment 2: Lines 74. "Heterogeneous" should be "homogeneous"?**

**Response:** Thanks for pointing out the mistake. We have modified the word of "heterogeneous" as "homogeneous".

**Comment 3: Line 97-98. "vegetation height" and "Leaf Area Index" should be "vegetation height datasets" and "Leaf Area Index datasets", respectively?**

**Response:** We have modified the dataset names of "vegetation height" and "Leaf Area Index" as "*vegetation height datasets*" and "*Leaf Area Index datasets*", respectively.

**Comment 4: Figure 1. I suggest enlarging the size of the font in the legend and adjusting the position of the legend.**

**Response:** Thanks. We have revised Figure 1 in this revision, and please see below (Figure R1) or the revised manuscript.

[Figure]

*Figure R1: Locations of the used EC sites in this study over Köppen-Geiger (KG) climate regions (Beck et al., 2018). International Geosphere-Biosphere Programme (IGBP) classification system: CRO—cropland; GRA—grasslands; DBF—deciduous broadleaf forest; EBF—evergreen broadleaf forest; ENF—evergreen needleleaf forest; MF—mixed forest; CSH—closed shrubland; WSA—woody savannah; SAV—savannah; OSH–open shrubland. GLASS-GLC classification system: FR—ENF, EBF, DNF, DBF and MF; SHRB—SAV CSH, OSH and WSA; GRA; CRO.*

**Comment 5: Line 144-145. How did the authors consider the uncertainty from gap-filled heat fluxes? As far as I know, the MDS method was applied to fill the gap of ET measurements in the FLUXNET2015 datasets. Recently, Jiang et al. (2022) have developed a novel physical full-factorial scheme for gap-filling of EC-measured ET. Their scheme has been demonstrated to outperform the MDS and MDV gap-filling methods.**

**The authors should introduce this new advance in the gap-filling of ET and at least discuss the possible uncertainty caused by using the MDS gap-filled ET.**

**Response:** We thank this reviewer for the valuable suggestions. We have checked the EC data with no water stress, and found that the gap-filling data points only accounted for less than 30% of the EC data used in this study. To answer this question, we only remained the data points without gap-filling to recalibrate and revalidate the established SW model (Figures R2 and R3). Overall, the new final $r_{smin}$ values have no evident changes relative to our original values (Figure R2). In addition, seen from the validation results, we also found that the new results became worse but much limited compared to our original results (Figure R2). These suggested that the gap-filling data points may have limited impacts on our results.

Notably, we have contacted the authors of the novel physical full-factorial scheme, but it is unfortunate that we have not obtained the corresponding data for comparison before the deadline of this response. Moreover, it may be difficult for us to fill the data gaps using this novel physical full-factorial scheme, mainly because the $z_{0m}$ (which is a critical parameter for conduct the physical full-factorial scheme of Jiang et al. (2022)) was needed for this scheme but was also unavailable. Anyway, the gap-filling data may introduce more or less into our results, and therefore the related discussions have been added to the revised manuscript, such as "*In this study, for maximizing the use of data, the marginal distribution sampling method was employed to fill the gaps in the EC LE measurements. However, it should be noted that even though the controlling thermodynamic and kinetic factors of the atmosphere are similar between the missing and retrieved moments (Jiang et al., 2022), the gap-filled LE may be of low confidence, especially when soil moisture has abrupt changes. To quantify potential impacts of the gap-filled values, the SW model was re-calibrated and re-validated against the data points without gap-filling. Relative to the SW model used in this study, the new $r_{smin}$ and the validation metrics changed insignificantly (not shown here), suggesting that the uncertainties induced by the gap-filled LE were limited.*" **(L564-570)**

[Figure]

Figure R2: The final $r_{smin}$ values for FR, SHRB, GRA and CRO.

[Figure]

Figure R3: Validation results for the SW model calibrated using the data points without gap-filling. (a-d) Scatter-plots of daily observations against simulations for each LULC type. (e-h) Scatter-plots of site mean observations against simulations for each LULC type. (i) Scatter-plot of daily observations against simulations for all of 96 EC sites. (j) Scatter-plot of site mean observations against simulations for all of 96 EC sites.

*References:*

*Jiang, Y., Tang, R. and Li, Z. L.: A physical full-factorial scheme for gap-filling of eddy covariance measurements of daytime evapotranspiration. Agricultural and Forest Meteorology, 323, 109087, 2022.*

**Comment 6: Line 149-154. How did the authors consider the effect of nighttime ET on daily ET? The authors seemed to take the daytime ET as a surrogate of daily ET.**

**Response:** Thank you very much for this comment. Yes, this study did not consider the nighttime ET ($ET_n$). Our considerations are two-fold. First, we have to admitted that $ET_n$ does exist. For example, Padrón et al. (2020) showed that the proportion of $ET_n/ET$ is approximately 6.3% based on the FLUXNET2015 dataset, and 7.9% based on the multiple global model simulations. Relatively, the proportion is small and may exert limited impacts on our PET estimates. Second, due to impacts of many biological and abiotic factors (Han et al., 2021), the $ET_n$ process is very complex, and the related controlling mechanisms are still not clear to date. For example, Novick et

al (2009) and Groh et al (2019) found that vapour pressure deficit (VPD) and wind speed have a significant impact on $ET_n$, while Groh et al. (2019) state that the contributions of night dew to $ET_n$ cannot be ignored. As an important component of $ET_n$, the nighttime transpiration ($Tr_n$) is not only related to the incomplete stomatal closure (Dawson et al., 2007; Duursma et al., 2019) but also the circular regulation of nighttime water uses by plants (De Dios et al, 2015). To date, the factors influencing $Tr_n$ are still disputed. Dawson et al. (2007) and Moore et al. (2008) found a positive correlation between $Tr_n$ and VPD and soil moisture content (SWC), while Barbour and Buckley (2007), Phillips et al. (2010) and De Dios et al. (2015) pointed out that no or negative correlation existed between $Tr_n$ and the two variables aforementioned. Meanwhile, the biological factors (e.g., plant species and ecosystem types) can also significantly influence $Tr_n$ (O'keefe and Nippert, 2018; Zeppel et al., 2014). Just due to the complex mechanisms, to establish a common model for estimating $ET_n$ across various ecosystems is still a challenging to date, and may lie beyond the scope of our study.

Anyway, neglecting $ET_n$ process may result in the estimated PET to be smaller than the truth values, and therefore, we have added the related discussions in the revised manuscript, such as "*Padrón et al. (2020) showed that on average the fraction of nighttime ET accounts for approximately 6.3% of the total ET informed by the FLUXNET2015 dataset while 7.9% based on multiple global models. Notably, this fraction may exceed to 15% in mountain forest with snowy and windy winter. Ignoring vegetation canopy interception and nighttime ET may underestimate PET (Tourula and Heikinheimo, 1998; Lawrence et al., 2007; Mu et al., 2011; Padrón et al., 2020; Singer et al.; 2021). Subsequent research will be done to integrate these two processes in the SW model to further enhance the model's physical mechanism.*" **(L540-545)**

*References:*

*Barbour, M. M., and Buckley, T. N.: The stomatal response to evaporative demand persists at night in Ricinus communis plants with high nocturnal conductance. Plant, Cell and Environment, 30(6), 711–721, 2007.*

*Dawson, T. E., Burgess, S. S., Tu, K. P., Oliveira, R. S., Santiago, L. S., Fisher, J. B., Simonin, K. A. and Ambrose A. R.: Nighttime transpiration in woody plants from contrasting ecosystems. Tree Physiology, 27, 561–575, 2007.*

*De Dios, V. R., Roy, J., Ferrio, J. P., Alday, J. G., Landais, D., Milcu, A. and Gessler, A.: Processes driving nocturnal transpiration and implications for estimating land evapotranspiration. Scientific Reports, 5(1), 10975, https://doi.org/10.1038/srep10975, 2015.*

*Duursma, R. A., Blackman, C. J., Lopéz, R., Martin-StPaul, K., Cochard, H. and Medlyn, B. E.: On the minimum leaf conductance: Its role in models of plant water use, and ecological and environmental controls. New Phytologist, 221(2), 693–*

705, 2019.

Groh, J., Pütz, T., Gerke, H., Vanderborght, J. and Vereecken, H.: Quantification and prediction of nighttime evapotranspiration for two distinct grassland ecosystems. Water Resources Research, 55(4), 2961–2975, 2019.

Han, Q, Wang, T., Wang, L., Smettem, K., Mai, M. and Chen X.: Comparison of nighttime with daytime evapotranspiration responses to environmental controls across temporal scales along a climate gradient. Water Resources Research, 57(7). https://doi.org/10.1029/2021WR029638, 2021.

Moore, G. W., Cleverly, J. and Owens, M. K.: Nocturnal transpiration in riparian Tamarix thickets authenticated by sap flux, eddy covariance and leaf gas exchange measurements. Tree Physiology, 28(4), 521–528, 2008.

Novick, K. A., Oren, R., Stoy, P. C., Siqueira, M. and Katul, G. G.: Nocturnal evapotranspiration in eddy-covariance records from three co-located ecosystems in the Southeastern US: Implications for annual fluxes. Agricultural and Forest Meteorology, 149(9), 1491–1504, 2009.

O'keefe, K. and Nippert, J. B.: Drivers of nocturnal water flux in a tallgrass prairie. Functional Ecology, 32(5), 1155–1167, 2018.

Padrón R. S., Gudmundsson, L., Michel, D. and Seneviratne, S. I.: Terrestrial water loss at night: Global relevance from observations and climate models. Hydrology and Earth System Sciences, 24(2), 793–807, 2020.

Phillips, N. G., Lewis, J. D., Logan, B. A. and Tissue, D. T.: Inter- and intra-specific variation in nocturnal water transport in Eucalyptus. Tree Physiology, 30(5), 586–596, 2010.

Zeppel, M. J. B., Lewis, J. D., Phillips, N. G. and Tissue, D. T.: Consequences of nocturnal water loss: A synthesis of regulating factors and implications for capacitance, embolism and use in models. Tree Physiology, 34(10), 1047–1055, 2014.

**Comment 7: Line 163-164. In arid areas, a day may not be identified to have no water limits even when its corresponding EF exceeded the 95th percentile EF threshold.**

**Response:** Thanks for your comment. Based on Köppen-Geiger (KG) climate regions, we selected 8 sites in arid areas, including one FR (i.e., AU-Lox), four GRA (i.e., AU-Emr, AU-Stp, US-SRG, and US-Wkg) and three SHRB (ES-Amo, AU-TTE, and SD-Dem) sites. For knowing impacts of the EF threshold on the SW model, we re-calibrated $r_{smin}$ at each of the selected 8 sites with the 96th and 98th percentile EF thresholds, and then compared these new $r_{smin}$ values with our used values in the manuscript (Figure R4). Overall, the $r_{smin}$ values were around the 1:1 line with a regression coefficient above 0.96, implying slight changes $r_{smin}$ when the threshold changed from 95th to 96th or 98$^{th}$. Figure R5 shows the changes (i.e., $\Delta x$ = 95th EF – 96th (98th) EF); $x$ represents the validation metric) in validation metrics of the SW model calibrated with the 96th and 98th percentile EF thresholds

relative to the original values. For most of the selected 8 sites, the $\Delta R$, $\Delta ME$, $\Delta ubRMSE$ and $\Delta KGE$ values are between -0.1 and 0.1, between -0.02 mm/day and 0.04 mm/day, between -0.05 mm/day and 0.1 mm/day, and between -0.05 and 0.05. This suggested that the changes in the EF threshold may have limited impacts on the performance of the SW model at site scale. Additionally, using the 98th percentile EF threshold for theses 8 sites and the 95th percentile EF threshold for other 88 sites, we re-estimated the new optimal $r_{smin}$ for each LULC and show the scatter-plots in Figure R6. Comparing Figures 4 and 5 in the manuscript vs. Figure R6, the validation metrics for each LULC have no evident changes when the EF threshold changed from 95th to 98th.

To sum up, we think that the 95th EF threshold could be used to define the condition with no water limits in arid areas, based on limited impacts of the EF threshold changes on the $r_{smin}$ values and performance of the SW model. However, we have to admit that the possible cases may existed in arid areas, i.e., a day may not be identified to have no water limits even when its corresponding EF exceeded the 95th percentile EF threshold. Therefore, for more serious, the related discussion has been added in the revised manuscript, such as "*Second, due to usual water deficits in arid regions, the EF threshold may exceed to the 95th percentile of EF. Thus, the identified unstressed days based on this criterion may actually include the stressed days. This potentially biased the PET estimates in arid regions. Through increasing the EF threshold (e.g., 96th and 98th; not shown here) for several EC sites in arid regions, we found that selecting 95th percentile as threshold has limited impact on the PET estimates.*" **(L555-564)**

[Figure]

Figure R4: Scatter-plots of the recalibrated $r_{smin}$ against with those used in the manuscript for the selected 8 EC sites in the KG arid areas

[Figure]

Figure R5: Changes in the validation metrics of the SW model calibrated with the 96th and 98th percentile EF thresholds for 8

EC sites in the KG arid areas, relative to those of the SW model used in the manuscript.

[Figure]

Figure R6: Validation results for the SW model calibrated with the 98th percentile EF threshold for the 8 sites in arid areas and

the 95th percentile EF threshold for other 88 sites. (a-d) Scatter-plots of daily observations against simulations for each LULC

type. (e-h) Scatter-plots of site mean observations against simulations for each LULC type. (i) Scatter-plot of daily observations

against simulations for all of 96 EC sites. (j) Scatter-plot of site mean observations against simulations for all of 96 EC sites.

**Comment 8: Line 170. The CO₂ concentration observed at Mauna Loa cannot represent the spatial**

**heterogeneity of CO₂ concentration over the globe. How did the authors consider the uncertainty from this**

**act?**

**Response:** Thanks for your comments. For considering the spatial differences in $CO_2$, we have recalibrated the

SW model and reproduced PET, PT and PE with the gridded $CO_2$ dataset (i.e., the monthly $CO_2$ concentration with

a spatial resolution of 1◦ × 1◦ and a time span of 1850–2013 from https://doi.org/10.5281/zenodo.5021361 (Cheng

et al., 2022), and the monthly Global $CO_2$ Distribution product from Japan Meteorological Agency with a spatial resolution of 2∘ × 2∘ and a time span of 1985–2021). The detailed information could be found in the revision.

**Comment 9: Lines 178. I suggest specifying the full name of CRUTS like the previous dataset. Please keep CRUTS consistent with the CRU TS below.**

**Response:** Thanks for your suggestion. The corrections have been done, and please see the revision.

**Comment 10: Figure 2. I suggest adjusting the color configuration of the flow chart, such as reducing the color saturation.**

**Response:** Thanks. We have revised this figure according to the suggestion, and please see below or the revised manuscript.

[Figure]

Figure 2: Workflow of this study. The blue and the yellow-green colors show operating procedures for calibrating and validating the SW model and for producing a global 5 km monthly PET dataset, respectively, while the green color presents the related analyses.

**Comment 11: Lines 265. W/m2 should be corrected to W/m².**

**Response:** Thanks. This mistake has been corrected in this revision.

**Comment 12: Section 2.1.4. I suppose that the authors use the vegetation canopy height to estimate surface roughness length and further aerodynamic resistance. I have two questions. First, the authors have used multiple datasets as inputs or to obtain vegetation canopy height (h). How did the authors consider the difference between different datasets? Second, how to parameterize the spatially and temporally variable surface roughness length by using the time-invariant vegetation height? The authors are suggested to refer to the work of Peng et al. (2022) who developed for the first time a practical method for global estimates of 500 m daily aerodynamic roughness length with a combination of machine learning techniques, wind profile equation, observations from 273 sites and MODIS remote sensing data.**

**Response:** Thanks for the valuable suggestions. Although the vegetation height is a critical parameter for establishing and running the SW model, the existing several forest height datasets could not fully cover the globe. Therefore, a vegetation height dataset with a global coverage was reconstructed in this study based on the several popular forest height datasets, the nearest neighbor interpolation method, and heights of the typical croplands and the typical grasslands. Additionally, to reduce uncertainties of forest height datasets (which were related to observation time, instruments, algorithms, pretreatments for the radiation signals, and spatial resolutions) and then the PET estimates, the ensemble mean forest height was used as the final values for the forests in this study. It is admitted that although such reconstructed dataset could fully cover the globe, the uncertainties still existed, potentially resulting in the biases of the estimated PET. Despite that, we believed that our PET datasets may be better than others (e.g., CRU, GLEAM, PT-JPL, and hPET) based on two considerations, such as (1) vegetation height considered here (relative to studies without considerations of vegetation height), and (2) spatial differences in the vegetation height used in this study (relative to studies which used the height of each typical LULC but ignored spatial differences in vegetation height for a given LULC). Additionally, we have shown some discussion about the uncertainties induced by the vegetation height in the manuscript, such as "*The reconstructed global vegetation canopy height also has limitations, which may raise from (1) uncertainties in the retrieval algorithms and remote sensing data (Simard et al., 2011; Wang et al., 2016; Potapov et al., 2020; Lang et al., 2021, 2022), (2) neglecting the spatial differences in CRO and GRA heights and using an alternative specific value, and (3) not considering the inter-annual changes in the FR and SHRB canopy heights and the intra-annual cycle in the CRO and GRA heights. These limitations undermine the accuracy of the PET estimates.*" **(L585-589)**

Following Zhou et al. (2006), we employed an empirical model to estimate aerodynamic roughness length ($z_0$) in this study, which combined effects of vegetation height and leaf area index (LAI). Seen from the model, the spatial

variations of $z_0$ were mainly controlled by LAI and vegetation height, while LAI should be responsible for the temporal variations of $z_0$. Some studies have stated that besides LAI the temporal changes in vegetation height could also impact on $z_0$ (Peng et al., 2022; Guo et al., 2010; Zheng et al., 2014; Masseroni et al., 2015; Huang et al., 2016; Colin et al., 2006). However, due to lack of the time-varying vegetation height dataset, to consider the temporal changes in $z_0$ caused by the time-varying vegetation height is difficult. Therefore, without impacts of vegetation height, the temporal changes in $z_0$ based on the empirical model exist uncertainties.

We thank this reviewer very much for the valuable information about the 500 m daily aerodynamic roughness length developed by Peng et al. (2022). When we received the decision of this manuscript, we have tried to contact the authors, and would like to get this dataset to compare $z_0$ estimated in this study against $z_0$ from Peng et al. (2022). However, it is unfortunate that when we finished the responses, we still did not get this dataset (they replied our email but did not provide the dataset) and could not conduct comparisons. Moreover, we note that the time span of this dataset was shorter than our study period (mainly due to the availability of MODSI data). Therefore, even if we could get this dataset, it is still difficult to use this dataset to produce the PET before 2000.

*References:*

*Colin, J., Menenti, M., Rubio, E. and Jochum, A.: Accuracy Vs. Operability: a Case Study Over Barrax In The Context Of The DEMETER Project. AIP Conference Proceedings 852, 75–83, 2006.*

*Guo, X., Qin, J., Zhou, D., Yang, K. and Chen, Y.: Improving the Noah land surface model in arid regions with an appropriate parameterization of the thermal roughness length. Journal of Hydrometeorology, 11, 995–1006, 2010.*

*Huang, Y., Salama, M.S., Su, Z., Van Der Velde, R., Zheng, D., Krol, M. S., Hoekstra, A. Y. and Zhou, Y.: Effects of roughness length parameterizations on regional-scale land surface modeling of alpine grasslands in the Yangtze River basin. Journal of Hydrometeorology, 17, 1069–1085, 2016.*

*Masseroni, D., Facchi, A. and Gandolfi, C.: Estimation of zero-plane displacement height and aerodynamic roughness length on rice fields. Italian Journal of Agrometeorology, 20, 67–75. 2015.*

*Peng, Z., Tang, R., Jiang, Y., Liu, M. and Li, Z. L.: Global estimates of 500 m daily aerodynamic roughness length from MODIS data. ISPRS Journal of Photogrammetry and Remote Sensing, 183, 336–351, 2022.*

*Zheng, D., Van Der Velde, R., Su, Z., Booij, M. J. and Hoekstra, A. Y.: Assessment of roughness length schemes implemented within the Noah land surface model for highaltitude regions. Journal of Hydrometeorology, 15, 921–937, 2014.*

*Zhou, M. C., Ishidaira, H., Hapuarachchi, H.P., Magome, J., Kiem, A. S., Takeuchi, K.: Estimating potential evapotranspiration using Shuttleworth-Wallace model and NOAA-AVHRR NDVI data to feed a distributed hydrological model over the Mekong*

*River basin. Journal of Hydrology, 327(1–2): 151–173, 2006.*

**Comment 13: Line 246, Eq. (2c). The extinction coefficient 0.6 is empirical and may not be suitable for all vegetation types and fractional vegetation coverages. How did the authors consider this uncertainty?**

**Response:** Thanks for your comments. In the original manuscript, the extinction coefficient values have no differences among vegetation. Therefore, for considering such differences, we have given different extinction coefficient values for different LULC types (Table S5 in the revision) through reviewing the existing references, and recalibrated and reproduce the PET dataset.

In this study, we did not consider changes in the extinction coefficient with fractional vegetation coverages. To that end, the related discussion about this issue has been added in the revision, such as "*The canopy light extinction coefficient of $k_{ex}$, which represents a partitioning of radiant energy over vegetation canopy and soil surface, is a key factor affecting ecosystem carbon, water, and energy processes (Tahiri et al., 2006; Zhang et al., 2014). As a result, the accuracy of the SW model was believed to be associated with the $k_{ex}$ parameterization used in this study. Considering the physiological and morphological differences among terrestrial ecosystems (Emami-Bistghani et al., 2012; Zhang et al., 2014), we followed the popular biogeochemical models and remote sensing models of ET and gross primary productivity (e.g., Lund-Potsdam-Jena Dynamic Global Vegetation Model, and Vegetation Photosynthesis Model; Xiao et al., 2004; Sitch et al., 2003), and assumed this coefficient as a constant for every LULC type (Table S1). Despite that, it is notable that the $k_{ex}$ values vary with the growth of plant and/or vegetation coverages (Lindroth and Perttu, 1981; Aubin et al., 2000; Maddoni et al., 2001; Emami-Bistghani et al., 2012; Fauset et al., 2017). Emami-Bistghani et al. (2012) stated that with an increase of vegetation coverages, the $k_{ex}$ values decreased especially in early reproductive stage in sunflower cultivars. It suggested that the fixed $k_{ex}$ within the SW model existed limitations, potentially leading to uncertainties of the PET estimates. Implied by Tahiri et al. (2006), relative to the variable $k_{ex}$ values, the fixed values gave a less precise estimation of plant transpiration under irrigated maize.*" **(L522-534)**

**Comment 14: Line 265-266. How did the authors consider the uncertainty from the minimum threshold (30 W/m² for forests and 100 W/m² for other vegetation)?**

**Response:** The settings of the minimum threshold of canopy radiation ($K_{\downarrow dbl}$) for each vegetation type are mainly based on various references. For forests, some scholars recommend $K_{\downarrow dbl}$ to be 30 W/m² (Dickinson et al., 1984; Noilhan and Planton, 1989). Additionally, when conducting this study, we compared impacts of $K_{\downarrow dbl}$ (i.e., = 30

W/m$^2$ and 100 W/m$^2$) on performance of the SW model. Results showed that the SW model with $K_{\downarrow dbl}$ = 30 W/m$^2$ is slightly better than that with $K_{\downarrow dbl}$ = 100 W/m$^2$. This comparison confirms that the $K_{\downarrow dbl}$ = 30 W/m$^2$ for forests is reasonable. For other vegetation, $K_{\downarrow dbl}$ is recommended to be 100W/m$^2$ (Dickinson, 1984; Mu et al., 2017. Lo et al., 1997; Noilhan and Planton, 1989). Overall, the $K_{\downarrow dbl}$ values for forests and other vegetation have been proved to be reasonable by the previous studies, and we believed that the uncertainties induced by the different settings of $K_{\downarrow dbl}$ existed but limited.

*References:*

*Dickinson, R. E.: Modeling evapotranspiration for three-dimensional global climate models. Climate processes and climate sensitivity, 29, 58–72, 1984.*

*Lo, S. D., Chehbouni, A., Njoku, E., Saatchi, S., Mougin, E., Monteny, B.: An approach to couple vegetation functioning and soil-vegetation-atmosphere-transfer models for semiarid grasslands during the HAPEX-Sahel experiment. Agricultural and Forest Meteorology, 83(1–2), 49–74, 1997.*

*Mu, Y., Li, J. and Tong, X.: Evapotranspiration simulated by Penman-Monteith and Shuttleworth-Wallace models over a mixed plantation in the southern foot of the Taihang Mountain, northern China. Journal of Beijing Forestry University, 39(11), 35–44, 2017.*

*Noilhan, J. and Planton, S.: A simple parameterization of land surface processes for meteorological models. Monthly Weather Review, 117, 536–549, 1989.*

**Comment 15: Figure 5 was not referenced in the text.**

**Response:** Thanks for pointing out the mistake, and it has been corrected in this revision.

**Comment 16: Line 428. "Figure 1b" should be "Figure 11b"?**

**Response:** Thanks for pointing out the mistake. We have changed "Figure 1b" as "Figure 11b" in this revision.

**Comment 17: Line 440. "PE was the major contributor of PE"? Perhaps the later PE should be PET.**

**Response:** Sorry for this mistake. We have corrected "PE was the major contributor of PE" as "*PE was the major contributor of PET*" in this revision.

**Reviewer#2**

**Sun et al. provide a valuable estimate of global potential evapotranspiration (PET), which is highly beneficial for various research applications. The authors have taken into account various factors that influence PET estimation and have prepared multiple datasets to derive the PET values. The manuscript is well-structured, with clear method descriptions, data processing procedures, and explanations. Overall, the manuscript could be accepted with some comments to further enhance its quality.**

**Response:** We thank this reviewer very much for the positive comments and the valuable suggestions, which are believed to be very useful for us to improving the study. Seriously according to these suggestions, we have revised this manuscript, and the detailed information could be found below and the revised version.

*Major comments:*

**Comment 1: What are the valid temporal resolutions (e.g., hourly, monthly) for the inputs used in different models (i.e., equations)? Different meteorological datasets have varying temporal resolutions, such as 3-hourly for MSWX and monthly for others. When equations are used to calculate PET and related variables (e.g., D, Rn) to derive PET, it is important to consider whether these equations, as presented in the main text and supplementary materials, are valid for different temporal resolution inputs (e.g., 3-hourly vs. monthly). For example, can the SW equation be applied to different temporal resolution inputs (e.g., hourly or monthly)? It would be helpful to provide information on the validity of the equations for different temporal resolutions of inputs, whenever applicable. Additionally, the SW model was calibrated based on daily inputs (as shown in Figure 3). However, when applying the SW model globally, monthly inputs were used. The question arises whether it is appropriate to use a daily calibrated model for monthly inputs application.**

**Response:** Thank for you recommendations. When calibrating the SW model at the EC sites, the daily inputs was used. However, when applying the calibrated SW model at the globe, the monthly mean inputs was used. For confirming that the monthly inputs could be used to drive the SW model calibrated using the daily inputs, we have re-produce PET based on the daily meteorological variables from MSWX-Past, MERRA-2 and ERA-5, and then compared the new estimates to the original ones PET based on the monthly meteorological variables from the three datasets (Figure R1). Seen from Figure R1, it is not difficult to find that there are no evident differences in the two PET estimates, expect for April and May with larger ME (around 6 mm) and ubRMSE (around 12 mm) and lower KGE (around 0.4). This implies that the SW model established based on the daily data could be driven using the

monthly meteorological variables. Moreover, for stating that the SW model based on the daily data could be applicable at the monthly scale, we have added the related description in the revision, such as "*Considering that the SW model was calibrated with the daily EC measurements, it was necessary to examine whether this model could be applicable at the monthly scale. Therefore, we firstly compared the monthly PET estimated based on the daily and monthly meteorological variables from MSWX-Past, MERRA-2 and ERA-5 (not including CRU TS4.06 mainly due to it with a monthly scale). Various validation metrics showed that there were generally no evident differences in the two PET estimates (Figure S4). That is, the model established with the daily EC measurements could be driven using the monthly meteorological variables.*" **(L312-317)**

[Figure]

Figure R1: Comparison of the monthly PET estimates based on the daily and monthly meteorological variables. The outer edges of the boxes and the horizontal lines within the boxes indicate the 25th, 75th, and 50th percentiles of the validation metrics.

**Comment 2: It would be highly valuable if the authors could provide the datasets that were used to derive PET. This would include the following: EC related datasets, e.g., the original datasets after quality control, selected datasets with no soil water limits, etc; Finally processed canopy height, and/or its source datasets; Land use/land cover, LAI, saturated water content in soil, and the $CO_2$ concentration. Question for the $CO_2$ concentration, the seasonal cycle of $CO_2$ is different among different locations, e.g., between south and north hemisphere, will this affect your PET estimation?**

**Response:** We thank this reviewer very much for the suggestions. In fact, all the original datasets could be found in the corresponding websites or the literatures, which have been introduced in details in the paper. We are

pleasured to distribute the processed datasets, only if the readers contact us. Additionally, we have also showed a statement in the acknowledgement, such as "*The source code for the model used in this study and input files necessary to reproduce the simulations is available from the authors upon request (sun.s@nuist.edu.cn).*"

Thanks for your comments. For considering spatial differences in $CO_2$, we have recalibrated the SW model and reproduced PET, PT and PE with the gridded $CO_2$ dataset (i.e., the monthly $CO_2$ concentration with a spatial resolution of 1◦ × 1◦ and a time span of 1850–2013 from https://doi.org/10.5281/zenodo.5021361 (Cheng et al., 2022), and the monthly Global $CO_2$ Distribution product from Japan Meteorological Agency with a spatial resolution of 2◦ × 2◦ and a time span of 1985–2021). The detailed information could be found in the revision.

**Comment 3: About SW model:**

**1)  How about you add a concept diagram to show the structure of SW and related equation variables. One example for your reference, Figure 5 in Kochendorfer, J. P. and Ramírez, J. A.: Modeling the monthly mean soil-water balance with a statistical-dynamical ecohydrology model as coupled to a two component canopy model, Hydrol. Earth Syst. Sci., 14, 2099-2120, https://doi.org/10.5194/hess-14-2099-2010, 2010.**

**2)  EQ 1a, should the $C_c$ be removed. Why not the total latent heat doesn't equal the sum of canopy and vegetation latent heat fluxes, but need multiply the coefficients, could you add some explanation?**

$$\lambda ET = \cancel{C_c}PM_c + \cancel{C_s}PM_s?$$

**3)  It would be helpful if the authors could provide an explanation for the calculation of LAIe in EQ2b and clarify the reasoning behind this approach. What is the underlying assumption or basis for this equation? When you calibrate the SW model using EC data with filtering out the rain effects, so most of the LAI should be likely effective (no rain coverage over the leaves), why the LAIe is still calculated in EQ2b (i.e., LAIe is less than LAI when LAI>2)?**

**When you apply the calibrated SW model for global, the effective LAI should be considered due the reasons of rain. The LAIe should be related to different conditions (e.g., different rainfall intensity) but not considered in the EQ2b.**

**It seems there is inconsistence. When calibrating SW model using no-rain effects data, but applying effective LAI (i.e., from EQ2) when the rain effects is small. But the same LAIe equation are used for the global application, when under some conditions rain effects may be large. The PET calculation should also include the maximum ET during rain or after rain events, right?**

**4) I wonder how to consider the LAIe for PET calculation at EC site level and global grid. How is the SW**

**PET sensitivity to LAIe?**

**Here I only say rain effects on LAIe, but other factors may also effect LAIe calculations, e.g., snow.**

**Response:** 1) Thanks for your suggestion. The schematic of the Shuttleworth-Wallace (1985) two component canopy model has been shown below (Figure R2) and in the revised manuscript.

[Figure]

Figure R2: Schematic description of the energy partitioning for a canopy with the SW model.

2) We are sorry that the EQ. 2 confuses you. Now, we have rewritten this equation below and in the revision.

$$\begin{cases} \lambda ET = \lambda Tr + \lambda E & (R1) \\ \lambda Tr = C_c PM_c & (R2) \\ \lambda E = C_s PM_s & (R3) \end{cases}$$

It is not difficult to find that the total latent heat ($\lambda ET$) equals to the sum of canopy ($\lambda Tr$) and vegetation latent heat fluxes ($\lambda E$). In the following text, we will explain why $PM_c$ dose not equal to $\lambda Tr$ and $PM_s$ dose not equal to $\lambda E$. Based on Shuttleworth-Wallace (1985), $\lambda Tr$ and $\lambda E$ can be expressed as,

$$\begin{cases} \lambda Tr = \dfrac{\Delta(A - A_{soil}) + \rho c_p D_0 / r_a^c}{\Delta + \gamma(1 + r_s^c / r_a^c)} & (R4) \\ \lambda E = \dfrac{\Delta A_{soil} + \rho c_p D_0 / r_a^s}{\Delta + \gamma(1 + r_s^s / r_a^s)} & (R5) \\ D_0 = D + [\Delta A - (\Delta + \gamma)\lambda ET] r_a^a / \rho c_p & (R6) \end{cases}$$

where $D_0$ represents the vapour pressure deficit at the canopy source height. Therefore, by introducing EQ. (R6) into EQs. (R4) and (R5) and then EQ. (R1), we can obtain,

$$\lambda ET\{[(\Delta + \gamma)r_a^s + \gamma r_s^s][(\Delta + \gamma)r_a^c + \gamma r_s^c] + (\Delta + \gamma)r_a^a[(\Delta + \gamma)r_a^c + \gamma r_s^s] + (\Delta + \gamma)r_a^a[(\Delta + \gamma)r_a^s + \gamma r_s^s]\}$$

$$= (\Delta A_{soil}r_a^s + \rho c_p D + \Delta A r_a^a)[(\Delta + \gamma)r_a^c + \gamma r_s^c]$$

$$+ [\Delta(A - A_{soil})r_a^c + \rho c_p D + \Delta A r_a^a][(\Delta + \gamma)r_a^s + \gamma r_s^s] \qquad (R7)$$

If we define

$$
\begin{cases}
R_a = (\Delta + \gamma)r_a^a & (R8)\\
R_s = (\Delta + \gamma)r_a^s + \gamma r_s^s & (R9)\\
R_c = (\Delta + \gamma)r_a^c + \gamma r_s^c & (R10)
\end{cases}
$$

and substitute these into EQ. (R7), we can get

$$
\lambda ET(R_s R_c + R_c R_a + R_s R_a)
$$

$$
= [\Delta A(r_a^a + r_a^s) + \rho c_p D - \Delta(A - A_{soil})r_a^s]R_c
$$

$$
+ \left[\Delta A(r_a^a + r_a^c) + \rho c_p D - \Delta A_{soil}r_a^c\right]R_s \qquad (R11)
$$

Based on $R_a + R_s = (\Delta + \gamma)(r_a^a + r_a^s) + \gamma r_s^s$ and $R_a + R_c = (\Delta + \gamma)(r_a^a + r_a^c) + \gamma r_s^c$, we can rewrite EQ. (R11) as,

$$
\lambda ET(R_s R_c + R_c R_a + R_s R_a) = PM_s R_c(R_s + R_a) + PM_c R_s(R_c + R_a) \qquad (R12)
$$

where $PM_s = \dfrac{\Delta A + [\rho c_p D - \Delta r_a^s(A - A_{soil})]/(r_a^a + r_a^s)}{\Delta + \gamma[1 + r_s^s/(r_a^a + r_a^s)]}$ and $PM_c = \dfrac{\Delta A + (\rho c_p D - \Delta r_a^c A_{soil})/(r_a^a + r_a^c)}{\Delta + \gamma[1 + r_s^c/(r_a^a + r_a^s)]}$.

Finally, EQ. (R12) can be rewritten as,

$$
\begin{cases}
\lambda ET = C_c PM_c + C_s PM_s & (R13)\\
C_c = \left[1 + \dfrac{R_c R_a}{R_s(R_c + R_a)}\right]^{-1} & (R14)\\
C_s = \left[1 + \dfrac{R_s R_a}{R_c(R_s + R_a)}\right]^{-1} & (R15)
\end{cases}
$$

Seen from the derivation above, we could find that $PM_c$ dose not equal to $\lambda T$ and $PM_s$ dose not equal to $\lambda E$. However, $\lambda Tr$ and $PM_c$ ($\lambda E$ and $PM_s$) exist a certain functional relationship. Because the derivations above can be found in Shuttleworth-Wallace (1985), in the revision we will not provide the complete derivation. However, the related revision has been shown in the revision for more clarity.

3) Maybe this reviewer misunderstands the definition of the effective LAI (*LAIe*) used in this study, mainly due to our uncomplete description of *LAIe*. Here, *LAIe* is the LAI that actively contributes to the surface heat and vapour transfer, and is generally the upper, sunlit portion of a dense canopy (Allen et al., 1998). Therefore, the canopy resistance ($r_s^c$) is not dependent on LAI rather than *LAIe*. Many studies (Gardiol et al., 2003; Li et al., 2016; Zhang et al., 2016) have showed that the function between $r_s^c$ and *LAIe* can be expressed as $r_s^c = r_{smin}/LAIe$, when no considering other environmental factors (vapour pressure deficit, air temperature, soil moisture and $CO_2$ concentration). Due to illumination-induced stomatal closure deeper in the canopy, there exist a complex functional relationship between LAI and *LAIe* (Gardiol et al., 2003), such as

$$
LAIe = \begin{cases}
LAI, & LAI \leq 2\\
2, & 2 < LAI < 4\\
LAI/2, & LAI \geq 4
\end{cases} \qquad (R16)
$$

In this study, the calibrations for the SW PET model were based on the EC observations in days without rain, mainly because the EC system can not observe ET in rainy days. The processing procedure of filtering out rainy days only aims to remove the invalid ET observations and their corresponding climate variables.

Seen from definition, the used *LAIe* in this study is independent on rain. However, we agree with the reviewer that when rain happens, the leaves will be covered by rain at a certain time and then the *LAIe* will be smaller compared to the period without rain. Therefore, the corresponding $r_s^c$ decreases, potentially increasing transpiration and then ET. Considering the rainy or snow days to be smaller relative to other days, we believe that such impacts on *LAIe* and then PET may be much limited. Anyway, we have showed the related discussion in this revision, such as "*Considering that this LAI product was based on the 8-day maximum value composite for removing impacts of cloudy days, the LAIe (based on EQ. 3b) was potentially larger than its authentic value due to some leaves covered by rain or snow. Thus, from EQ. 3b, $r_s^c$ may be slightly underestimated, leading to an overestimation in PT and PET.*" **(L578-580)**

4) For quantitatively examining impacts of *LAIe* on the PET estimates, we have designed four experiments with *LAIe* increases by 1%, 5%, 10% and 20% at EC site level. Comparison between the original and the new estimates showed that with increases in *LAIe* the PET and PT (PE) would like to increase (decrease), mainly due to $r_s^c$ reductions induced by increased *LAIe* (Figure R3(a1-3)). Furthermore, we have also calculated sensitivity of PET, PT and PE to *LAIe* changes (i.e., PET, PT or PE changes in response to 1% changes in LAIe). Results showed that in response to LAIe increases by 1%, PET (PT) would like to generally increase by 0.4-0.6% (0.4-0.8%) while PE would like to decline by 0.05-0.4% (Figure R3(b1-3)). Moreover, the sensitivity of PET, PT and PE to *LAIe* changes varied among LULC types. Overall, the PET, PT and PE is sensitive to *LAIe* changes. However, we should note the fact that relative to days without rain and snow, the rainy and snow days were usually much smaller, and the rain or snow intercepted by leaves may be evaporated quickly or blown away by wind. Therefore, we believed that the potential uncertainties related to *LAIe* was much limited.

[Figure]

Figure R3: Changes in PET (a1), PT (a2) and PE (a3) with *LAIe* increases by 1%, 5%, 10% and 20%, and sensitivity of PET (b1), PT (b2) and PE (b3) to *LAIe* changes (which represents PET, PT or PE changes in response to 1% changes in *LAIe*)

*References:*

*Allen, R.G., Smith, M., Perrier, A., Pereira, L.S., 1993. Updated reference evapotranspiration definition and calculation procedures, Revision of FAO Methodologies for Crop Water Requirements. 36 pp.*

*Gardiol, J. M., Serio, L. A. and Maggiora, A. I. D.: Modeling evapotranspiration of corn (Zea mays) under different plant densities. Journal of Hydrology, 217, 188–196, 2003.*

*Li, X., Kang, S., Li, F., Jiang, X., Tong, L., Ding, R., Li, S. and Du, T.: Applying segmented Jarvis canopy resistance into Penman-Monteith model improves the accuracy of estimated evapotranspiration in maize for seed production with film-mulching in arid area. Agricultural Water Management, 178, 314–324, 2016.*

*Zhang, B. Z., Xu, D., Liu, Y., Li, F. S., Cai, J. B. and Du, L. J.: Multi-scale evapotranspiration of summer maize and the controlling meteorological factors in north China. Agricultural and Forest Meteorology, 216, 1–12, 2016.*

**Comment 4: It would be beneficial to have a clear explanation of how PT and PE are calculated differently. Currently, there are no specific equations provided to illustrate the calculations for PE and PT. It appears that PT is derived from PMc, while PE is also derived from PMc. It is important to clarify this distinction and explicitly mention that PT and PE are derived from PMc in the study. Additionally, please review the sentence in line 238 that states "while PMc and PMs are the soil and vegetation latent heat fluxes (W/m2)" to ensure the correct explanation of parameters and variables throughout the equations. Furthermore, it is worth considering the inclusion of discussions on the explicit consideration of plant hydraulics in recent**

land surface models (e.g., CLM5 in 2019, NOAH-MP in 2021, CoLM in 2022) as it relates to transpiration simulations. I am interested to know whether the SW model implicitly incorporates plant hydraulics or if there are potential improvements that could be made to the PET estimation by integrating plant hydraulics within the framework of the SW model. Section 4.2 would be an appropriate place to include such discussions.

Related references:

https://agupubs.onlinelibrary.wiley.com/doi/10.1029/2018MS001500

https://agupubs.onlinelibrary.wiley.com/doi/full/10.1029/2020MS002214

**Response:** Thanks for your suggestions. Now, we have shown the specific equations (EQs. 1b and 1c in the revision) to illustrate the calculations for E or PE and T or PT in this revision. We have corrected the mistake as "*Based on EQ. 1b (EQ.1c), $T_r$ (E) can be obtained with $C_cPM_c$ ($C_sPM_s$) divide by $\lambda$.*", and details could be found in the revised manuscript.

We thank the reviewer very much for providing us the useful references about transpiration simulations. In this study, we employed an empirical model (i.e., Jarvis model) to describe impacts of environmental factors on $r_s^c$ and then transpiration, mainly because this model has relatively simple parameterizations and has been widely and successfully used in many hydrological, ecological, meteorological and agricultural studies. However, it should be noted that this study focuses on PET, evaporation and transpiration rather than their actual values. That is, the soil moisture stress was not considered here (i.e., $F_4 = 1$ within *EQ. 3f*), i.e., no water stress for evapotranspiration process. Through reading the two important references recommended by the reviewer, we found that these two papers focused on improvement of vegetation water stress and root water uptake, and therefore to discuss potential applications for the SW PET model may be beyond of our scope. Anyway, we have to admit that the reviewer provided us a valuable suggestion for our future study, i.e., taking the two literatures as reference to define the water stress factor through incorporating plant hydraulics, and then estimating ET, evaporation and transpiration using our PET estimates.

**Comment 5:** You have presented the trends of PET, PE, and PT, as well as the contributions of changes in PE and PT to changes in PET. I am curious to know which factors, such as changes in meteorological forcings, contribute to the observed changes in PET, PE, or PT. For instance, could the global temperature increase be a significant driver? Furthermore, it would be valuable to include a discussion on how the phenomenon of Earth greening, such as an increase in LAI, may influence your trend analysis. Consider

**commenting on the potential impacts of Earth greening on the observed trends in PET, PE, and PT.**

**Response:** Thanks for your suggestion. This manuscript aims to introduce the SW "dual source" PET dataset. The work related to attribution analysis that you mentioned is currently underway, and the preliminary results are listed here (Figure R4). As you mentioned, the global temperature increases and the LAI increases are the main factors affecting potential global evapotranspiration changes. Notably, compared to the dominant factor of TA for changes in PET, the area percentages for the dominant factor of LAI for changes in PT and PE is much larger. This is mainly because of the offset effects between positive contributions of greening to PT and negative contributions of greening to PE. Considering that this paper is mainly about the description of the SW PET dataset, the greening impacts on PET will not be shown in this paper. Actually, just considering the greening impacts on evapotranspiration process (e.g., we have stated its importance in the manuscript, such as "*Recently, with climate change and/or intensified human activities, vegetation has greatly changed on regional and even the global scales (Zhu et al., 2016; Chen et al., 2019), including shifts in vegetation types and vegetation greening (i.e., increases in LAI or other vegetation indices), which have altered the allocation of available water and energy (Zhou et al., 2016, 2018; Sun et al., 2022).*"), the observed LAI was selected an important input to accurately estimate PET.

[Figure]

Figure R4: The average contribution of $CO_2$, LAI, net radiation (RN), relative humidity (RH), temperature (TA), and wind speed (WS) to the global and Köppen-Geiger climate regions annual PET (a), PT (c), and PE (e) trends from 1982 to 2015, and the spatial distribution of dominant factors for annual PET (b), PT (d), and PE (f) trends.

*Minor comments:*

**Comment 1: In the introduction, the authors mentioned different types of models to calculate the PET, and give examples for each type model (e.g., Penman-Monteith). It would be great if the authors can also provide the equations for these example models in the supplementary, so the readers can better compare them. Also please provide the information about the common temporal resolutions of the inputs for these models, hourly or daily or, …**

**Response:** Thanks for your suggestion. The typical PET models have been added in the revised supplementary materials (i.e., Table S1 in the revision), and please see below.

Table R1: Some typical PET models

| Proposed by | Equation | Timescale |
|---|---|---|
| Dalton (1802)[a] | $PET = (0.3648 + 0.07223u)(e_s - e_a)$ | Monthly |
| Thornthwaite (1948)[b] | $PET = 16N_m(10T_{mean})$ | Monthly |
| Turc (1961)[c] | $PET = 0.013[N_{mean}/(T_{mean} + 15)](R_n + 50)$ | Daily/Monthly |
| Hargreaves and Samani (1985)[c] | $PET = 0.0145K_{RS}R_e(T_a + 17.8)T_d^{0.5}$ | Daily/Monthly/Yearly |
| Penman (1948)[d] | $PET = \dfrac{\Delta H + \gamma(e_s - e_a)f(u)}{\Delta + \gamma}$ | Daily |
| Monteith (1965)[d] | $PET = \dfrac{\Delta(R_n - G) + [\rho c_p(e_s - e_a)]/r_a}{\Delta + \gamma(1 + r_s/r_a)}$ | Daily |
| Allen et al. (1998) (FAO-56 Penman-Monteith)[d] | $PET = \dfrac{0.408\Delta(R_n - G) + \gamma u(e_s - e_a)[900/(T_{mean} + 273)]}{\Delta + \gamma(1 + 0.34u)}$ | Hourly/Daily/Monthly |

Note: [a], [b], [c] and [d] represent mass-transfer-based, temperature-based, radiation-based, and combination PET models, respectively. $T_d$ are differences in the maximum ($T_{max}$) and the minimum ($T_{min}$) temperatures, i.e., $T_d = T_{max} - T_{min}$. $K_{RS}$ is empirical coefficient fitted to $R_{ds}/R_e$ versus $T_d$ data. $f(u)$ is a function of wind speed. $r_s$ represents surface or canopy resistance, while and $r_a$ represents aerodynamic resistance.

**Comment 2: In the supplementary, where is "EQ. S7a", should be EQ S4a?. Bold the titles of "The ERA-5 D" and "The MERRA-2 D"**

**Response:** Thanks. We have corrected this mistake. The titles of "ERA-5D" and "MERRA-2D" have been bolded.

**Comment 3: Lang et al has another dataset from the webpage: https://langnico.github.io/globalcanopyheight/, what is the difference between this version of data and the Lang data you used in your study. Should this new data better than what the Lang data you used. You may add some discussion of this. It seems that the canopy height is temporally static, but the LULC changes yearly. How to make the consistence for each year's LULC's canopy height for a given grid if LULC changes happens.**

**Response:** Thanks for your suggestion. The data you mentioned is indeed more novel, high-resolution, and covers a wider range than the Lang data used in this study. It was updated in May 2020, when our global vegetation height data production work was completed. For quantifying impacts of different canopy height data on our PET estimates, we have used this new canopy height data to re-estimate PET (named as $PET_{new}$) during 1982-2015 in 3 FR and 3 SHRB plots (Figure R5), and then compared the two PET estimates (represented as ($PET_{new}$ – PET in this study) divided by PET in this study). Overall, the different canopy height datasets could cause differences in the PET and its two sub-components estimates, but we should note that the differences of the three variables were generally between -6% and 6%, especially for the FR plots generally between -1% and 1%. This suggested that the differences in PET and its two sub-components induced by different canopy height datasets were limited. Moreover, when producing PET, we have used four canopy height datasets for decreases uncertainties. Meanwhile, considering uncertainties related to the canopy height datasets, the related discussion was also shown in the manuscript, such as "*The reconstructed global vegetation canopy height also has limitations, which may raise from (1) uncertainties in the retrieval algorithms and remote sensing data (Simard et al., 2011; Wang et al., 2016; Potapov et al., 2020; Lang et al., 2021, 2022), (2) neglecting the spatial differences in CRO and GRA heights and using an alternative specific value, and (3) not considering the inter-annual changes in the FR and SHRB canopy heights and the intra-annual cycle in the CRO and GRA heights. These limitations undermine the accuracy of the PET estimates.*" **(L585-589)**

[Figure]

Figure R5: Spatial distribution of 3 FR and 3 SHRB plots (a), and differences in PET (b1-3 and c1-3), PT (b4-6 and c6-6) and

PE (b7-9 and c7-9)

To make the consistence for each year's LULC's canopy height for a given grid if LULC changes happens, we

obtained the canopy height at a grid with LUCC using the mean height from the four nearest neighboring grids

with the same LULC. Now, we have added the related description in this revision, such as "*In the grid with LULC*

*changes in a certain year, its new h value was assigned as the mean h value of its four nearest neighboring grids*

*with the same LULC.*" **(L234-236)**

**Comment 4: L230, $r_{smin}$ should be defined when it first appears.**

**Response:** Thanks. The definition of $r_{smin}$ has been added where it first appears.

**Comment 5: How are the averages of PET, PE, PT are calculated based on grid average or area average?**

**Please mention it in the text.**

**Response:** The averages of PET, PE, and PT are estimated based on the area-weighted method. For clarity, the

method has been mentioned in this revision.

**Comment 6: Check the Figure 8, the colors for scatters and PE PT lines are not consistent.**

**Response:** Thanks for pointing out the mistake. The mistake has been corrected, and please see below or this revision.

[Figure]

Figure R6: Climatological monthly PET, PE, PT, PE/PET and PT/PET averaged over the globe, each hemisphere, and each KG climate region.

**Comment 7: For the calibration of $r_{smin}$, it would be helpful to know the range of variation among the 10 $r_{smin}$ values for each specific plant functional type (PFT) site.**

**Response:** Thanks for your suggestion. For obtaining the first 10 best $r_{smin}$ values, the method of Hu et al. (2009) was used in this study, and the highest $KGE$ was used as the criteria. Therefore, we believe that the first 10 best $r_{smin}$ values should be close. Through checking the range (reflected by the standard deviation) of variation among the 10 $r_{smin}$ values for each site (Table R2), we could find that the standard deviation does be much small at each site. Therefore, we think that it is not necessary to show the range the range of variation among the 10 $r_{smin}$ values for each site, because the first 10 highest $KGE$ correspond to the first 10 best $r_{smin}$ and the range should be small.

Table R2: Standard deviation of the 10 $r_{smin}$ values for each site

| GLASS-GLC types | Names | Standard deviation | GLASS-GLC types | Names | Standard deviation | GLASS-GLC types | Names | Standard deviation |
|---|---|---|---|---|---|---|---|---|

| | | | | | | | | |
|---|---|---|---|---|---|---|---|---|
| CRO | US-Bo1 | 0.33 | FR | DE-Obe | 0.17 | SHRB | ES-Lma | 0.60 |
| CRO | IT-CA2 | 0.26 | FR | DE-Tha | 0.31 | SHRB | AU-TTE | 0.23 |
| CRO | US-CRT | 0.49 | FR | DK-Sor | 0.29 | SHRB | SD-Dem | 0.17 |
| CRO | US-Twt | 0.52 | FR | FI-Hyy | 0.31 | SHRB | AU-Dry | 0.15 |
| CRO | BE-Lon | 0.28 | FR | FR-Pue | 0.39 | SHRB | AU-DaS | 0.52 |
| CRO | DE-Kli | 0.27 | FR | IT-Col | 0.37 | SHRB | AU-Cpr | 0.30 |
| CRO | FR-Gri | 0.24 | FR | IT-Lav | 0.21 | GRA | AU-Sam | 0.20 |
| CRO | US-ARM | 0.28 | FR | IT-PT1 | 0.22 | GRA | US-Aud | 0.34 |
| CRO | DE-Geb | 0.45 | FR | IT-Ren | 0.21 | GRA | PT-Mi2 | 0.19 |
| CRO | US-Ne1 | 0.33 | FR | IT-Ro2 | 0.17 | GRA | ES-VDA | 0.13 |
| CRO | US-Ne2 | 0.40 | FR | IT-SRo | 0.38 | GRA | HU-Bug | 0.32 |
| CRO | US-Ne3 | 0.41 | FR | NL-Loo | 0.36 | GRA | US-Fpe | 0.27 |
| CRO | MSE | 0.27 | FR | RU-Fyo | 0.37 | GRA | CN-Du2 | 0.26 |
| FR | AU-Cow | 0.46 | FR | US-Blo | 0.31 | GRA | CN-Du3 | 0.33 |
| FR | AU-Ctr | 0.20 | FR | US-Me2 | 0.30 | GRA | RU-Ha1 | 0.41 |
| FR | CA-Qcu | 0.17 | FR | US-NR1 | 0.33 | GRA | US-ARb | 0.37 |
| FR | DE-Bay | 0.19 | FR | US-Syv | 0.22 | GRA | US-ARc | 0.37 |
| FR | FHK | 0.22 | FR | FR-Hes | 0.31 | GRA | CN-HaM | 0.29 |
| FR | AU-Lox | 0.38 | FR | GDK | 0.52 | GRA | IT-Tor | 0.15 |
| FR | AU-Rob | 0.36 | FR | TMK | 0.20 | GRA | US-LWW | 0.42 |
| FR | AU-Tum | 0.13 | FR | TSE | 0.32 | GRA | AT-Neu | 0.44 |
| FR | AU-Wom | 0.22 | FR | US-Moz | 0.36 | GRA | AU-Rig | 0.20 |
| FR | BE-Vie | 0.20 | FR | US-SP1 | 0.11 | GRA | AU-Emr | 0.17 |
| FR | BR-Sa3 | 0.28 | FR | US-SP2 | 0.20 | GRA | US-AR2 | 0.11 |
| FR | CA-Gro | 0.12 | FR | US-SP3 | 0.17 | GRA | CN-Cng | 0.26 |
| FR | CA-Qfo | 0.37 | SHRB | US-KS2 | 0.21 | GRA | US-Goo | 0.22 |
| FR | CA-SF1 | 0.39 | SHRB | IT-Noe | 0.29 | GRA | US-AR1 | 0.26 |
| FR | CA-SF2 | 0.17 | SHRB | CA-SF3 | 0.43 | GRA | DE-Gri | 0.28 |
| FR | CA-TP1 | 0.41 | SHRB | ES-Amo | 0.35 | GRA | AU-Stp | 0.20 |

| FR | CA-TPD | 0.38 | SHRB | AU-RDF | 0.17 | GRA | US-SRG | 0.19 |
| FR | DE-Hai | 0.31 | SHRB | US-Ton | 0.20 | GRA | US-Wkg | 0.45 |
| FR | DE-Lkb | 0.21 | SHRB | BW-Ma1 | 0.21 | GRA | US-Var | 0.24 |

**A LIST OF ALL RELEVANT CHANGES MADE IN THE REVISION**

1. This sentence in L33-35 has been changed as "*The global mean annual PET was 1198.96 mm with PT/PET of 41% and PE/PET of 59%, and moreover controlled by PT and PE over 41% and 59% of the globe, respectively. Globally, the annual PET and PT significantly (p<0.05) increases by 1.26 mm/yr and 1.27 mm/yr over the last 34 years, followed by a slight increase in the annual PE.*"

2. "*in Table S1*" has been inserted in L51, L53, L55 and L57.

3. This sentence in L71-75 has been changed as "*Usually, many vegetation types co-exist over the land, and there are always some parts or periods where or when the vegetation not "closed" (i.e., open canopy that light can penetrate to the ground). The big leaf assumption potentially limits the applicability of the Penman-Monteith models under various vegetation distribution conditions, e.g., better (worse) performance under complete and homogeneous (sparse and inhomogeneous) vegetation distribution conditions (Shuttleworth and Wallace, 1985; Stannard, 1993; Yang and Shang, 2012; Huang et al., 2020).*"

4. "vegetation height" in Line 97 has been changed as "*vegetation height datasets*".

5. "Leaf Area Index" in Line 98 has been changed as "*Leaf Area Index datasets*".

6. "Table S1" in Line 173 has been changed as "*Table S2*".

7. "CRU" in Line 179 has been changed as "*Climatic Research Unit (CRU)*".

8. "*(Table 1)*" has been inserted after "was used here" in L190.

9. "(Xiao et al., 2016, 2017)" in L196 has been changed as "*(Xiao et al., 2016, 2017; Table 1)*".

10 "The global saturated water content in soil at the first soil layer … (Thoning et al., 2022), respectively." in L196 has been revised as "*The global saturated soil water content in the first soil layer (i.e., 0–0.0451 m) was collected from http://globalchange.bnu.edu.cn/research/soil5.jsp (Dai et al., 2019a, 2019b; Table 1), while the 1850–2013 monthly $CO_2$ concentration at 1° × 1° spatial resolution was downloaded from https://doi.org/10.5281/zenodo.5021361 (Cheng et al., 2022; Table 1). Because this $CO_2$ dataset missed the data in 2014 and 2015, the 1985–2021 monthly Global $CO_2$ Distribution (GCD) product from Japan Meteorological Agency at a 2° × 2° spatial resolution (https://www.data.jma.go.jp/ghg/kanshi/co2data/co2_mapdata_e.html; Maki et al., 2010; Nakamura et al., 2015; Table 1) was used to estimate the monthly $CO_2$ concentration in the two missing years by the linear regression method. In detail, we firstly resampled the GCD data into 1o × 1o resolution, and used the 1985–2013 GCD as independent variable and Cheng's data as dependent variable to fit the linear*

*regressions for each month and each grid. Subsequently, based on the GCD data, these regressions were used to calculate the monthly $CO_2$ concentration at each grid in 2014 and 2015. The validation results of the established regression were in Figure S2.*"

11. "*Table 1*" has been inserted after "and h-Lang" in L211 and "(Yu et al., 2020" in L212.

12. "Table S2" has been changed as "*Table S3*" in L230.

13. "Table S3" has been changed as "*Table S4*" in L232.

14. The sentence of "*In the grid with LULC changes in a certain year, its new h value was assigned as the mean h value of its four nearest neighboring grids with the same LULC. An example of the reconstructed canopy h map in 1982 was shown in Figure S3.*" has been inserted after "where the h values varied due to LULC changes" in L234.

15. The sentence in L240-241 has been changed as "*(cyan color in Figure 2), (2) generating the global monthly PET using the calibrated SW model with the final minimum stomatal resistance ($r_{smin}$, s/m) values and various inputs (orange color in Figure 2)*".

16. "*Figure 3;*" has been inserted before "Shuttleworth and Wallace, 1985, 1990" in 246.

17. EQ.1 has been changed as:

$$
\begin{cases}
\lambda ET = \lambda T_r + \lambda E & \text{(1a)} \\
\lambda T_r = C_c PM_c & \text{(1b)} \\
\lambda E = C_s PM_s & \text{(1c)} \\
PM_c = \dfrac{\Delta A + (\rho c_p D - \Delta r_a^c A_{soil})/(r_a^a + r_a^c)}{\Delta + \gamma[1 + r_s^c/(r_a^a + r_a^c)]} & \text{(1d)} \\
PM_s = \dfrac{\Delta A + [\rho c_p D - \Delta r_a^s (A - A_{soil})]/(r_a^a + r_a^s)}{\Delta + \gamma[1 + r_s^s/(r_a^a + r_a^s)]} & \text{(1e)} \\
C_c = [1 + \dfrac{R_c R_a}{R_s(R_c + R_a)}]^{-1} & \text{(1f)} \\
C_s = [1 + \dfrac{R_s R_a}{R_c(R_s + R_a)}]^{-1} & \text{(1g)} \\
R_a = (\Delta + \gamma) r_a^a & \text{(1i)} \\
R_s = (\Delta + \gamma) r_a^s + \gamma r_s^s & \text{(1j)} \\
R_c = (\Delta + \gamma) r_a^c + \gamma r_s^c & \text{(1k)}
\end{cases}
$$

18. The sentence in L248-250 has been revised as "*where λET is the total latent heat flux (W/m²), i.e., the sum of canopy (λTr) and vegetation latent heat fluxes (λE), where Tr and E represent transpiration and soil evaporation, respectively; PMc and PMs are the closed vegetation and bare soil latent heat fluxes (W/m²)*".

19. "1b", "1d", "1e" and "1f" in L250-253 have changed as "*1d*", "*1e*", "*1f*" and "*1g*".

20. The sentence of "*Based on EQ. 1b (EQ.1c), $T_r$ (E) can be obtained with $C_c PM_c$ ($C_s PM_s$) divide by λ.*" has been inserted after "resistance (s/m)" in L256.

21. EQ. 2c has been changed as: "$R_{n,soil} = R_n exp\,(-k_{ex} LAI)$" in L259.

22. The sentence of "*$k_{ex}$ represents light extinction coefficient, which varies with LULC types (Table S5).*" has

been inserted after "(i.e., EQ. 2c; Mo et al., 2004)" in L260.

23. "Table S4" has been changed as "*Table S5*" in L265.

24. "2a", "2b", "2c", "2d", "2e", "2f", and "2g" has been changed as "*3a*", "*3b*", "*3c*", "*3d*", "*3e*", "*3f*", and "*3g*" in L274.

25. "Table S4" has been changed as "*Table S5*" in L279.

26. "EQ. 3d" in L286 has been changed as "*EQ. 4d*".

27. "Table S5" in L291 has been changed as "*Table S6*".

28. "unRMSE" in L295, L299, L333, L356, L344, L349, L358 and L359 has been changed as "*ubRMSE*"

29. These sentences of "*Considering that the SW model was calibrated with the daily EC measurements, it was necessary to examine whether this model could be applicable at the monthly scale. Therefore, we firstly compared the monthly PET estimated based on the daily and monthly meteorological variables from MSWX-Past, MERRA-2 and ERA-5 (not including CRU TS4.06 mainly due to it with a monthly scale). Various validation metrics showed that there were generally no evident differences in the two PET estimates (Figure S4). That is, the model established with the daily EC measurements could be driven using the monthly meteorological variables. Mainly due to the GLASS-GLC with the shortest time span, the global SW PET was produced at GLASS-GLC grid and monthly scale during 1982–2015. Therefore, before running the calibrated SW model, the meteorological, the GLASS LAI, and the $CO_2$ concentration datasets were resampled to a spatial resolution of 5000 m.*" has been added before "The monthly mean meteorological and LAI datasets" in L319.

30. The sentence of "*Notably, the area-weighted method was used to estimate the regional mean PET, PT and PE.*" has been added after "characteristics of PET partitioning" in L325.

31. "(Figures 3a and 3c)" in L330 has changed as "*(Figures 4a and 4c)*".

32. "(Figure 3a)" in L332 has changed as "*(Figures 4a)*".

33. "-0.06" in L336 has changed as "*0.06*".

34. "0.88" in L336 has changed as "*0.60*".

35. "0.87" in L336 has changed as "*0.85*".

36. "Figure 3a vs. 3c, and Figure 3b vs. 3d" in L338 has changed as "*Figure 4a vs. 4c, and Figure 4b vs. 4d*".

37. "Figure 4" in L339 has changed as "*Figure 5*".

38. "R was above 0.84" in L339 has changed as "*R was above 0.83*".

39. "> 0.70 mm/day" in L345 has changed as "*between 0.53 mm/day and 0.85 mm/day*".

40. "*for FR*" has been inserted before "in cross-validation mode" in L346.

41. "*(Figure 6)*" has been inserted before", the simulated PET" in L348.

42. "with *R* above 0.75, *ME* 330 between -0.11 and 0.11 mm" has changed as "*with R above 0.74, ME between -0.12 and 0.11 mm*".

43. "(Figure 5a vs. 5e, Figure 5b vs. 5f, Figure 5c vs. 5g, and Figure 5d vs. 5h)" in L350 has changed as "*(Figure 6a vs. 6e, Figure 6b vs. 6f, Figure 6c vs. 6g, and Figure 6d vs. 6h)*".

44. "which are shown in Figure 6. Except for several sites for FR and CRO," in L354 has changed as "*which are shown in Figure 7. Except for several sites for SHRB, GRA and CRO,*".

45. "SHRB" in L359 has been changed as "*CRO*".

46. "(Figure 6c)" in L359 has been changed as "*(Figure 7c)*".

45. "(Figure 6d)" in L359 has been changed as "*(Figure 7d)*".

47. "(Table S5)" in L366 has been changed as "*(Table S6)*".

48. "S3" in L367 has been changed as "*S5*".

49. "As seen from … and 728 mm, respectively" in L369-370 has changed as "*As seen from Figures 8a, 8c and 8e, the global mean climatological annual PET, PT and PE were 1198.96 mm, 481.12 mm and 717.74 mm, respectively.*"

50. "with a range from … in Arid region for" in L372-374 has been changed as "*with a range from 319.29 mm in Polar region to 1590.57 mm in Tropical region for PET, from 37.95 mm in Polar region to 1122.42 mm in Tropical region for PT, and from 248.31 mm in Cold region to 1379.16 mm in Arid region for*".

51. "Figure 7" in L376, L378 and L382 has been changed as "*Figure 8*".

52. "Figure 8" in L388 and L393 has been changed as "*Figure 9*".

53. "Figures 8" in L388 and L391 has been changed as "*Figures 9*".

54. "around 136 mm, 88 mm and 40 mm" in L393 has been changed as "*around 136 mm for monthly PET, 88 mm for monthly PT and 40 mm for monthly PE*".

55. "*generally*" in L388 and L391 has been inserted before "characterized" in L394.

56. "Figures 8e, 8g, and 8h" in L394 has been changed as "*Figures 9e, 9g, and 9h*".

57. "Figures 8" in L388 and L395 has been changed as "*Figures 9*".

58. "(Figures 8 and 9). As depicted in Figures 9a and 9c, the global mean annual PT/PET and PE/PET were 40% and 60%, respectively" in L397-399 has been changed as "*(Figures 9 and 10). As depicted in Figures 10a and 10c, the global mean annual PT/PET and PE/PET were 41% and 59%, respectively*".

59. "63% and 51%" in L400 has been changed as "*62% and 50%*".

60. "Overall, the annual PT/PET (PE/PET) had … controlled by PT (PE)." in L400-403 has been changed as "*Overall, the annual PT/PET (PE/PET) had evident regional differences, and was above 53% (below 47%) in Tropical, Temperate and Cold regions but below 12% (above 88%) in other two climate regions (Figures 10a and 10c). These indicated that the annual PET in Tropical, Temperate and Cold regions (Arid and Polar regions) was controlled by PT (PE)*".

61. "Figure 9" in L405 and L407 has been changed as "*Figures 10*".

62. "60% and 40%" in L408 has been changed as "*59% and 41%*".

63. "8" in L410, L413-415, L417, and L418 has been changed as "*9*".

64. "Globally, …(Figures 10a, 10c and 10e)" has been changed as "*Globally, annual PET and PT significantly (p<0.05) increased by 1.26 mm/yr and 1.27 mm/yr, respectively, with a slight and insignificant decrease in annual PE (Figures 11a, 11c and 11e)*".

65. "Figures 10b and 10c)" in L424 has been changed as "*Figures 11b and 11c)*".

66. "1.36" in L425 has been changed as "*1.32*".

67. "1.84" in L426 has been changed as "*1.83*".

68. "Figures 10a, 10c and 10e)" in L426 has been changed as "*Figures 11a, 11c and 11e)*".

69. "Figure 10" in L432, L434, L436 and L443 has been changed as "*Figure 11*".

70. "13" in L439 has been changed as "*17*".

71. "Figure 10g vs. Figures 9b and 9d" in L441 has been changed as "*Figure 11g vs. Figures 10b and 10d*".

72. "Figure 11" in L446, L449 and L458 has been changed as "*Figure 12*".

73. "Figures 11" in L447 and L452 has been changed as "*Figures 12*".

74. "*generally*" has been added after "but" in L450 and L455.

75. "June–October" in L451 has been changed as "*August and October*".

76. "PE" in L460 has been changed "*PET*".

77. "January–March for SH," in L460 has been changed "*January–March for SH, February–May for Tropical region*".

78. "Through considering … based on the SW model" in L464-465 has changed as "*Through considering the joint effects of various ET process-related factors, we have developed a new global PET dataset based on the SW model in this study.*"

79. "First, …impacts of elevated $CO_2$ on ET." in L498-475 has changed as "*First, the dataset considered more realistic land surface information, including spatial differences in LULC and vegetation parameters (e.g., canopy*

*height and $r_{smin}$), and time-varying LULC and LAI datasets, leading the new PET estimates more realistic. Second, the established SW model used in this study was based on more realistic physical processes and rendered the SW PET dataset an explicit physical significance, and meanwhile provided the PT and PE (which are crucial for understanding PET and ET. Third, a number of studies have found that the elevated atmospheric $CO_2$ concentration could exert clear impacts on ET process through controlling plant stomatal resistance (Gedney et al., 2006; Piao et al., 2007; Sun et al., 2014; Roderick et al., 2015; Milly and Dunne, 2016; Yang et al., 2019; Zhao and Cao, 2022). Through introducing a stress function of $CO_2$ concentration on $r_s^c$ our SW PET dataset is able to reflect impacts of elevated $CO_2$ on ET.*"

80. "In view of these advantages, …, e.g., rainfall, agriculture, drought, hydrology and biodiversity." in L476 has been changed as "*In view of these advantages, our global PET dataset can apply to multiple properties, e.g., the analysis of rainfall, agriculture, drought, hydrology and biodiversity.*"

81. "spatio-temporal differences" in L478 has been changed as "*impacts of spatio-temporal differences.*"

82. "(i.e., our datasets) benefit" in L492 has been changed as "*will benefit*".

83. "to local environment changes (including changes in LULC and vegetation), because this variable" has been changed as "*to local environment changes such as LULC, and PET*".

84. The paragraph of "*The canopy light extinction coefficient of $k_{ex}$, which represents a partitioning of radiant energy over vegetation canopy and soil surface, is a key factor affecting ecosystem carbon, water, and energy processes (Tahiri et al., 2006; Zhang et al., 2014). As a result, the accuracy of the SW model was believed to be associated with the $k_{ex}$ parameterization used in this study. Considering the physiological and morphological differences among terrestrial ecosystems (Emami-Bistghani et al., 2012; Zhang et al., 2014), we followed the popular biogeochemical models and remote sensing models of ET and gross primary productivity (e.g., Lund-Potsdam-Jena Dynamic Global Vegetation Model, and Vegetation Photosynthesis Model; Xiao et al., 2004; Sitch et al., 2003), and assumed this coefficient as a constant for every LULC type (Table S1). Despite that, it is notable that the $k_{ex}$ values vary with the growth of plant and/or vegetation coverages (Lindroth and Perttu, 1981; Aubin et al., 2000; Maddoni et al., 2001; Emami-Bistghani et al., 2012; Fauset et al., 2017). Emami-Bistghani et al. (2012) stated that with an increase of vegetation coverages, the $k_{ex}$ values decreased especially in early reproductive stage in sunflower cultivars. It suggested that the fixed $k_{ex}$ within the SW model existed limitations, potentially leading to uncertainties of the PET estimates. Implied by Tahiri et al. (2006), relative to the variable $k_{ex}$ values, the fixed values gave a less precise estimation of plant transpiration under irrigated maize.*" has been added before L535.

85. "Another part of ET, vegetation canopy interception is ignored in this study, although it can occupy a certain

proportion in total ET" in L536-537 has been changed as "*Another parts of ET, vegetation canopy interception and nighttime ET are ignored in this study. It is reported that vegetation canopy interception can occupy a certain proportion in total ET*".

86. "Undoubtedly, ignoring vegetation canopy interception … the model's physical mechanism." In L540-545 has been changed as "*Padrón et al. (2020) showed that on average the fraction of nighttime ET accounts for approximately 6.3% of the total ET informed by the FLUXNET2015 dataset while 7.9% based on multiple global models. Notably, this fraction may exceed to 15% in mountain forest with snowy and windy winter. Ignoring vegetation canopy interception and nighttime ET may underestimate PET (Tourula and Heikinheimo, 1998; Lawrence et al., 2007; Mu et al., 2011; Padrón et al., 2020; Singer et al.; 2021). Subsequent research will be done to integrate these two processes in the SW model to further enhance the model's physical mechanism.*"

87. "the observed PET and issue of … we should note the fact that" L549-552 has been changed as "*the observed PET, issue of non-closure of the energy balance at the EC system level and the gap-filling methods (e.g., marginal distribution sampling method; Reichstein et al., 2005). The energy balance-based criterion employed in this study was proved to be efficient to select unstressed days (Maes et al., 2019), but this method may still result in two uncertainties. First, we should note that*".

88. These sentences of "*Second, due to usual water deficits in arid regions, the EF threshold may exceed to the 95th percentile of EF. Thus, the identified unstressed days based on this criterion may actually include the stressed days. This potentially biased the PET estimates in arid regions. Through increasing the EF threshold (e.g., 96th and 98th; not shown here) for several EC sites in arid regions, we found that selecting 95th percentile as threshold has limited impact on the PET estimates.*" have been added after "estimated PET during this season" in L555.

89. "impacts of non-closure" in L559 has been changed as "*the impacts of the non-closure*".

90. "Although such processing … Elfarkh et al., 2020; Chen et al., 2022)" in L562-564 has been changed as "*Although such processing could reduce uncertainties, the imbalance of energy was not fully solved in our study and may lead to inevitable errors in the calibrated parameter of $r_{smin}$ and therefore the global PET estimates (Hu et al., 2009; Elfarkh et al., 2020; Chen et al., 2022). In this study, for maximizing the use of data, the marginal distribution sampling method was employed to fill the gaps in the EC LE measurements. However, it should be noted that even though the controlling thermodynamic and kinetic factors of the atmosphere are similar between the missing and retrieved moments (Jiang et al., 2022), the gap-filled LE may be of low confidence, especially when soil moisture has abrupt changes. To quantify potential impacts of the gap-filled values, the SW model was re-calibrated and re-validated against the data points without gap-filling. Relative to the SW model used in this*

*study, the new $r_{smin}$ and the validation metrics changed insignificantly (not shown here), suggesting that the uncertainties induced by the gap-filled LE were limited.*"

91. ", in which some" in L572 has changed as "*while with certain*".

92. "As a critical variable describing vegetation condition" in L573 has changed as "*As the reflection of vegetation growth*".

93. "further affects" in L574 has been revised as "*influence the*".

94. "the performance of this LAI product was slightly unsatisfactory in grassland plots compared to other products" in L577 has changed as "*this LAI product slightly underperformance over grassland compared to other products*"
*95. This sentence of "Considering that this LAI product was based on the 8-day maximum value composite for removing impacts of cloudy days, the LAIe (based on EQ. 3b) was potentially larger than its authentic value due to some leaves covered by rain or snow. Thus, from EQ. 3b, $r_s^c$ may be slightly underestimated, leading to an overestimation in PT and PET." has been inserted after "(Liu et al., 2018)" in L588.*

96. "misclassification" in L583 has been revised as "*the misclassification*".

97. "The reconstructed global vegetation canopy height has some limitations, which may come from (1) uncertainties of" in L585 has been revised as "*The reconstructed global vegetation canopy height also has limitations, which may raise from (1) uncertainties in the*".

98. "neglect of … CRO and GRA heights" in L587-589 has changed as "*neglecting the spatial differences in CRO and GRA heights and using an alternative specific value, and (3) not considering the inter-annual changes in the FR and SHRB canopy heights and the intra-annual cycle in the CRO and GRA heights.*"

99. "Finally, these limitations exerted adverse impacts on accuracy of the PET estimates." in L589 has changed as "*These limitations undermine the accuracy of the PET estimates.*"

100. "evident discrepancies" in L590 has been revised as "*the discrepancies*".

101. "could partly" in L591 has been changed as "*may*".

102. "the remaining uncertainties would likely propagate into the PET estimates." in L592 has been changed as "*there is still likelihood that the remaining uncertainties might be propagated into the PET estimates. In this study, even though soil does experience water stress, we assumed that the soil water supply for ET was unconstrained in estimating PET. As a result, the two conditions with and without soil water stress corresponded to different meteorological variables, when considering land-atmosphere interaction (Crago and Crowley, 2005; Kahler, and Brutsaert, 2006; Aminzadeh et al., 2016; Maes et al., 2019). For example, air temperature under the unstressed condition would like to be lower than that under the stressed condition, because of the lower sensible heating and*

*the stronger evaporative cooling from the wetter land surface to atmosphere (Maes and Steppe, 2012; Maes et al., 2019). Thus, the mismatch between the assumption of no soil water stress and the observed meteorological variables would like to introduce biases into our PET estimates.*"

103. "(1) it provides more realistic PET … (3) it can reflect impacts of elevated $CO_2$ on PET" in L604-607 has changed as "*(1) it provides more realistic the PET estimates by clearer physical processes, since we take the spatial differences and temporal changes of land surface properties into consideration, (2) it provides not only PET estimates but also PT and PE, and moreover (3) it can take the impacts of elevated $CO_2$ into PET estimation*".

104. "*Aminzadeh, M., Roderick, M. L. and Or, D.: A generalized complementary relationship between actual and potential evaporation defined by a reference surface temperature. Water Resources Research, 52, 385–406, 2016.*" has been added in L627.

105. "*Aubin, I., Beaudet, M. and Messier, C.: Light extinction coefficients specific to the understory vegetation of the southern boreal forest, Quebec. Canadian Journal of Forest Research, 30, 168–177, 2000.*" has been added in L637.

106. "*Cheng, W., Dan, L.i., Deng, X., Feng, J., Wang, Y., Peng, J., Tian, J., Qi, W., Liu, Z., Zheng, X., Zhou, D., Jiang, S., Zhao, H. and Wang, X.: Global monthly gridded atmospheric carbon dioxide concentrations under the historical and future scenarios. Scientific Data, 9(1), https://doi.org/10.1038/s41597-022-01196-7, 2022.*" has been added in L656.

107. "*Crago, R. and Crowley, R.: Complementary relationships for near-instantaneous evaporation, Journal of Hydrology, 300, 199–211, 2005.*" has been added in L659.

108. "*Emami-Bistghani, Z., Siadat, S. A., Torabi, M., Bakhshande, A., Alami, S. K. and Shiresmaeili, H.: Influence of plant density on light absorption and light extinction coefficient in sunflower cultivars. Research on Crops, 13(1), 174–179, 2012.*" has been added in L681.

109. "*Fauset, S., Gloor, M. U., Aidar, M. P. M., Freitas, H. C., Fyllas, N. M., Marabesi, M. A., Rochelle, A. L. C. A., Shenkin, Vieira, S. A. and Joly, C. A.: Tropical forest light regimes in a human-modified landscape. Ecosphere 8(11): e02002. https://doi.org/10.1002/ecs2.2002, 2017.*" has been added in L688.

110. "*Jiang, Y., Tang, R. and Li, Z. L.: A physical full-factorial scheme for gap-filling of eddy covariance measurements of daytime evapotranspiration. Agricultural and Forest Meteorology 323, 109087, https://doi.org/10.1016/j.agrformet.2022.109087, 2022.*" has been added in L767.

111. "*Kahler, D. M. and Brutsaert, W.: Complementary relationship between daily evaporation in the environment and pan evaporation, Water Resour. Res., 42, W05413, https://doi.org/10.1029/2005WR004541, 2006.*" has been

added in L776.

112. "*Lindroth, A. and Perttu, K.: Simple calculation of extinction coefficient of forest stands. Agricultural Meteorology, 25, 97–110, 1981.*" has been added in L807.

113. "*Maes, W. H. and Steppe, K.: Estimating evapotranspiration and drought stress with ground-based thermal remote sensing in agriculture: a review. Journal of Experimental Botany, 63, 4671–4712, 2012.*" has been added in L840.

113. "*Maki, T., Ikegami, M., Fujita, T., Hirahara, T., Yamada, K., Mori, K., Takeuchi, A., Tsutsumi, Y., Suda, K. and Conway, T.J.: New technique to analyse global distributions of CO2 concentration and fluxes from non-processed observational data. Tellus B: Chemical and Physical Meteorology, 62(5), 797–809, https://doi.org/10.1111/j.1600-0889.2010.00488.x, 2010.*" has been added in L842.

114. "*Maddoni, G. A., Otegui, M. E. and Cirilo. A. G.: Plant population density, row spacing and hybrid effects on maize canopy architecture and light attenuation. Field Crops Research, 71, 183–193, 2001.*" has been added in L863.

115. "*Nakamura, T., Maki, T., Machida, T., Matsueda, H., Sawa, Y. and Niwa, Y.: Improvement of atmospheric CO$_2$ inversion analysis at JMA, A31B-0033. In Proceedings of the AGU Fall Meeting, San Francisco, CA, USA, 14–18 December 2015.*" has been added in L875.

116. "*Padrón, R. S., Gudmundsson, L., Michel, D. and Seneviratne, S. I.: Terrestrial water loss at night: Global relevance from observations and climate models. Hydrology and Earth System Sciences, 24(2), 793–807, 2020.*" has been added in L890.

117. "*Reichstein, M., Falge, E., Baldocchi, D., Papale, D., Aubinet, M., Berbigier, P. and Valentini, R.:On the separation of net ecosystem exchange into assimilation and ecosystem respiration: review and improved algorithm. Global Change Biology 11(9), 1424–1439, 2005.*" has been added in L917.

118. "*Sitch, S., Smith, B., Prentice, I. C., Arneth, A., Bondeau, A., Cramer, W., Kaplan, J. O., Levis, S., Lucht, W., Sykes, M. T., Thonicke, K. and Venevsky, S.: Evaluation of ecosystem dynamics, plant geography and terrestrial carbon cycling in the LPJ dynamic global vegetation model. Global Change Biology, 9(2), 161–185. 2003.*" has been added in L936.

119. "*Tahiri, A.Z., Anyoji, H. and Yasuda, H.: Fixed and variable light extinction coefficients for estimating plant transpiration and soil evaporation under irrigated maize. Agricultural Water Management, 84, 184–192, 2006.*" has been added in L969.

120. "*Xiao, X. M., Zhang, Q. Y., Braswella, B., Urbanskib, S., Boles, S., Wofsy, S., Moore, B. III. and Ojima, D.:*

*Modeling gross primary production of temperate deciduous broadleaf forest using satellite images and climate data. Remote Sensing of Environment, 91(2), 256–270, 2004.*" has been added in L1027.

121. "*Zhang, L., Hu, Z., Fan, J., Zhou, D. and Tang, F.: A meta-analysis of the canopy light extinction coefficient in terrestrial ecosystems. Frontiers of Earth Science, 8, 599–609.*" has been added in L1066.

122. Figure 1 has been redrawn, such as

123. Figure 2 has been redrawn, such as

[Figure]

124. Figure 3 has been changed as,

[Figure]

**Figure 3:** Schematic description of the energy partitioning for a canopy with the SW model.

125. Figure 4 has been changed as,

126. Figure 5 has been changed as,

127. Figure 6 has been changed as,

128. Figure 7 has been changed as,

129. Figure 8 has been changed as,

130. Figure 9 has been changed as,

130. Figure 10 has been changed as,

131. Figure 11 has been changed as,

132. Figure 12 has been changed as,

133. Table1 has been changed as,

| Datasets | | Basic information | | |
| --- | --- | --- | --- | --- |
| | | Spatio-temporal resolution and time coverage | Variables | Sources and references |
| Meteorological datasets | MSWX-Past | 0.1º × 0.1º; 3-hourly; 1979−present. | Mean, maximum and minimum temperatures, *RH*, *u* at 10 m, downward shortwave radiation and downward longwave radiation. | http://www.gloh2o.org/mswx/; Beck et al. (2022) |
| | CRU TS4.06 | 0.5º × 0.5º; monthly; 1901−2021. | Mean, maximum and minimum temperatures, cloud cover and *ea*. | https://crudata.uea.ac.uk/cru/data/hrg/; Harris et al. (2020) |
| | ERA-5 | 0.25º × 0.25º; monthly; 1959−present. | Mean, minimum, maximum and dewpoint temperatures, surface pressure, *u* at 10 m, net shortwave radiation and net longwave radiation. | https://cds.climate.copernicus.eu/cdsapp#!/home/; Hersbach et al. (2020) and Berrisford et al. (2021) |
| | MERRA-2 | 0.5º × 0.625º; monthly; 1980−present. | Mean, minimuma and maximuma temperatures, specific humidity, surface | https://disc.gsfc.nasa.gov/; Molod et al. (2015) |

| | | | |
|---|---|---|---|
| | | pressure, $u$ at 10 m wind speed, net shortwave radiation and net longwave radiation. | |
| GLASS AVHRR LAI | 0.05º × 0.05º; 8-day; 1981−2018. | LAI | http://www.glass.umd.edu/; Xiao et al. (2016, 2017) |
| GLASS-GLC | 5000 m × 5000 m; yearly; 1982−2015. | LULC | https://doi.org/10.1594/PANGAEA.913496; Liu et al. (2020b) |
| Saturated water content in soil | 0.0833º × 0.0833º; static. | Saturated water content in the first soil layer (i.e., 0–0.0451 m) | http://globalchange.bnu.edu.cn/research/soil5.jsp; Dai et al. (2019a, 2019b) |
| Forest canopy height from Potapov | 30 m×30 m; static. | Forest canopy height | https://glad.umd.edu/dataset/gedi/; Potapov et al. (2020) |
| Forest canopy height from Wang | 500 m × 500 m; static. | Forest canopy height | http://www.nsmc.org.cn/NewSite/NSMC/Home/Index.html; Wang et al. (2016) |
| Forest canopy height from Simard | 1000 m × 1000 m; static. | Forest canopy height | https://webmap.ornl.gov/wcsdown/dataset.jsp?ds_id=10023; Simard et al. (2011) |
| Forest canopy height from Lang | 0.5º × 0.5º; static. | Forest canopy height | https://doi.org/10.5281/zenodo.5704852; Lang et al. (2021) |
| SPAM V2.0 | 0.0833º × 0.0833º; static. | Cropland distribution map | https://doi.org/10.7910/DVN/PRFF8V; International Food Policy Research Institute (2019) |
| Cropland height | Static | Height for various cropland | Details in Table S4; Allen et al. (1998) |
| GRA and tundra height | Static | Typical height for the 5 CSCS-based GRA groups | Details in Table S3 |
| $CO_2$ concentration from Cheng | 1º × 1º; Monthly; 1850−2013 | $CO_2$ concentration | https://doi.org/10.5281/zenodo.5021361; Cheng et al. (2022) |
| GCD $CO_2$ concentration | 2º × 2º; Monthly; 1985−2021 | $CO_2$ concentration | https://www.data.jma.go.jp/ghg/kanshi/co2data/co2_mapdata_e.html; Nakamura et al. (2015) |

---

## Author Response (AR2)

**RESPONSES TO COMMENTS FROM REFEREE #1**

The manuscript has been much improved, but some of my comments and suggestions were not well addressed. I'd like to remind the authors that all responses should be incorporated into the main-body manuscript or supplementary file.

**Responses:** We thank this reviewer very much for the valuable comments and suggestions, which are believed to be very useful for us to improving the study. Now, we have revised this manuscript, and the detailed information could be found below and the revised version.

(1) Figure 1 is of poor quality should be improved in terms of its resolution and size of font.

**Responses:** Thanks for your comment. We have redrawn this figure (see below or the revised manuscript). In this redrawn figure, the size of font has been enlarged. We have checked the resolution of this figure, and it has a resolution of 600 dpi. Notably, when the word document was saved as a PDF format (all the files should be submitted according to the journal's requirements), the resolution may become lower.

[Figure]

**Figure 1: Locations of the used EC sites in this study over Köppen-Geiger (KG) climate regions (Beck et al., 2018). International Geosphere-Biosphere Programme (IGBP) classification system: CRO—cropland; GRA—grasslands; DBF—deciduous broadleaf forest; EBF—evergreen broadleaf forest; ENF—evergreen needleleaf forest; MF—mixed forest; CSH—closed shrubland; WSA—woody savannah; SAV—savannah; OSH–open shrubland. GLASS-GLC classification system: FR—ENF, EBF, DNF, DBF and MF; SHRB—SAV CSH, OSH and WSA; GRA; CRO.**

(2) The responses to my last comment 5 are not satisfying. I cannot agree with the authors. If the physical full-factorial scheme of Jiang et al. is of low confidence in filling ET gaps, the gap-filled ET by the marginal distribution sampling method (a look-up table method) should be of much lower confidence and thus cannot be applied in the authors' study. The authors should make a more objective and positive appraisal of the physical full-factorial scheme of Jiang et al.

**Responses:** Thanks for your comment. We are sorry for that our responses and revision confused the referee. Originally, we would like to state that the gap-filled ET by the marginal

distribution sampling method (***rather than the physical full-factorial scheme of Jiang et al.***)

may introduce uncertainties into our study. Then, we re-calibrated and re-validated the SW

model against the data points without gap-filling, and found that the gap-filled ET by the

marginal distribution sampling method had limited impacts our study. We agree with the referee

that the physical full-factorial scheme of Jiang et al. is better than the marginal distribution

sampling method. Now, for removing the confusion, we have reorganized these sentences, such

as "*In this study, for maximizing the use of data, the MSD method was employed to fill the gaps*

*in the EC LE measurements. However, we should note that if the controlling thermodynamic*

*and kinetic factors of the atmosphere and soil moisture conditions are different between the*

*missing and retrieved moments, the gap-filled LE based on the MSD method may be of low*

*confidence, especially when soil moisture has abrupt changes (Jiang et al., 2022). Recently,*

*Jiang et al. (2022) developed a physics-based full-factorial scheme to fill gaps in ET from EC*

*observations, and found that the gap-filled ET with this scheme showed higher confidence*

*relative to the existing typical gap-filling methods. Therefore, to reduce the uncertainties from*

*the MSD-based gap-filled LE, the physics-based full-factorial scheme could be a good*

*candidate in the future to fill the ET gaps. Here, to quantify potential impacts of the MSD-based*

*gap-filled values, the SW model was re-calibrated and re-validated against the data points*

*without gap-filling. Relative to the SW model used in this study, the new $r_{smin}$ and the validation*

*metrics changed insignificantly (Figures S6, and S7), suggesting that the uncertainties induced*

*by the gap-filled LE were limited.*" **(L575-585)**

(3) The Figs. R2 and R3 in the responses to my last comment 5 should be incorporated into the

main-body manuscript or supplementary file.

**Responses:** Thanks. The Figs. R2 and R3 (i.e., Figures S6 and S7 in the supplementary materials) has been added in the supplementary file. Please see the revised supplementary materials or below.

[Figure]

**Figure S6: The calibrated $r_{smin}$ based on all data and data without gap-filling.**

[Figure]

**Figure S7: Validation results for the SW model calibrated using the data points without gap-filling. (a): Comparison for daily PET at all of 96 sites. (b-e): Comparison for daily PET for each LULC. (f): Comparison for site mean PET at all of 96 sites. (g-j): Comparison site mean PET for each LULC.**

(4) Most (rather than a small fraction) of the responses to my last comment 6 should be incorporated into the main-body manuscript.

**Responses:** Thanks for your suggestion. We have added most of the responses to your last comment 6 into the revised manuscript, please see the revision or below, such as "*On average, the fraction of ETn accounts for approximately 6.3% of the total ET informed by the FLUXNET2015 dataset while 7.9% based on multiple global models (Padrón et al. (2020). This fraction may exceed to 15% in mountain forest with snowy and windy winter. Despite that, to accurately represent the ETn process is still difficult to date, mainly because the related controlling mechanisms are still not clear (Han et al., 2021). For example, Novick et al (2009) and Groh et al (2019) found that VPD and wind speed had a significant impact on ETn, while Groh et al. (2019) stated that the contributions of night dew could not be ignored. As an important component of ETn, the nighttime transpiration is not only related to the incomplete stomatal closure (Dawson et al., 2007; Duursma et al., 2019) but also the circular regulation of nighttime water uses by plants (De Dios et al, 2015). However, how the environmental factors alter nighttime transpiration is still disputed. Dawson et al. (2007) and Moore et al. (2008) reported a positive correlation between nighttime transpiration and VPD and soil moisture content, while Barbour and Buckley (2007), Phillips et al. (2010b) and De Dios et al. (2015) found no or negative correlation between nighttime transpiration and the two variables aforementioned. Moreover, the biological factors (e.g., plant species and ecosystem types) can also significantly influence nighttime transpiration (O'keefe and Nippert, 2018; Zeppel et al., 2014). Therefore, to establish a common model for estimating ETn across various ecosystems remains challenging. All in all, ignoring vegetation canopy interception and ETn may*

*underestimate PET (Tourula and Heikinheimo, 1998; Lawrence et al., 2007; Mu et al., 2011; Padrón et al., 2020; Singer et al.; 2021). Subsequent research will be done to integrate these two processes in the SW model to further enhance the model's physical mechanism.*" **(L540-555)**

(5) The responses to my last comment 7 are not satisfying. A day in arid areas may not be identified to have no water limits even though its corresponding EF exceeded the 95th (or 98th) percentile EF threshold mainly because the soil moisture can hardly reach the field capacity or saturation due to the very limited precipitation. It is not relevant to the percentile!

**Responses:** Thanks for your comment. Now, we have revised the description according to your comment, please see the revised manuscript or below, such as "*Second, due to frequent water deficits in arid regions, the EF threshold may exceed the 95th percentile. What is more, there may be no unstressed days in extreme arid regions, mainly because the soil moisture can hardly reach the field capacity or saturation due to the very limited precipitation. Thus, the identified unstressed days using the energy balance-based criterion may actually include the stressed days in arid regions, and potentially biased the PET estimates.*" **(L567-570)**

(6) None of the responses to my last comment 12 was incorporated into the main-body manuscript!!! The authors should at least discuss the potential of the work of Peng et al. (2022) that developed for the first time a practical method for global estimates of 500 m daily aerodynamic roughness length in improving the ET and PET estimations.

**Responses:** Thanks for your comment. Now, we have revised the description, please see the

revised manuscript or below, such as "*In recent, Peng et al. (2022) proposed a practical method for global estimates of 500 m daily aerodynamic roughness length with a combination of machine learning techniques, wind profile equation, observations from 273 sites and MODIS remote sensing data. Their results showed that the random forest model could well reproduce the magnitude and temporal variability of daily aerodynamic roughness length at most sites for all land cover types. We believed that the aerodynamic roughness length produced by this method has a potential to replace vegetation canopy height as an input to run the SW model, and thus reduce the vegetation canopy height-related uncertainties aforementioned and improve the accuracy of the PET estimates.*" **(L604-610)**

(7) Section 2.2.3 and Section 3. The root-mean-square-error (RMSE, besides or except the ubRMSE) should be introduced to measure the model performance.

**Responses:** Thanks for your suggestion. The ubRMSE has been replaced by the RMSE in this revision. Please see the revised manuscript.

**A LIST OF ALL RELEVANT CHANGES MADE IN THE REVISION**

1. "unbiased Root-Mean-Square-Error (RMSE)" in L295 has been changed as "Root-Mean-Square-Error (RMSE)".

2. The equation 4b in L299 has been changed as "$RMSE = \sqrt{\frac{\sum_{i=1}^{N}(S_i - O_i)^2}{N}}$".

3. "ubRMSE" in L335, L338, L346, L351, L365 and L366 has been changed as "RMSE".

4. "0.53" in L347 has changed as "0.54".

5. "2010" in L506 has been changed as "2010a".

6. "nighttime ET" in L546 has been changed as "nighttime ET ($ET_n$)".

7. "Padrón et al. (2020) …enhance the model's physical mechanism." in L550-566 has been changed as "On average, the fraction of ETn accounts for approximately 6.3% of the total ET informed by the FLUXNET2015 dataset while 7.9% based on multiple global models (Padrón et al., 2020). This fraction may exceed to 15% in mountain forest with snowy and windy winter. Despite that, to accurately represent the $ET_n$ process is still difficult to date, mainly because the related controlling mechanisms are still not clear (Han et al., 2021). For example, Novick et al (2009) and Groh et al (2019) found that VPD and wind speed had a significant impact on $ET_n$, while Groh et al. (2019) stated that the contributions of night dew could not be ignored. As an important component of ETn, the nighttime transpiration is not only related to the incomplete stomatal closure (Dawson et al., 2007; Duursma et al., 2019) but also the circular regulation of nighttime water uses by plants (De Dios et al, 2015). However, how the environmental factors alter nighttime transpiration is still disputed. Dawson et al. (2007) and Moore et al. (2008) reported a positive correlation between nighttime transpiration and VPD and soil moisture content, while Barbour and Buckley (2007), Phillips et al. (2010b) and De Dios et al. (2015)

found no or negative correlation between nighttime transpiration and the two variables aforementioned. Moreover, the biological factors (e.g., plant species and ecosystem types) can also significantly influence nighttime transpiration (O'keefe and Nippert, 2018; Zeppel et al., 2014). Therefore, to establish a common model for estimating ETn across various ecosystems remains challenging. All in all, ignoring vegetation canopy interception and ETn may underestimate PET (Tourula and Heikinheimo, 1998; Lawrence et al., 2007; Mu et al., 2011; Padrón et al., 2020; Singer et al.; 2021). Subsequent research will be done to integrate these two processes in the SW model to further enhance the model's physical mechanism."

8. "marginal distribution sampling method" in L571 has been changed as "marginal distribution sampling method (MSD)".

9. "Second, due to … potentially biased the PET estimates." in L520-523 has been changed as "Second, due to frequent water deficits in arid regions, the EF threshold may exceed the 95th percentile. What is more, there may be no unstressed days in extreme arid regions, mainly because the soil moisture can hardly reach the field capacity or saturation due to the very limited precipitation. Thus, the identified unstressed days using the energy balance-based criterion may actually include the stressed days in arid regions, and potentially biased the PET estimates."

[revised manuscript text omitted]

23. Figure 1 has been redrawn, such as

[Figure]

KG climate classification

Tropical: Af | Am | Aw
Arid: BWh | BWk | BSh | BSk
Temperate: Csa | Csb | Csc | Cwa | Cwb | Cwc | Cfa | Cfb | Cfc
Cold: Dsa | Dsb | Dsc | Dsd | Dwa | Dwb | Dwc | Dwd | Dfa | Dfb | Dfc | Dfd
Polar: ET | EF

LULC
IGBP: ENF | EBF | DNF | DBF | MF | OSH | CSH | SAV | WSA | GRA | CRO
GLASS-GLC: FR | SHRB | GRA | CRO

△ AsiaFlux    ○ FLUXNET2015
□ LaThuile    ★ OzFLUX

24. Figure 4 has been redrawn, such as

25. Figure 5 has been redrawn, such as

26. Figure 6 has been redrawn, such as

26. Figure 7 has been redrawn, such as

---

## Author Response (AR3)

**RESPONSES TO TECHNICAL CORRECTIONS**

Dear Editor:

Thanks very much for considering our research for publishing in ESSD. Now, we have conducted the technical corrections (i.e., the colour schemes) according to the requirements of ESSD. The detailed information could be found below or the revised manuscript.

**A LIST OF ALL RELEVANT CHANGES MADE IN THE REVISION**

The colour schemes have been improved for allowing readers with colour vision deficiencies

to correctly interpret our findings.

1. Figure 7 has been changed as:

[Figure]

2. Figure 8 has been changed as:

[Figure]

3. Figure 9 has been changed as:

[Figure]

4. Figure 10 has been changed as:

[Figure]

5. Figure 11 has been changed as:

[Figure]

6. Figure 12 has been changed as: